# Assimilation of radar freeboard and snow altimetry observations in the Arctic and Antarctic with a coupled ocean/sea ice modelling system

Aliette Chenal[1,2], Gilles Garric[1], Charles-Emmanuel Testut[1], Mathieu Hamon[1], Giovanni Ruggiero[1], Florent Garnier[3], Pierre-Yves Le Traon[1,4]

[1] Mercator Océan International, Toulouse, 31400, France
[2] CNES, Toulouse, 31400, France
[3] LEGOS, OMP, Toulouse, 31400, France
[4] Ifremer, Plouzané, 29280, France

*Correspondence to*: Aliette Chenal (achenal@mercator-ocean.fr) and Gilles Garric (ggarric@mercator-ocean.fr).

**Abstract.** Sea ice and snow volume are essential variables for polar predictions, but operational systems still struggle to accurately capture their evolution. Satellite measurements now provide estimates of sea ice freeboard and snow depth. The combined assimilation of sea ice concentration (SIC), along-track altimetry radar freeboard data from Cryosat-2 and observations of snow depth from Cryosat-2 and SARAL is implemented in a multivariate approach in a global ¼° ocean/sea ice coupled NEMO4.2/SI3 model. A multivariate experiment, performed on two full seasonal cycles 2017–2018, is compared to a free (no assimilation) and a SIC-only assimilation simulations. The multivariate technique increases the sea ice volume, even in the absence of freeboard and snow measurements during summer, and rapidly changes the spatial patterns of ice and snow thicknesses in both hemispheres, in accordance with the assimilated observations. The sea ice volume from the multivariate approach compares better with independent (not assimilated) estimates from ICESat-2 and CS2SMOS or SMOS in both hemispheres. The multivariate system performs better in the Arctic than in Antarctica where the ice and ocean separate analyses are not designed to handle properly the strong interactions between upper oceanic layers and sea ice cover in the Southern Ocean. These results also confirm the importance of using variable snow and ice densities in a freeboard assimilation context. This study shows promising results for enhancing the capacity of assimilation systems to monitor the volume of sea ice and snow and paves the way for future satellite missions.

## 1 Introduction

In response to climate change, Arctic sea ice is continuing to decline and is regularly breaking historically low records, and, more recently, the entire year of 2023 showed the lowest sea ice extent in Antarctica ever seen in the satellite

record (Gilbert and Holmes, 2024) October 2020 was the lowest end-of-summer sea ice volume since 2010 in the Arctic (Perovich et al., 2020). Given the rapid transformations affecting sea ice due to climate change, sea ice monitoring is of the utmost importance. Assimilation techniques allow us to combine models and observations to improve our ability to monitor the ocean and sea ice state. Sea ice concentration (SIC) is currently assimilated in most sea ice data assimilation systems using different methods: nudging, Kalman filter variants, or 3DVAR variants (Uotila et al., 2019). However, one of the challenges in assimilating SIC is to extend the SIC information to other prognostic model variables such as sea ice thickness (SIT). Tietsche et al. (2013) concluded that in their Arctic model configuration, a proportional relationship between SIT and the SIC update was most effective for adjusting the modelled SIT. Massonnet et al. (2015) and Kimmritz et al. (2018) used the model covariances with a multivariate Ensemble Kalman Filter (EnKF) to update different sea ice variables, propagating the information from the observed SIC to the unobserved variables. Experiments have used the EnKF or variations of this multivariate scheme with multidata frameworks: both SIC and SIT products have been assimilated in the Arctic (e.g. Chen et al., 2024; Cheng et al., 2023; Williams et al., 2023). The assimilation methods can vary, but the assimilated SIT products are usually thin SIT from the European Space Agency's (ESA) Soil Moisture and Ocean Salinity (SMOS) mission, thick SIT measured by the ESA satellite mission CryoSat-2 (CS2), with two processing techniques available (Kurtz and Harbeck, 2017 or Ricker et al., 2014), or an observational product that statistically combines information from the two (CS2SMOS, Ricker et al., 2017).

Xie et al. (2016) found that assimilating SMOS thin SIT data had significant benefits for SIC and SIT modelling in some regions near the ice edge. Mu et al. (2018) combined the use of both SMOS thin SIT and CS2 SIT product in their assimilation system and obtained better results than the observation-only CS2SMOS product, demonstrating the added value of the model dynamics. The assimilation of CS2SMOS merged product (Xie et al., 2018) reduced model biases compared to the assimilated data, and results were in better agreement with independent datasets, with no degradation of other sea ice variables. Fritzner et al. (2019) compared the assimilation of SIC combined separately with either CS2 SIT, SMOS SIT, or a snow thickness (SNT) dataset in a short simulation and concluded that CS2 SIT provides the best long-term model improvements compared to SMOS SIT. They also found that SNT assimilation had a weaker effect on the model than SIT assimilation. Other teams methods updated SIT in the Arctic with nudging (Balan-Sarojini et al., 2021; Blockley and Peterson, 2018; Fritzner et al., 2018), with ensemble optimal interpolation (Lee and Ham, 2022, 2023) and with an enthalpy-adjusting scheme to ensure a consistent update of all sea ice variables (Liu et al., 2024). These numerous studies highlight that sea ice assimilation remains an active and evolving research area. The absence of a clear consensus on the optimal method reflects the complexity of balancing model uncertainties, data availability, and computational efficiency to achieve the best possible agreement with observations.

Cipollone et al. (2023) and Mu et al. (2020) implemented multidata and multivariate sea ice assimilation in global configurations, but with Arctic-only CS2, SMOS, and CS2SMOS SIT products. They both found their experiments to agree with in-situ data. Luo et al. (2021) implemented a multivariate assimilation system in Antarctica and successfully assimilated SIC and SMOS SIT. They had to inflate their atmospheric ensemble forcing, even though it was unnecessary in a similar

Arctic assimilation scheme, suggesting differences in the impact of sea-ice data assimilation between the two poles. They stated that the implementation of Arctic sea-ice data assimilation cannot be simply extended to the Antarctic.

SIT can be retrieved from altimeter radar freeboard (RFB) measurements by using hydrostatic equilibrium and taking into account the height of the snow penetrated by the radar wave, a medium where the radar velocity is modified (Garnier et al., 2022). The sea water, ice and snow densities and the snow depth above the ice are required for the RFB-SIT conversion, and the assumptions made on these variables result in a significant uncertainty in the sea ice volume products (Kern et al., 2015; Kwok and Cunningham, 2015). The snow layer accounts for most of the uncertainty in the calculation of

SIT from RFB (Garnier et al., 2021). The CS2 SIT products mentioned above use the Warren 99 (W99) snow climatology (Warren et al., 1999) or a modified version of it which is now known to be outdated and unreliable in most regions of the Arctic (Kern et al., 2015). Fiedler et al. (2022) is the first study to use the along-track CS2 RFB data in the Arctic, and to convert it into SIT using the modelled snow cover prior to the assimilation step. Their study results in a general improvement of the modelled SIT, with, in particular, a bias reduction in the Canadian Basin. This improvement extends into the summer

period, when no data is assimilated. However, they noted no substantial improvement in the Beaufort region due to a degradation of ice thicknesses below 1 m. Mignac et al. (2022) performed the same experiment, adding the SMOS SIT data to the along-tracks SIT computed from CS2 RFB and modelled snow, arguing that the SMOS SIT product performs better in thin ice areas of the Arctic. The thin SMOS ice assimilation was able to counteract the SIT overestimation that happens in the Arctic marginal seas when assimilating only CS2 products.

Other sources of uncertainty in the RFB-SIT conversion stem from the choice of ice and snow densities. The NEMO model uses constant snow and ice densities, whereas the observation products usually parametrize the ice density depending on the ice type (multi-year ice MYI, or first-year ice FYI, see Alexandrov et al., 2010) in the Arctic and on the season (see Kurtz and Markus, 2012) in the Antarctic. The choice of snow density varies in different SIT retrievals from RFB measurements, including options such as constant density, seasonally varying density, climatology-based density, or

modelled density. Kern et al. (2015) stated the importance of having well calibrated density for the ice and they recommended using seasonally varying snow density instead of a constant. Positive model biases in sea ice volume compared to satellite altimetry estimates have been attributed mainly to ice density differences (Bocquet et al., 2024). New efforts are currently being made to get fresh measurements of sea ice densities: Jutila et al. (2022) measured ice densities on average higher than the values from Alexandrov et al. (2010) for both the FYI and MYI, resulting in 12.4 % and 16.7 %

larger sea ice thickness values for FYI and MYI.

Knowing the large uncertainty associated with the sea ice volume products derived from RFB measurements, Sievers et al. (2023) directly assimilated the radar freeboard in the Arctic. In their assimilation scheme, they used a varying density for the ice, set as a function of the modelled salinity of the ice, and a linearly varying snow density depending on the season, following Mallett et al. (2020). The densities were not modified in the sea-ice model physics. They used the

modelled snow to convert the freeboard to ice thickness and they updated sea ice concentration and sea ice thickness through data assimilation. They compared the resulting sea ice thickness with in-situ data, showing improvements in some regions of

the Arctic and degradation in others, using a simulation without assimilation and another with assimilation of sea ice concentration only as references.

In this study, we use the operational Kalman filter scheme deployed in the production of global reanalysis and forecast at Mercator Ocean to implement a multivariate sea ice assimilation scheme with sea ice concentration (SIC), sea ice volume (SIV) and snow volume (SNV). In contrast to the usual ice assimilation where the SIC model variable (univariate) is updated using SIC observations (monodata), this approach aims to assimilate along-track radar freeboard and altimetric snow depth observations in addition to the SIC observations (multidata) and to update SIC, SIV and SNV model variables (multivariate). We use the same assimilation method for the Arctic and Antarctic. We aim to provide first answers to the following scientific questions:

- Does the multivariate/multidata approach provide added value over the widespread univariate/monodata method? What are the impacts of using altimetric radar freeboard and altimetric snow observations in addition to the SSMIS SIC data?

- Are the current parametrizations in sea ice models sufficient for accurate assimilation of radar freeboard and snow measurements?

- What challenges arise when applying the same sea ice assimilation scheme to both the Arctic and Antarctic, given their differing physical environments and ice dynamics?

Our work is in line with that of Sievers et al. (2023). However, we decided to assimilate RFB together with snow thickness observations to update the snow in addition to the sea ice variables at a global scale, i.e. including the Arctic and Antarctica. Moreover, we kept a coherent parametrization between the assimilation scheme and the sea ice model, so we used the model fixed snow and ice densities. Data using varying sea ice and snow densities are only shown in the figures indicatively for users of the original product.

Prior studies have shown that assimilating SIC alone significantly reduces concentration errors but yields limited improvement in ice thickness, despite strong correlations between both variables (Dulière and Fichefet, 2007; Lisæter et al., 2003). Moreover, there is no a priori link between SIC and the depth of the snow over sea ice. We therefore anticipate the following outcomes for each experiment: monodata/univariate SIC assimilation should improve modelled SIC but may degrade SIT and SNT due to the necessary adjustment for SIV and SNV implemented in the analysis scheme (Table 2). Conversely, the multidata/multivariate assimilation is expected to better fit all assimilated variables (SIC, RFB, SNT), but may impact SIC accuracy due to uncertain SIC-SIT/SNT covariances. The different spatio-temporal resolutions of SIC, RFB, and SNT (e.g. daily gridded SIC vs. sparse altimeter tracks with seasonal gaps) may also introduce uncertainty into the impact of assimilation. Finally, few studies have focused on the constraints of the ice/snow system by assimilation in Antarctica, a region where the interaction between the ice and the upper ocean is much more dynamic than in the Arctic. In regions of open water surrounded by sea ice — known as polynyas — the ice-ocean interactions are particularly strong (e.g. Cheon and Gordon, 2019; Kjellsson et al., 2015) and difficult to reproduce by models (Mohrmann et al., 2021). The

outcomes of the assimilation experiments could reveal whether improvements in SIC are offset by errors in SIT/SNT, how additional data sources interact, and how the scheme affects coupled ice–ocean behaviour.

We describe the modelling and assimilation components, the data assimilated in the analysis system, and the experimental design in Section 2. Section 3 focuses on the performances of the assimilation setup while section 4 presents a comparison with independent satellite observations. Section 5 discusses the main results and conclusions are given in section 6.

## 2 Analysis system and experimental design

### 2.1 Global ice-ocean coupled model configuration

We use the ocean/sea ice coupled model Nucleus for European Modelling of the Ocean (NEMO) version 4.2 (Madec et al., 2022), coupled to the Sea Ice modelling Integrated Initiative (SI3, Vancoppenolle et al., 2023). Simulations are run on a ¼ degree tripolar horizontal grid (Madec and Imbard, 1996) with 75 oceanic vertical levels. The atmospheric forcing is the European Centre for Medium-Range Weather Forecasts (ECMWF) ERA5 atmospheric reanalysis (Hersbach et al., 2020) with a 1h frequency

The sea ice model SI3 describes the ice and snow behaviour with assumptions that for dynamics, ice is a non-newtonian 2D continuum, whereas for thermodynamics, it is a mushy layer covered by snow. Subgrid variability is represented through 11 sea ice thickness categories, with fixed boundaries. Global prognostic variables in SI3 are the sea ice velocity u and its stress tensor σ, and quantities computed in each thickness category: sea ice concentration, sea ice and snow volume per unit area, sea ice and snow enthalpy per unit area, and sea ice salt content. The model uses constant densities for the sea water, sea ice and snow with respective values of 1026, 917 and 330 kg/m$^3$. Snow exclusively comes from the solid precipitations of the atmospheric forcing and disappears either by melting processes or by snow-ice conversion. The model accounts for snow-ice formation when snow is deep enough to depress the snow-ice interface below the sea level. Then seawater infiltrates and refreezes into the snow, creating a new ice layer whose thickness depends on the ice and snow densities (Fichefet and Maqueda, 1997; Vancoppenolle et al., 2023).

In this study, we use the adaptative elastic-viscous plastic rheology and a parametrization to represent landfast sea ice. The ice model component is called every 3 ocean timesteps, that is, every 30 minutes.

### 2.2 Assimilated observing network

| Observations | SIC SSMIS | RFB-LEGOS | SNOW-KaKu |
|---|---|---|---|
| Producer | EUMETSAT OSI-SAF | LEGOS | LEGOS |
| Temporal resolution | Daily | 20 Hz | Monthly → weekly (linear interpolation) |

| Temporal coverage | All-time | Winter: November to April in the Arctic; May to October in the Antarctic. | |
|---|---|---|---|
| Spatial resolution | 40 km (effective resolution); 25 km (grid resolution). | Along tracks | 12.5 km (grid resolution). |
| Spatial gaps | None (reprocessed). | Central Arctic (latitude > 88°N); in-between satellite tracks. | Central Arctic (latitude > 81.5°N); coastal areas. |
| nsel | 400 | 4000 | 400 |

**Table 1: Assimilated observation products and their specificities.**

### 2.2.1 Sea ice concentration SSMIS.

The observation data used for sea ice concentration (SIC) assimilation is the global daily reprocessed passive microwave dataset, measured with Special Sensor Microwave Imager / Sounder (SSMIS) satellites instruments, from the European Organization for the Exploitation of Meteorological Satellites (EUMETSAT) Ocean and Sea Ice Satellite Application Facility (OSISAF) OSI-450 (OSI SAF, 2022) (Table 1). Considering the large errors in satellite measurement in low SIC regions (Ivanova et al., 2015), we arbitrarily set to 0 the data values below 7.5%. Moreover, we only consider nominal data from the OSISAF algorithm, excluding data with coastal correction, interpolation, or climatology corrections. We use the daily- and spatially-varying "standard_error" provided with the dataset to construct the observation error for the assimilation but we inflate linearly the error to obtain a maximum of 25% in the Arctic (same value as Lellouche et al. (2021) and 40% in Antarctica, and we set a minimum value of the error to 1%.

### 2.2.2 Radar freeboard RFB-LEGOS

The "laboratoire d'etudes en géophysique et océanographie spatiales" (LEGOS) scientists have used along tracks measurement from the CS2 satellite to create a freeboard dataset (Guerreiro, 2017; Laforge et al., 2021). Thanks to hydrostatic equilibrium, freeboard can provide sea ice thickness values using information of snow depth, and water, ice and snow densities. Altimetry measurements measure radar freeboard (RFB) due to the slower velocity of the radar wave when travelling through the snow (see equations in Bocquet et al., 2023). Radar freeboard measurements depend on the radar speed reduction in the snow layer and are consequently not physical measurements. The ice/snow interface is therefore not necessarily underwater when the RFB is negative.

We multiply the RFB values by the SSMIS data to assimilate radar freeboard volume per unit area (RFBV) in consistency with volumetric prognostic model quantities. We use the uncertainty provided for each track as the observation error, constraining it to a range of 0.01 m to 5 m. The RFBV model equivalent is calculated from Bocquet et al. (2023) with constant sea water, sea ice and snow densities (Eq. 1).

$$RFB = \frac{\rho_w - \rho_{ice}}{\rho_w} \cdot SIT - \left(\frac{\rho_{snow}}{\rho_w} + (1 + 0.00051\rho_{snow})^{1.5} - 1\right) \cdot SNT = 0.106 \cdot SIT - 0.584 \cdot SNT \qquad (1)$$

180   We use the LEGOS data because it provides concomitant RFB and snow data in both hemispheres. We assimilate two modes of CS2 instruments: the Synthetic Aperture Radar (SAR) for offshore regions and SAR Interferometric (SARin) for coastal areas. Due to potential truncation problems with the filtering of RFB measurements, and to be able to use the same method across different spatial resolutions of the configuration, we kept the full scales of SAR and SARin measurements. The data are only available during winter in both hemispheres, November to April in the Arctic and May to

185 October in the Antarctic (Table 1). Apart from north of 88°N, CS2 satellite tracks cover the entire ice domain of each hemisphere in about a month: during each assimilation cycle, important areas remain unobserved, especially at lower latitudes (Antarctica).

### 2.2.3 Snow thickness SNOW-KaKu

   Snow thickness (SNT) data come from the KaKu LEGOS data (Garnier et al., 2021) and consist in the difference

190 between CS2 Ku-band altimetric measurements, reflected by the ice, and SARAL Ka-band altimetric measurements, reflected by the snow. The data are provided in monthly gridded files, available during the same winter periods as RFB, in each hemisphere (Table 1). A temporal linear interpolation is applied to get SNT data at each weekly analysis. Due to SARAL orbital characteristics, no data are available for latitudes higher than 81.5°N. The observation error used in the analysis comes from the monthly varying uncertainty supplied with the data, constrained to an arbitrary range of 0.01 m to 5

195 m. The snow data are assimilated as a thickness quantity, with the snow volume increment subsequently computed using the Kalman filter. Multiple processings are applied to the Ku-band CryoSat-2 measurements to create the SNOW-KaKu product: a degraded version of the SAR measurements (pseudo-LRM mode) is used to get a similar footprint as the SARAL-AltiKa measurements, a 25 km radius median smoothing is applied, and the data is gridded at a monthly frequency, as described by Garnier et al. (2021). However, the SNOW-KaKu product remains not fully independent from RFB-LEGOS measurements.

200 ## 2.3 Assimilation scheme

   The assimilation system is the one used in the current near real time operational system (Lellouche et al., 2021). The 7-day assimilation cycle proceeds as follows: firstly, the model runs for the full cycle length for a 'forecast' trajectory, resulting in a forecast state. Observations available during the cycle time are loaded and processed as needed, with special care taken to define the observation errors. Using the forecast output and an observation operator, model variables are

205 transformed into observation-equivalent variables that are consistent in space and time with the assimilated observations. Then, the analysis step produces 4D increments or model updates of the forecast trajectory. The increment depends on the innovation (observation minus model equivalent), weighted by the Kalman gain. We use a reduced-order Kalman filter derived from a singular evolutive extended Kalman (SEEK) filter (Brasseur and Verron, 2006; Lellouche et al., 2021). The Kalman gain is meant to balance the information from the model and the observations to get closer to the real ocean and sea

210 ice state: as such, it is based on the error covariance of the forecast and the observation errors. The model forecast error covariance is computed from a fixed ensemble of 4D ocean and ice state anomalies that vary seasonally.

The static anomalies are computed from a long simulation (2010-2020) without assimilation, using the same model configuration and parameters with respect to a 7-day running mean. This approach is based on statistical ensembles in which the ensemble of these anomalies is representative of the error covariances (Lellouche et al., 2013).

The increments at each model grid point are calculated independently in a local scheme, where a localization algorithm controls the spatial influence of observations. This approach helps to limit the impact of sampling noise on the increments. The radius of the localization scheme is set as the minimum between an arbitrary fixed distance of 176 km and a radius defined by the inclusion of a number of observations nsel (see the chosen nsel values in Table 1). The last step of the assimilation cycle is the Incremental Analysis Update (IAU) that allows us to gradually introduce the analysis increments into the model (Benkiran and Greiner, 2008). The model runs a second time over the 7-day cycle for a 'best' trajectory; and at each timestep a tendency term is added to the model variables in the prognostic equations. The tendency term comes from the increment, modulated by a distribution function (Lellouche et al., 2013).

The ice and ocean analysis are separate, which means that ocean covariances are used for the ocean variables only, and the same applies for sea ice variables. The ocean analysis is multivariate and multidata, using sea level anomaly datasets from satellite altimetry (SEALEVEL_GLO_PHY_L3_NRT_008_044, 2023), sea surface temperature (SST) from OSTIA ( Operational Sea Surface Temperature and Sea Ice Analysis, SST_GLO_SST_L4_NRT_OBSERVATIONS_010_001, 2023), and temperature and salinity vertical profiles from in situ ARMOR and CORA-REP measurements (INSITU_GLO_PHYBGCWAV_DISCRETE_MYNRT_013_030, 2024). The ocean observations are not assimilated under the sea ice in the original operational system. Following experiments to set up the new ice assimilation system, instabilities in the water column appeared in the Southern Ocean. To reduce these static instabilities, we activated the OSTIA SST assimilation under the ice to maintain the ocean temperature at the freezing point. We also stopped assimilating in situ data to the south of 60°S, regardless of the season, because the surface thermohaline properties were being durably modified on large spatial scales, despite the few profiles present. Assimilating these in situ data modified ocean stratification, causing upwellings of warm water at the surface and creating unrealistic open water areas within the sea ice cover.

Assimilation systems can be described by the terms monodata or multidata, depending on the number of observations assimilated. Two different methods exist for the assimilation system: univariate and multivariate. They refer to the number of variables in the Kalman filter state vector, determining for which variables the increments are created. In a univariate configuration, the Kalman filter runs for each observation to create only one increment. In a multivariate configuration, multiple analysis increments are created at once, using the model covariances to simultaneously correct a number of variables in a coherent manner. Hence, different assimilation systems could be defined: monodata/univariate, monodata/multivariate and multidata/multivariate.

In the univariate configuration, only a SIC increment is created, and only SIC observations are assimilated. In the multivariate configuration, the state vector is made of sea ice concentration SIC, sea ice volume SIV, snow volume SNV, radar freeboard volume RFBV, and snow thickness SNT. This multivariate configuration allows us to assimilate a larger variety of data and to update the modelled ice accordingly. It is not required to use observational data on each of the state

vector variables: when no data are given, the Kalman filter uses the model covariances to propagate the information from the observed variables to the unobserved ones. Similarly, the model covariances are used to create increments where and when there are data gaps in the assimilated observations. RFBV and SNT variables are included in the state vector due to the availability of observation datasets for these quantities. SIV and SNV are included because they are global prognostic variables of the ice model, essential for accurately describing the model state. SIC is included for both reasons.

The different variables updated in the sea ice assimilation cycle are listed in Table 2. The increments do not distinguish ice categories; they present total values aggregated over each grid cell. All increments are tempered by the IAU factor. The first updated model variable is the SIC. The analysis is created by adding the increment to the forecast: $SIC_a = SIC_f + SIC_{inc}$. Then, the total ice concentration is redistributed into each existing thickness category using a Gamma-type distribution commonly found in observed measurements (Petty et al., 2020; Toppaladoddi et al., 2023). This chosen distribution (with parameters k=2.0 and theta=0.4) adds most of the increment to the middle and smallest thickness categories and less to the extreme categories.

In the univariate system, all other updates are computed from this SIC increment: following Tietsche et al. (2013), the SIV is proportional to the sea ice concentration, with a constant varying depending on the hemisphere: $h_{SH}^* = 1m$ and $h_{NH}^* = 2m$. The SNV increment is set to zero in the univariate method. In the multivariate method, SIV and SNV increments come directly from the Kalman filter algorithm. The algorithm updates the total ice and snow volumes for each grid cell, and then redistributes the updates to the individual ice categories. For the SIV, the algorithm adjusts the SIT in each category, starting with the thinnest ice. This prioritizes melting thinner ice first when the ice volume increment is negative. Changes are applied proportionally to the analysis SIC in each category, ensuring larger changes in categories with greater ice surface area. If the change of thickness of a category exceeds its bounds, any excess or deficit in volume is transferred to the next thicker category, and this redistribution continues until the entire SIV increment is applied. The SNV update accounts for the forecast SNT, analysis SIC, and SNV increment. When the SNV increment is zero, corrections are still applied, aiming at maintaining a constant SNV even under varying SIC conditions. Redistribution preserves the snow distribution across ice categories by adjusting the forecast SNT by the same ratio in each category.

Then, the volumetric ice salinity and enthalpy are corrected in both methods by adjusting the previous ice salinity and enthalpy to the new ice volume $SIV_a = SIV_f + SIV_{inc}$. The volumetric snow enthalpy is also corrected following the same procedure. The updated volumetric ice salinity and enthalpy and the volumetric snow enthalpy are used to compute the ice salinity vertical profile, the salt mass content, and the snow and ice vertical temperature profile.

| Updated variable | Univariate method | Multivariate method |
|---|---|---|
| SIC | Increment | Increment |
| SIV | $SIV_{inc} = h^* \times SIC_{inc}$ | Increment |
| SNV | $SNV_{inc} = 0$ | Increment |
| Volumetric ice salinity | Computed from $SIV_{inc}$ and forecast value. | |

| Volumetric ice enthalpy | Computed from $SIV_{inc}$ and forecast value. | |
|---|---|---|
| Volumetric snow enthalpy | No update | Computed from $SNV_{inc}$ and forecast value. |

**Table 2: Variables updated during the assimilation cycle and their origin in both the univariate and multivariate systems.**

## 2.3 Experiments setup

To assess the impact of the multivariate and/or the multidata approach versus the more widespread SIC monodata/multivariate assimilation approach, we have not considered the most relevant approaches that can be combined with a single-variety or multi-variety approach and the use of data in multi-data or single-data mode. We performed a monodata/multivariate experiment assimilating the SIC OSISAF SSMIS product only with the multivariate assimilation system described previously. The results of this experiment are presented in supplementary material (Section 2) to let the article focus on the major differences brought by the innovative multidata/multivariate configuration. We then restricted the study to the comparison of the results using the monodata/univariate and the multidata/multivariate configurations. Three experiments have been performed to assess the performance of the assimilation and the impact of the multivariate approach:

● FREE: experiment without any assimilation, used as a baseline of the model capacities which has consistent biases in all sea ice variables due to model and forcing limitations, providing a baseline for evaluating the impact of assimilation.

● UNIVAR: experiment similar to the current operational system, using the previously described univariate SIC assimilation method. Assimilating SIC alone is expected to significantly reduce sea ice concentration errors but may induce unrealistic adjustments in sea ice thickness (SIT) and snow depth (SNT), due to uncertain or non-physical covariances.

● MULTIVAR: experiment with the multivariate assimilation scheme described previously, assimilating SIC, RFB and SNT observations, and updating the SIC, SIV and SNV model variables. Assimilating multiple variables is anticipated to improve agreement with all assimilated observations (SIC, RFB, SNT), though possibly at the cost of reduced SIC accuracy and increased risk of numerical or dynamical imbalances, especially in a coupled ice–ocean model.

Characteristics of the three experiments are summarized in Table 3. All three experiments were conducted over two full annual cycles, 2017 and 2018, covering the period from 14/12/2016 to 26/03/2019. Initial conditions are based on the reanalysis GLORYS12V1 (Lellouche et al., 2021).

| Experiment name | Assimilated data | Analysis increments | Updated model variables |
|---|---|---|---|
| FREE | None | None | None |
| UNIVAR | SSMIS | SIC | SIC, SIV |
| MULTIVAR | SSMIS, RFB-LEGOS, SNOW-KaKu | SIC, SIV, SNV, RFBV, SNT | SIC, SIV, SNV |

**Table 3: Experiments setup in terms of assimilated data, analysis increments and updated model quantities.**

# 3 Performances of the assimilation system

## 3.1 Sea ice concentration and sea ice leads

As expected, the two assimilation experiments outperform the FREE experiment during summertime in terms of sea
ice concentration coverage. In both hemispheres, FREE is not able to prevent excessive melting and shows a significant lack of sea ice, mainly in marginal areas, during July-October in the Arctic (i.e. Fig. 1(a) for July 2017) and in January-April in Antarctica (See Figures S1 and S2).

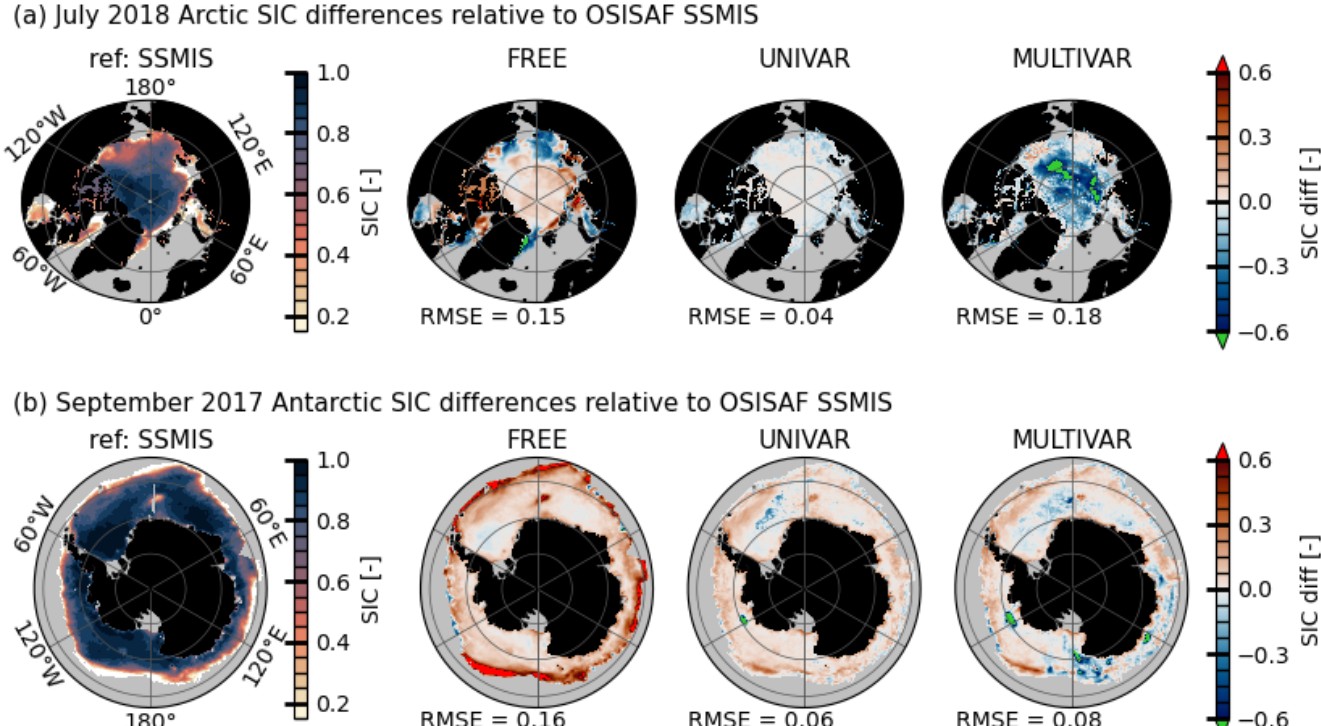

**Figure 1: July 2018 in the Arctic (a) and September 2017 in the Antarctic (b) maps of the sea ice concentration, representing the**
**observation SSMIS on the first column, and the difference between the experiments and the reference SSMIS observation on the following columns. The simulations are, in that order: FREE, UNIVAR and MULTIVAR. Root mean squared errors (RMS) are provided under each map.**

Maps of the sea ice concentration in the assimilated observations and their difference to the experiments are shown on Figure 1 for both hemispheres. The well-known Weddell Sea "Maud-rise polynya" that appeared in winter 2017 (Jena et
al., 2019) is not reproduced by the FREE experiment (Figure 1(b)). The UNIVAR and MULTIVAR experiments are able to reproduce this polynya. However, in the assimilated simulations, the Maud-rise polynya begins to take shape from June 2017, earlier than in the observations, and the system struggles to keep an ocean uniformly covered in ice in the Weddell Sea. Other polynyas are present in few locations around the Antarctic: in the Amundsen Sea offshore of Pine Island Bay at

120°W in the UNIVAR and MULTIVAR simulations (Figure 1(b)), and near Iselin Bank at 180°E in the Ross Sea in the MULTIVAR simulation. These events appear repeatedly during the ice freezing period in 2017 and 2018.

On the maps on Figure 1, sea ice concentration modelled by the UNIVAR simulation stands out and compares very well with the assimilated SSMIS dataset in the Arctic (RMSE of 0.04 in July 2018) and remains below the observation error in Antarctica (RMSE of 0.06 in September 2017). Multivariate assimilation of RFB and SNT data reduces the Arctic SIC compared to SSMIS, mainly in the central Arctic. This lower SIC in the central Arctic results in a RMSE of 0.18 for July 2018, the highest among the experiments. In that summer period, there are no RFB and SNT observations and the multivariate assimilation system creates the SIV and SNV increments from SIC observations and model covariances only. During the other months, the RMSE of 0.08 for the MULTIVAR simulation is lower, falling between the mean RMSEs of the UNIVAR and FREE simulations, which are 0.04 and 0.13, respectively. The Arctic mean RMSE of the UNIVAR and MULTIVAR simulations are similar in winter, but they differ in summer with the MULTIVAR simulation RMSE being 0.07 higher. In Antarctica, the FREE simulation presents mainly positive SIC biases in winter, particularly in the marginal ice zone (MIZ, defined by SIC values between 15% and 80%), and places the ice edge too far north compared to SSMIS observations (Figures 1 and S2) with mean RMSEs of 0.16 in September 2017 and 0.23 over the whole 2017-2018 months. The ice edge overestimation in the FREE experiment is corrected by the SIC assimilation in both UNIVAR and MULTIVAR simulations with comparable RMSEs of respectively 0.06 and 0.08 in September 2017 and the same values for the mean RMSEs over the whole 2017-2018 months.

We also assess the experiments on their ability to correctly reproduce the amount of open waters within the sea ice extent, referred to as "leads" hereafter. The area of sea ice leads offers valuable insights for predicting the Arctic sea ice extent (Zhang et al., 2018). The daily sea ice leads area timeseries are represented on Figure 2(a) in the Arctic and Figure 2(b) in the Antarctic. The sea ice leads area is computed by subtracting the sea ice area from the sea ice extent defined by cells where SIC>15%. We use two others different SIC datasets in order to quantify the spread among observations (Ivanova et al., 2015): the OSI-408 product (OSI SAF, 2017), derived from AMSR-2 satellite measurements and processed by the EUMETSAT OSISAF; and the Climate Data Record (CDR) dataset (Meier et al., 2017; Peng et al., 2013) from the National Snow and Ice Data Center (NSIDC). All SIC data are interpolated on the polar stereographic SSMIS grid and use a consistent continental mask, ensuring the same area coverage.

In the Arctic, the maximum lead surface area occurs in summer, more precisely at the beginning of the melting season. The daily surface area of leads peaks in July and then decreases with the retreat of the sea ice extent. The amount of leads remains constant from October to May in all the observations. In Antarctica, the lowest lead surface area is synchronous with the sea ice extent minimum in February-March. The observations then show an increase in leads area until its peak in November–December, corresponding to the first third of the melting season. The southern observational datasets show strong agreement regarding the minimum lead surface; but diverge as the lead area increases. In both hemispheres, NSIDC and SSMIS observations respectively display the smallest and the largest amount of leads. The FREE experiment shows the smallest amount of leads remaining outside the range of the observations for most of the year in both hemispheres,

and has a weaker seasonal amplitude in the Arctic than the assimilated experiments and SSMIS and AMSR2 estimates, but comparable to NSIDC's amplitude. Despite leads metrics that moderately resemble the observations on average in the FREE experiment, its Arctic RMSE of 0.15 on Figure 1(a) highlights inconsistencies in the modelled spatial patterns of sea ice concentration. The assimilation process rapidly and realistically increases the amount of leads in both the Arctic and Antarctic sea ice cover. The two assimilated experiments remain very close to the NSIDC leads area estimates during the northern hemisphere constant sea ice leads period, and they reproduce very well the rapid increase in lead surface area during spring. The UNIVAR experiment remains within the range of observational estimates throughout the year. The MULTIVAR simulation exhibits the highest amount of leads during the peak period in July, even higher than the SSMIS observations.

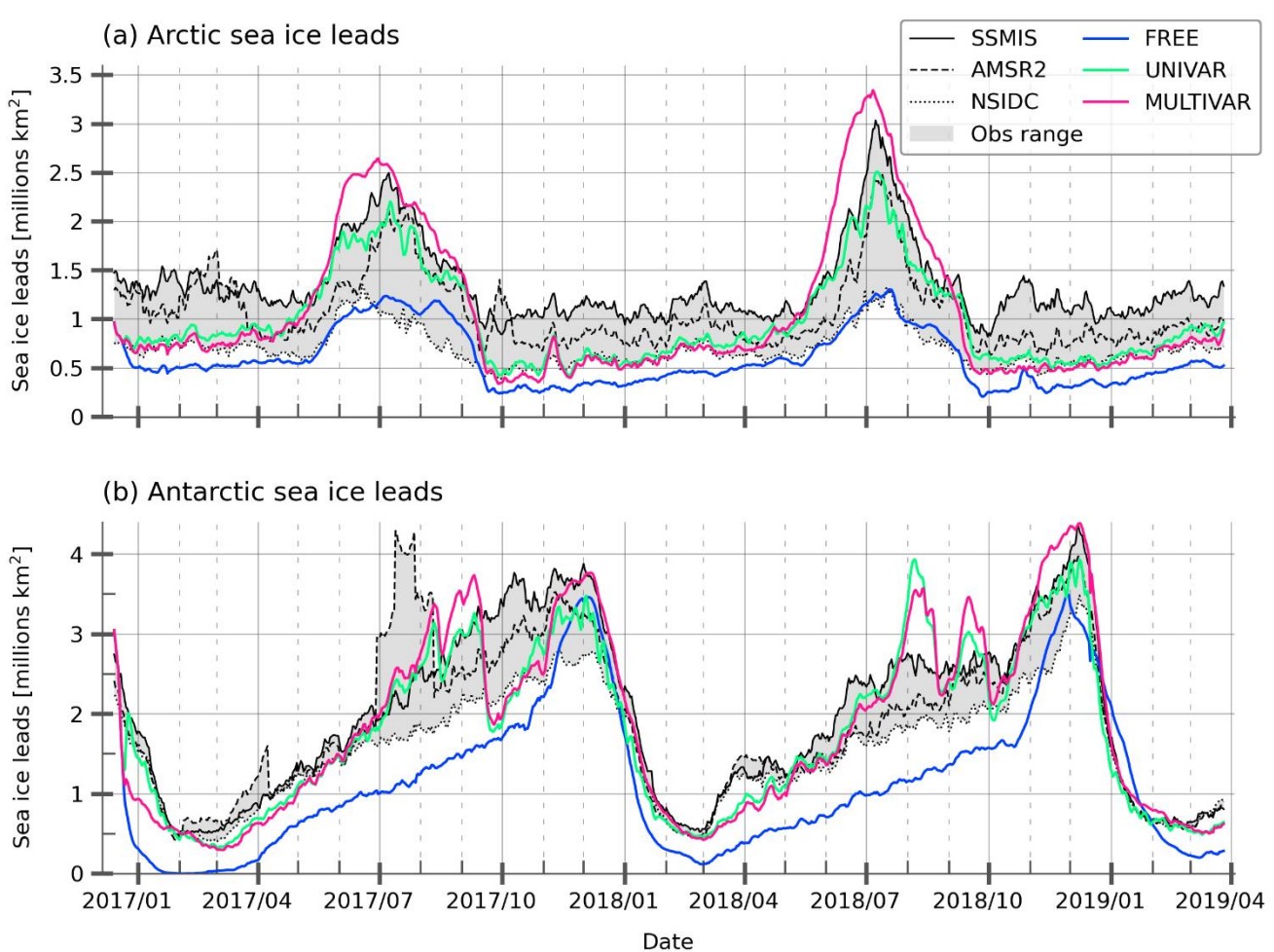

**Figure 2: Daily time evolution of Arctic (a) and Antarctic (b) surface covered by sea ice leads in millions of km² for SSMIS (black), AMSR2 (dashed black), NSIDC (dotted black) satellite data with the range covered by them (shaded grey) and for FREE (blue), UNIV AR (green) and MULTIVAR (pink) experiments.**

In Antarctica, both the UNIVAR and MULTIVAR experiments have a consistently higher sea ice leads area than the FREE experiment and are thus in better agreement with the observations. They correctly reproduce the minimum leads area and its maximum, with the MULTIVAR experiment showing the highest amount of leads during the peak period in early December, still coherent with the SSMIS observations. However, during the second half of the increase in lead surface, the assimilated experiments show significant fluctuations that exceed the range of the observations. The fluctuations are linked to the occurrence of localized low-SIC and thin ice areas in the ice cover, called polynyas when they become open-water areas.

In both hemispheres, the assimilation of SIC creates a larger lead area in the sea ice cover, in accordance with the SSMIS assimilated observations. The multivariate experiment alone even overestimates the quantity of leads during the seasonal maximum in the Arctic summertime. In the Antarctic, the two assimilated experiments generate a large number of polynyas that are not detected by the satellite observations, with the MULTIVAR experiment showing them more frequently and broadly across the region (Figure S2). While some smaller polynyas may go undetected in the observational data, the modelled polynyas are likely overestimated.

## 3.2 Snow volume

Figure 3(a) shows the probability density functions for snow thickness, radar freeboard using SAR mode, and radar freeboard using SARin mode, along with their model equivalents for the three experiments in the Arctic in April 2017. The SNOW-KaKu data in the Arctic present a zero-inflated bimodal, asymmetrical and positively skewed snow distribution with the first mode representing a snow thickness of 0 cm (no snow observed on the grid cell), and the second mode increasing in thickness as winter progresses and peaking at 13.6 cm in April 2017. The MULTIVAR snow distribution is very close to the Arctic SNOW-KaKu during winter (Figures S3 and S5) and matches perfectly in April. The UNIVAR and especially the FREE simulations accumulate excessive snow as winter progresses, leading to a positive bias by the end of the winter assimilation period as shown on Figure 3(a). The linear correlation (r-value) computed against the SNOW-KaKu observations in the Arctic results is consistently above 0.5 for MULTIVAR, peaking at 0.7 in December 2018 (Figure S6). The FREE and UNIVAR experiments exhibit systematic lower r-values, with the UNIVAR experiment having the lowest average correlation of 0.37. Compared to SNOW-KaKu estimates, the FREE and UNIVAR simulations present a spatially homogeneous overestimated snow thickness in Central Arctic and an underestimation in few areas such as north of the Canadian Archipelago, the east coast of Greenland, and in the Barents and Greenland seas (Figure 3(b)). This results in an excessive total snow volume of 1.24 $Mkm^3$ in the FREE experiment compared to that of 0.94 $Mkm^3$ estimated by SNOW-KaKu observations. In April 2017 (Fig. 3 (b)), the MULTIVAR simulation represents closely both the SNOW-KaKu spatial pattern and the total snow volume amount with 0.91 $Mkm^3$. This result is robust and remains valid for the other months of the year.

(a) April 2017 Arctic distributions

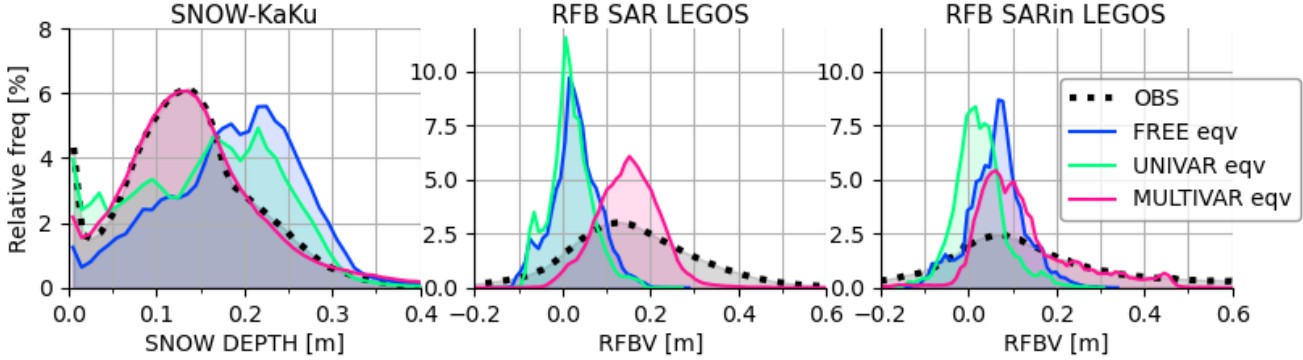

(b) April 2017 Arctic snow volume differences relative to SNOW-KaKu

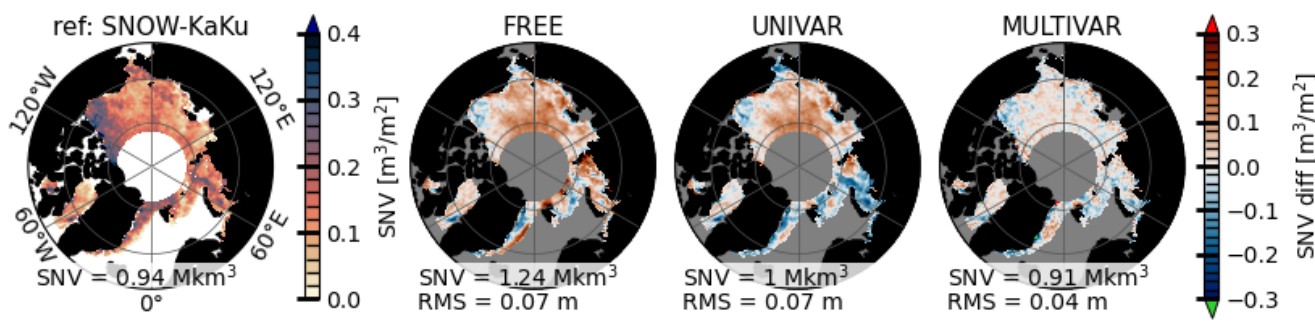

(c) April 2017 Arctic radar freeboard volume differences relative to RFB LEGOS

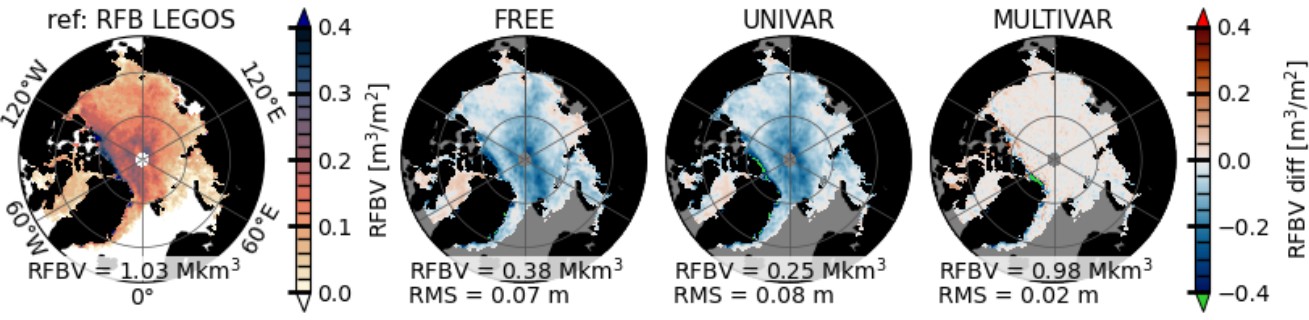

**Figure 3: Top panels (a): Probability density functions (%) of the snow thickness, the radar freeboard SAR and radar freeboard SARin observations (dotted black) and their model equivalent for the FREE (blue), UNIVAR (green) and MULTIVAR (pink) experiments. Total snow and RFB volumes values and root mean squared difference (RMS) are provided under each map.**

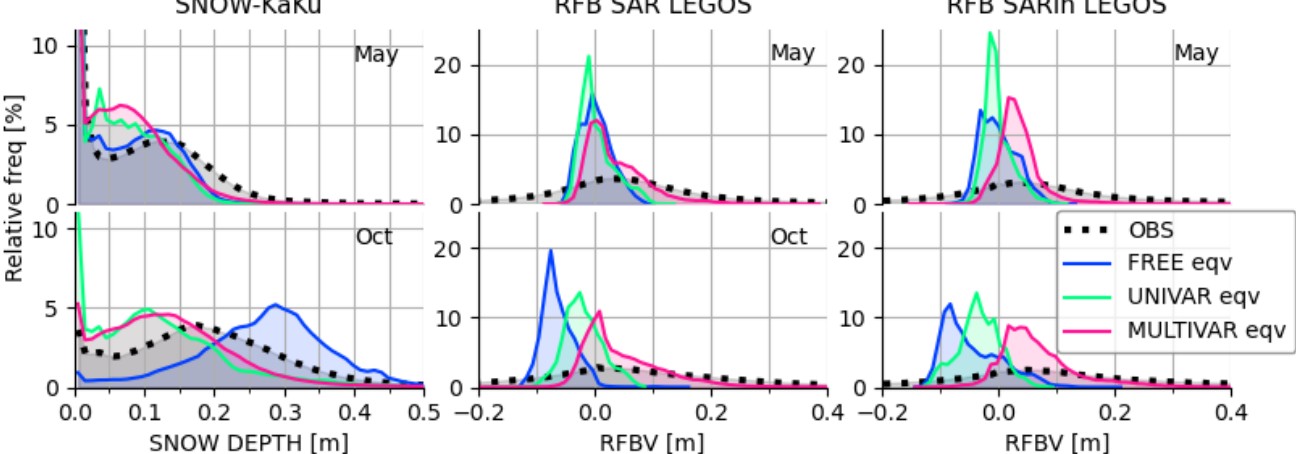

(a) May and October 2017 Antarctic distributions

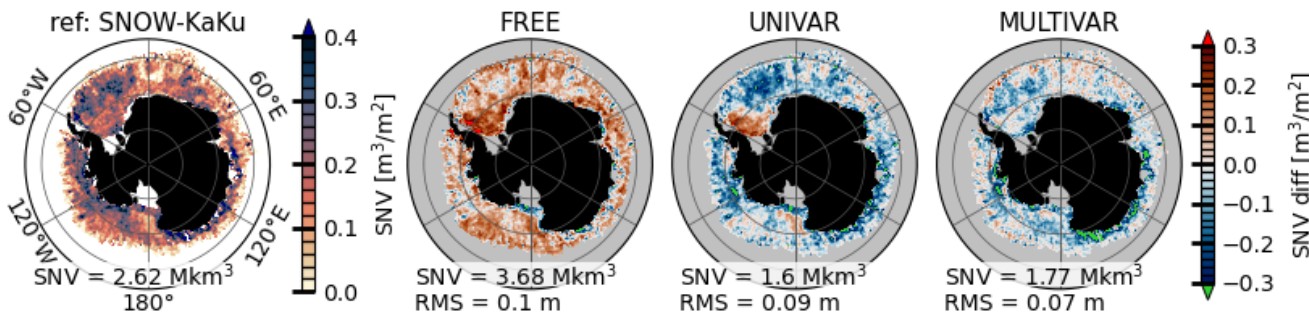

(b) October 2017 Antarctic snow volume differences relative to SNOW-KaKu

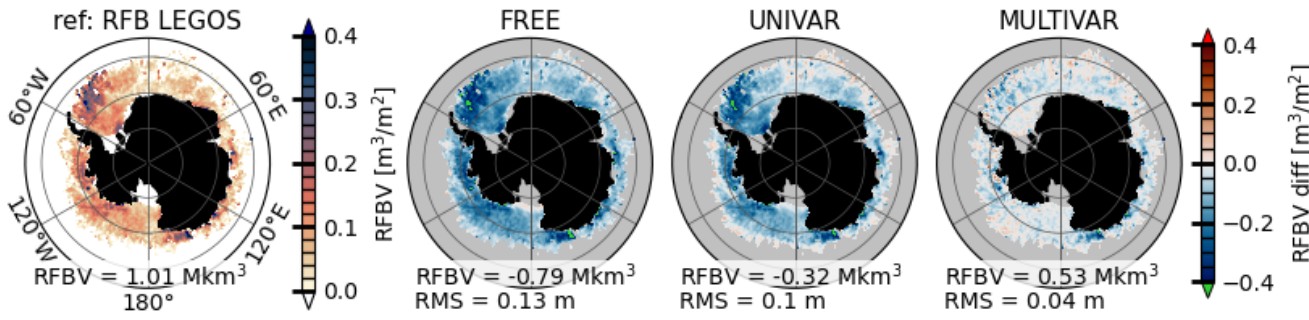

(c) October 2017 Antarctic radar freeboard volume differences relative to RFB LEGOS

**Figure 4: Top panels (a): Probability density functions (%) of the snow thickness, the radar freeboard SAR and radar freeboard SARin observations (dotted black) and their model equivalent for the FREE (blue), UNIVAR (green) and MULTIVAR (pink) experiments in the Antarctic for May and October 2017. Middle (b), resp. bottom (c), row panels: snow volume per unit area, resp. radar freeboard volume per unit area, from SNOW-KaKu, resp. RFB LEGOS, (first column) and differences with FREE, UNIVAR and MULTIVAR experiments in October 2017. Total snow and RFB volumes values and root mean squared difference (RMS) are provided under each map.**

In the Antarctic, the SNOW-KaKu data again exhibit a bimodal and positively skewed distribution, with a mode at 0.6 cm another at 11.6 cm in the first month of assimilation in May 2017 on Figure 4(a). As winter progresses, the second mode gets thicker and more frequent, peaking at 17.6 cm in October 2017. Among the simulations, the FREE experiment matches better the observations in May 2017 but then diverges the most from the observations, showing an increasing accumulation of snow as winter progresses, with a main mode 11.2 cm higher than the observed mode in October 2017. The UNIVAR and MULTIVAR experiments present lower snow thickness values compared to the observations during the whole 2017 and 2018 seasons, with main modes respectively 8.2 cm and 7.5 cm lower than the observed mode. The most significant snow positive biases in the FREE experiment are associated with thinner snow measurements in the SNOW-KaKu data, suggesting a thicker and more uniform snow cover, with a snow accumulation in the interior of the Weddell Sea, resulting in an excess of 1.06 million km$^3$ of snow compared to the SNOW-KaKu estimate (see Figure 4(b)). In comparison, the UNIVAR simulation presents a general thinner snow depth, maintaining however the accumulation in the Southwestern part of Weddell Sea. The MULTIVAR simulation has the weakest biases and is even able to reduce the high snow accumulation in the Weddell Sea present in the FREE simulation and to represent the thicker snow pattern measured in the SNOW-KaKu product downstream the Antarctica Peninsula. The biggest incoherence between the MULTIVAR simulation and the SNOW-KaKu observations is on the Pacific Ocean/Eastern Antarctic coastal sector, where the assimilated experiment does not reproduce the high snow thicknesses. The UNIVAR and MULTIVAR simulations have respectively 1.02 and 0.85 million km$^3$ less snow than SNOW-KaKu estimations in October 2017. The two simulations underestimate the SNOW-KaKu snow volume estimate for all the winter months of 2017 and 2018.

In both hemispheres, the MULTIVAR experiment consistently simulates snow depths closest to those used in the multivariate assimilation scheme. While a localized assimilation scheme is expected to modify the spatial distribution of the variable to match the observations, it is noteworthy that the assimilation of SNT leads to rapid corrections, with most spatial biases already reduced within the first month (Figure S3). The agreement between the MULTIVAR experiment's snow thickness and the observations is higher in the Arctic than in the Antarctic.

### 3.3 Radar freeboard volume

FREE and UNIVAR have biases of respectively -6.6 cm and -7.9 cm in RFBV compared to the LEGOS observations in April 2017 (Figure 3(c)). The MULTIVAR simulation logically exhibits a very small bias of -0.5 cm in the assimilated region and a RMSE of 2.2 cm, below the observation error of both the SAR and SARin data. The largest differences compared to the LEGOS RFB estimates are located along the coasts around the Canadian Archipelago and to the east of Greenland, i.e. in SARin areas. The SARin data are provided with higher observation errors compared to SAR data, with mean values of 19.2 cm and 9.2 cm, respectively. The highest difference (> 40 cm) between MULTIVAR RFB values and LEGOS RFB estimates arises at the end of both 2017 and 2018 winters in the north of Greenland, an area where snow observations are not available. In summer, when no RFB observations are assimilated, the probability density function of the MULTIVAR RFB values remains more positively skewed than in other simulations. In November, when the observed data

return after the summer break, the MULTIVAR experiment shows the lowest RMSE (2.6 cm) compared to the FREE (7.6 cm) and UNIVAR (8.3 cm) experiment based on the 2017 and 2018 averages. However, the MULTIVAR simulation presents larger RFB biases in November, than during the rest of the winter months when the errors relative to the RFB LEGOS dataset stay consistent.

LEGOS RFB measurements in the Antarctic present a similar gamma-type distribution as in the Arctic, with a decreasing SAR mode (from 3.3 cm to 0.5 cm) and increasing SARin mode (from 3.9 cm to 4.9 cm) between May and October 2017 (Figure 4(a)). The simulations exhibit more uniform RFB values than in the Arctic with up to 20% of the RFB having the same value in the UNIVAR experiment in May 2017. The FREE, UNIVAR and MULTIVAR experiments have similar RFB SAR modes of respectively -0.4 cm, -1.0 cm and 0.3 cm in May 2017, lower than the observed SAR mode of

3.3 cm. As the season progresses, the FREE and UNIVAR simulations present an even more negative bias, with RFB modes respectively 8 cm and 3 cm lower than the LEGOS RFB SAR mode in October 2017. A similar behaviour is shown for RFB SARin model equivalents, with the FREE and to a lesser extent the UNIVAR simulations frequently modelling negative RFB values that decrease as winter progresses. The MULTIVAR experiment is the only experiment to show a positively skewed distribution with positive modes in both SAR and SARin model equivalents throughout the duration of the

simulation, aligning more closely to the LEGOS observations variability for the positive RFB values. The FREE and UNIVAR simulations display a general low bias in RFB all around the Antarctic (respectively -13.1 cm and -9.6 cm in average), with the most significant negative biases located in the two thicker RFB areas, indicating a more uniform RFB spatial distribution (Figure 4(c)). The MULTIVAR experiment has the lowest biases, -3.5 cm in average, and a RMSE of 4.47 cm. The FREE, UNIVAR and MULTIVAR simulations represent respectively 1.80, 1.33 and 0.48 million km$^3$ less

RFBV than the LEGOS dataset. The underestimation of the southern RFB in the FREE simulation is likely due to the overestimation of the snow thickness in the Antarctic.

For all simulations and in both hemispheres, SAR measurements are in better agreement with the RFB model equivalent values compared to the SARin measurements. The MULTIVAR experiment shows the closest agreement with the observations among the simulations. The agreement between the RFB and SNV model equivalents from the MULTIVAR

experiment and the observations is not as high in the Antarctic as in the Arctic.

## 4 Validation with independent datasets

### 4.1 Total freeboard: ICESat-2 data

Both ICESat-2 (Ice, Cloud and Land Elevation Satellite) ATLAS and SARAL/AltiKa satellites measure total freeboard but the first one using a laser altimeter (Markus et al., 2017), and the second one with a radar altimeter. However,

the ICESat-2 product presents a smaller orbital hole (88° latitudinal limit) and a full-year availability, starting from the 14th of October 2018. The monthly ICESat-2 NSIDC ATL-20 gridded along-tracks product (Petty et al., 2023) is used on Figure 5, as a scatterplot between its total freeboard values and the total freeboard collocated in time and space for the LEGOS data

and the FREE, UNIVAR and MULTIVAR experiments in the Arctic. The LEGOS total freeboard is made using LEGOS RFB and SNOW-KaKu data, and the model constant water, ice and snow densities. The MULTIVAR simulation and LEGOS data present anticipated similar linear correlation statistics (slopes and r-values), MULTIVAR has then logically better statistics than the FREE and UNIVAR experiments, MULTIVAR simulation and the LEGOS data have similar mean RMSE compared to ICESat-2 data (6.7 cm and 7.2 cm respectively) and the MULTIVAR simulation and LEGOS data also display comparable mean total freeboard in January-February 2019, with values of 22.2 cm and 22.0 cm respectively, slightly thinner than the ICESat-2 estimate of 23.7 cm. The mean total freeboard for the FREE and UNIVAR experiments was found to be 19.4 cm and 15.0 cm, respectively, for the same period, due to thinner sea ice and snow cover in the UNIVAR experiment. The change in the total freeboard modelled by the MULTIVAR experiment is mainly due to a larger SIV, thanks to the assimilation update, compared to the UNIVAR experiment. The FREE and UNIVAR simulations consistently underestimate ICESat-2 total freeboard, especially in October 2018 with mean values of 9.2 cm and 6.6 cm respectively while the MULTIVAR experiment shows a mean value of 15.8 cm, aligning better with the mean total freeboard ICESat-2 estimate of 23.9 cm. In late summer, total freeboard has decreased during the melting season; however, the thinning is more pronounced in our simulations than in the ICESat-2 observations which does not seem to show a reduction in the mean freeboard compared with winter. The FREE experiment is not able to prevent excessive summer melting and exhibits unrealistic ice-free zones in October 2018. Higher statistical agreement in October 2018 for the MULTIVAR experiment shows that the data assimilation from the last winter positively impacts the simulation during the entire summer. However, compared to ICESat-2, MULTIVAR still underestimates the thickness of the total freeboard at the end of Arctic summer.

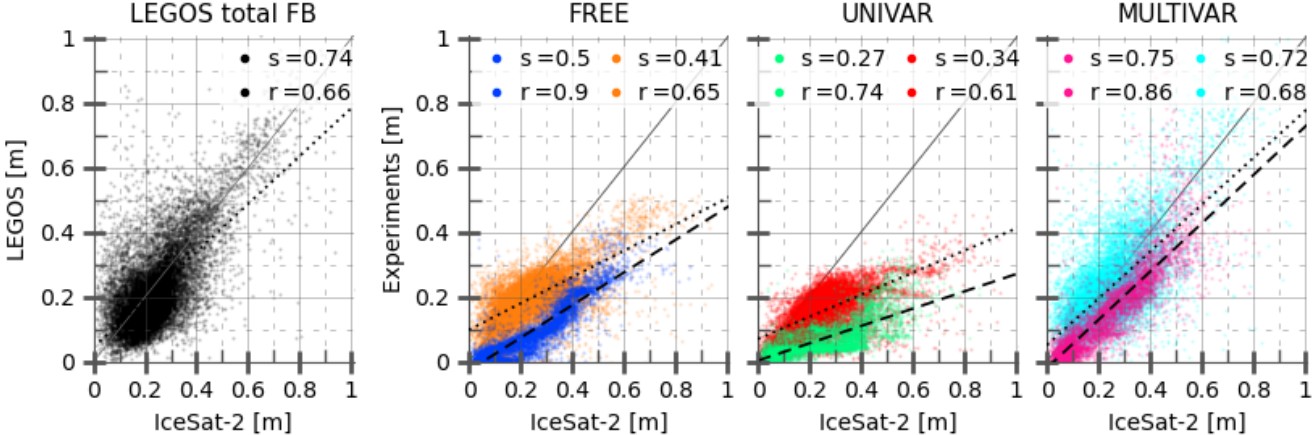

**Figure 5: Scatterplots of the monthly Arctic ICESat-2 total freeboard against FREE, UNIVAR, MULTIVAR experiments and LEGOS RFB/SND-KaKu data computed with model densities (black) for October 2018, beginning on the 14/10/2018 (experiments respectively in blue, green and pink; no LEGOS data), and for January-February 2019 (experiments respectively in orange, red and cyan). The x=y line (grey) and linear regressions for Oct 2018 (dashed black) and Jan-Feb 2019 (dotted black) are shown. Values of the linear slopes (s) and the r-values (r) are provided and all statistics are significant.**

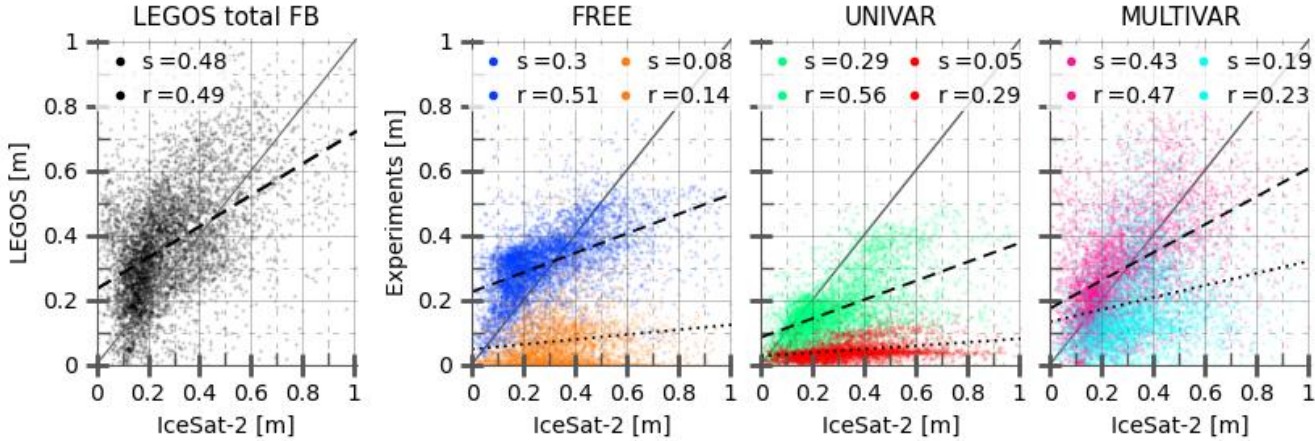

**Figure 6: Idem Figure 5 but for Antarctica.**

In Antarctica, simulated total freeboards show less agreement with ICESat-2 measurements compared to those in the Arctic (Figure 6). All the experiments and the LEGOS estimations present a general more scattered plot in the south than in the north. In October 2018, the last month of the assimilation season in the southern hemisphere, the MULTIVAR total freeboard shows a greater variability than the FREE and UNIVAR total freeboard, in accordance however with the dispersion of the assimilated CS2 LEGOS RFB and SNOW-KaKu datasets. Both the MULTIVAR experiment and LEGOS

data have a positive mean bias compared to the ICESat-2 data, of respectively +10.7 cm and +8.6 cm. The FREE simulation has a positive bias cluster for thin total freeboard but underestimates the thicker freeboard values, resulting in a mean bias of +2.4 cm. The UNIVAR experiment is underestimating ICESat-2 total freeboard values the most, with a mean bias of -11.9 cm. The melting season (January-February 2019) highlights the excessive thinning of the total freeboard in the simulations compared to the ICESat-2 data. The FREE experiment again has large unrealistic ice-free zones with total freeboard values

at 0 cm. The MULTIVAR experiment presents the highest total freeboard summer values among the experiments, with mean value of 19.6 cm (resp. 7.3 cm and 4.4 cm and for the FREE and UNIVAR experiments), still underestimating to a lesser extent the ICESat-2 mean values of 34.2 cm.

      Assimilating radar freeboard and snow depth observations in the multivariate framework significantly reduces biases found with ICESat-2 total freeboard in both hemispheres. The MULTIVAR shows systematic increase of the slopes in

winter as in summer. The agreement between modelled variables and ICESat-2 estimates is stronger in the north than in the south.

**4.2 Comparison with in-situ measurements**

      The in-situ data include Upward-Looking Sonar (ULS) moorings measurements in the Beaufort Sea, from the Beaufort Gyre Exploration Project (BGEP) with moorings A, B and D; and in the Fram Strait, from the Norwegian Polar

Institute (NPI) (Sumata et al., 2021) with moorings F11, F12, F13 and F14. We also use airborne laser and radar altimeter measurements in the western Arctic from the Operation Ice Bridge Quick Look product (OiB-QL, Kurtz et al., 2016).

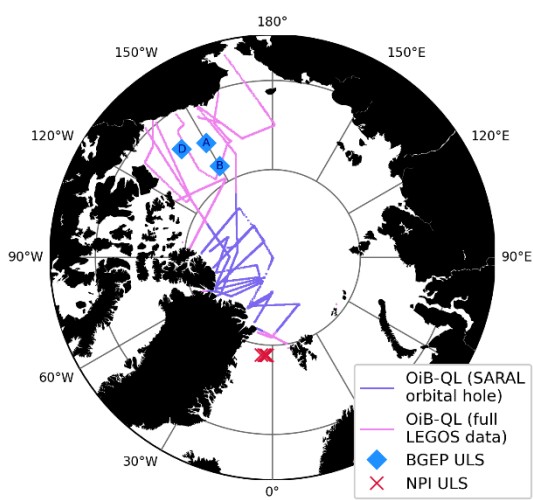

**Figure 7: Map of the Arctic and the different in-situ measurements used for validation of the simulations.**

The ULS moorings are located in regions where the LEGOS data are fully available (both RFB and SNOW-KaKu).
A distinction is made for OiB-QL measurements based on the availability of LEGOS data, highlighting the orbital hole that results from using SARAL-AltiKa measurements.

BGEP ULS measurements, available all year long, are available for the whole duration of the simulations, and the NPI ULS data are available until August 2018. Airborne OiB-QL observations are collected only in spring, but they sample a variety of ice (MYI and FYI) and cover a significant area in the Arctic. OiB-QL measurements campaigns took place during
7 days in March 2017, 3 days in April 2017, 1 day in March 2018 and 6 days in April 2018. The comparison for all measurements is made at monthly frequency. The LEGOS values presented in this section are made from the LEGOS RFB data, the SNOW-KaKu data, and the model fixed densities (LEGOS_mD).

#### 4.2.1 Beaufort Sea: BGEP ULS

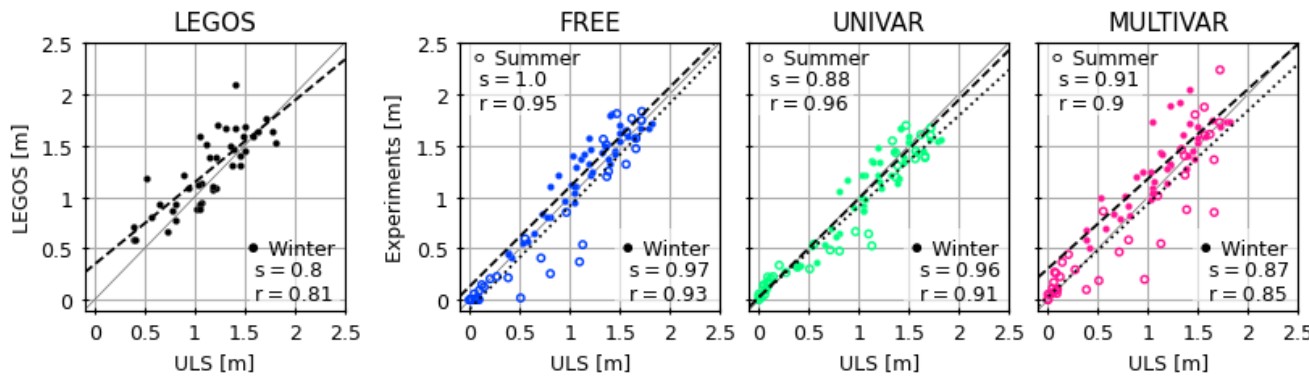

**Figure 8: Comparison of monthly average ice draft from LEGOS data, FREE, UNIVAR and MULTIVAR experiments within 200 km of the Beaufort Gyre Experiment Program ULS Moorings for the summer (empty circles) and winter (solid circles). The linear regression (dashed b lack line for winter, dotted black line for summer), slope (s) and r-value (r) are shown for each dataset. Methodology from (Laxon et al., 2013).**

The Figure 8 shows a remarkable agreement of ice drafts between BGEP data and all experiments. The LEGOS

observations have less coherence with the BGEP ULS measurements than the experiments but still with very high statistics.

The values that underestimate the BGEP measurements in all 3 experiments are mostly during summertime (Table 4). The

MULTIVAR experiment exhibits less accuracy than the FREE and UNIVAR simulations, with more scattered values and

higher RMSE (Table 4), inheriting the behaviour of assimilated LEGOS data. However, MULTIVAR ice drafts have higher

correlation than those from LEGOS estimates and, further, the MULTIVAR experiment is able to keep the strong correlation

obtained with the FREE ice draft values during summertime (Figure 8).

| BGEP ULS DATA | RMSE total | MD total | RMSE winter | MD winter | RMSE summer | MD summer |
|---|---|---|---|---|---|---|
| LEGOS | | | 0.194 | 0.113 | | |
| FREE | 0.134 | 0.011 | 0.121 | 0.095 | 0.150 | -0.087 |
| UNIVAR | 0.139 | -0.038 | 0.141 | -0.020 | 0.137 | -0.058 |
| MULTIVAR | 0.191 | 0.068 | 0.182 | 0.160 | 0.202 | -0.039 |

**Table 4: Root mean square error (RMSE) and mean differences (MD) between the BGEP ULS measurements and LEGOS data (only winter months: November to April), FREE, UNIVAR and MULTIVAR experiments, by season (summer: May to October and winter) and over the two seasons as a total.**

#### 4.2.2 Fram Strait: NPI ULS

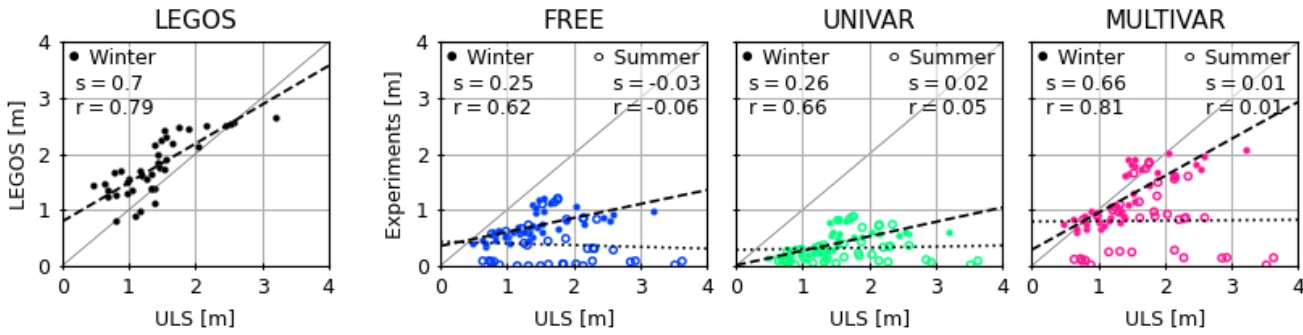

**Figure 9: Comparison of monthly average ice draft from LEGOS data, FREE, UNIVAR and MULTIVAR experiments within 200 km of the Norwegian Polar Institut (NPI) Fram Strait ULS Moorings for the summer (empty circles) and winter (solid circles). The linear regression (dashed black line for winter, dotted black line for summer), slope (s) and r-value (r) are given for each dataset.**

The ULS ice draft measurements are thicker in the Fram Strait than in the Beaufort Sea. The LEGOS data is in general agreement with the NPI data but presents mostly thicker ice drafts than the ULS measurements. The FREE and UNIVAR ice drafts consistently underestimate the ULS measurements, with very low slopes and r-values (Figure 9). These two experiments have most of the ice drafts at 0 m and show a deficit of up to 1.4 m compared with in-situ measurements (Table 5). Assimilating LEGOS RFB and SNOW-KaKu results in higher ice drafts, especially in winter when the assimilation is effective, and drastically reduces errors. Large errors in the MULTIVAR experiment's summer ice drafts values still remain in this region of the Fram Strait where the ice front is highly variable.

| NPI ULS DATA | RMSE total | MD total | RMSE winter | MD winter | RMSE summer | MD summer |
|---|---|---|---|---|---|---|
| LEGOS |  |  | 0.427 | 0.366 |  |  |
| FREE | 1.040 | -1.040 | 0.696 | -0.696 | 1.402 | -1.402 |
| UNIVAR | 1.238 | -1.238 | 1.029 | -1.029 | 1.458 | -1.458 |
| MULTIVAR | 0.645 | -0.571 | 0.316 | -0.189 | 0.991 | -0.972 |

**Table 5: Same as Table 4 with the NPI ULS measurements.**

### 4.2.3 Operation IceBridge QuickLook sea ice thickness

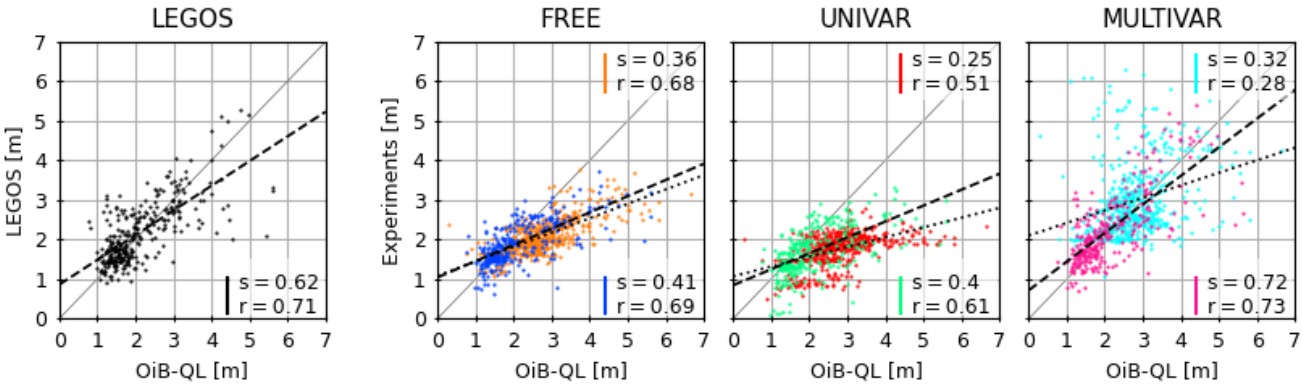

**Figure 10: Comparison of monthly average ice thickness from LEGOS data, FREE, UNIVAR and MULTIVAR experiments collocated with OiB-QL airborne measurements in the Arctic. Areas where LEGOS SNOW-KaKu and RFB measurements are available are respectively in black, blu e (FREE), green (UNIVAR) and pink (MULTIVAR) with linear regression in dashed black line; otherwise, orange (FREE), red (UNIVAR) and cyan (MULTIVAR) with linear regression in dotted black line refer to regions where SNOW-KaKu data are not available. All ice thickness values are gridded onto a 0.4° latitude by 4° longitude Arctic grid, following the methodology of (Tilling et al., 2018). The slope (s) and r-value (r) are given for each dataset.**

The LEGOS data and the OiB-QL ice thickness measurements are in general good agreement (Figure 10). The OiB-QL data presents a cluster of measurements between 1 and 2 m that is well reproduced by all experiments and by the LEGOS data. Thicker measurements from the OiB-QL 2017 and 2018 campaigns are underestimated by the FREE and UNIVAR experiments (Table 6). These two experiments do not show ice thickness values higher than 4 m, whereas the OiB-QL

measurements signal ice up to 6.6 m thick. The MULTIVAR simulation is able to reproduce thicker ice, resulting in a general reduction of errors, especially bias, with the OiB-SL measurements, in regions where all the assimilated data is available, and also where some or all of the assimilated data are missing (Table 6). However, the MULTIVAR experiment's ice thickness values are very scattered, especially in the region where the LEGOS data is not entirely available (no SNOW-KaKu poleward of 81.5°N; and no RFB LEGOS poleward of 88°N).

| OiB  AIRBORNE DATA | RMSE total | MD total | RMSE lat<81.5°N | MD lat<81.5°N | RMSE lat>81.5°N | MD lat>81.5°N |
|---|---|---|---|---|---|---|
| LEGOS | | | 0.449 | 0.068 | | |
| FREE | 0.639 | -0.503 | 0.459 | -0.200 | 0.744 | -0.681 |
| UNIVAR | 0.869 | -0.794 | 0.574 | -0.416 | 1.042 | -1.016 |
| MULTIVAR | 0.652 | 0.182 | 0.486 | 0.135 | 0.750 | 0.209 |

**Table 6: Same as Table 4 with the OiB Airborne data and, according to the areas where SNOW-KaKu data is present (<81,5°N) or not (> 81,5°N) and for all OiB Airborne data.**

## 4.3 Sea ice volume

### 4.3.1 Total sea ice volume

The daily total ice volume values for each experiment are shown on Figure 11 (dotted lines). Figure 11 also presents the experiments collocated within the spatial coverage of the assimilated observations, which excludes the central Arctic orbital gap and limited coverage of marginal seas (solid lines). This area, where both the RFB and KaKu data are available, is hereafter referred to as the "LEGOS zone" or the "LEGOS observations domain". Three different products are shown: (1) LEGOS_og, the original SIV LEGOS (Guerreiro et al., 2017), based on CS2 RFB and SNOW-KaKu measurements with varying snow and ice densities; (2) LEGOS_mD, which uses the same measurements but applies constant snow and ice densities from the SI3 model; and only in the Arctic (3) CS2SMOS AWI, which combines SIV estimates from CS2 altimetric freeboard measurements of thicker ice and SMOS brightness temperature measurements of thinner ice (Ricker et al., 2017), using a modified W99 snow climatology and variable ice and snow densities.

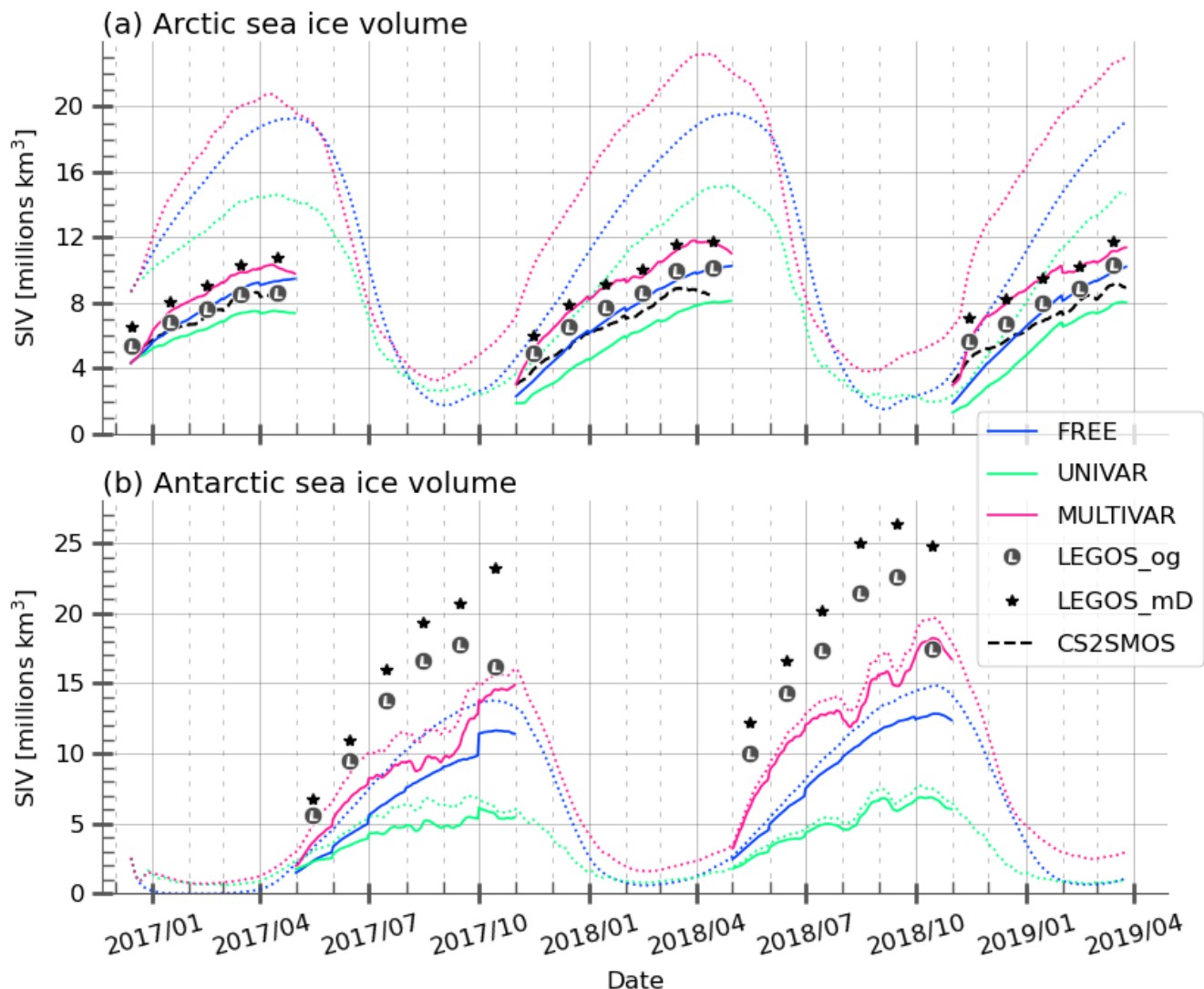

**Figure 11: Time evolution of Arctic (a) and Antarctic (b) sea ice volume. The daily values are presented for the simulations FREE (blue), UNIVAR (green) and MULTIVAR (pink), integrated over the whole hemisphere (dotted) and over the observation domain (plain lines). SIV observations used for comparison are computed over the LEGOS observation domain: LEGOS original SIT (LEGOS_og, grey L in circles), SIT constructed from LEGOS observations of RFB and snow and the model constant ice and snow densities (LEGOS_mD, black stars), and CS2SMOS AWI data in the Arctic (black dashes). The SIV is computed using either SIC data provided by the supplier or the SIC OSISAF SSMIS data.**

In the Arctic, the amount of sea ice remains consistently high throughout the entire simulation in the MULTIVAR experiment, resulting in sea ice maximums on average 13% and 48% higher than respectively the FREE and UNIVAR experiments (Figure 11(a)). The FREE and UNIVAR simulations start each winter with a low sea ice volume compared to the observations. The MULTIVAR experiment presents systematically higher volume estimates and aligns better with CS2SMOS product in the beginning of November 2017 and 2018. The MULTIVAR SIV values increase rapidly during the

first month of assimilation and follow closely the LEGOS_mD observations. Even in summer, the MULTIVAR simulation maintains more ice volume in the Arctic than the other simulations. The UNIVAR simulation shows a particularly drastic decrease in its ice volume estimate relative to the FREE experiment and is consistently lower than all the observation products. On average over the entire simulation period, the UNIVAR experiment shows a decrease in sea ice volume of 23% while the MULTIVAR experiment shows a 21% increase compared to the FREE experiment. The assimilation of CS2 LEGOS RFB and SNOW-KaKu in the MULTIVAR experiment modifies the seasonal cycle of the sea ice volume estimates, with a maximum earlier than in the other simulations, and is more consistent with the observations.

As in the Arctic, MULTIVAR has the highest freezing rate and the highest total sea ice volume in Antarctica among the experiments for the most part of the simulation periods (Figure 11(b)), with, on average, 25% and 141% higher ice volume than FREE and UNIVAR estimates respectively. UNIVAR consistently presents the lowest ice volume. The assimilated experiments have irregular time series during the second half of the growing season, the MULTIVAR simulation especially collapses many times before reaching its peak. These collapses coincide between the two assimilated experiments and are also present in the observation space (solid lines, Figure 11(b)). These sudden ice volume losses are due to the occurrence of large open waters or polynyas within the sea ice cover which first and foremost causes an increase of sea ice leads from July to September 2017 and in August and September 2018 (Figure 1(b)). Some of them also appear in the observation products such as the well-known Maud-rise polynya in the Weddell Sea in 2017.

The use of the model constant densities (LEGOS_mD) results in higher SIV estimates than the LEGOS_og product using seasonally varying ice and snow densities to convert RFB into ice thickness (Figure 11(b)). The deviation between these two datasets is maximum in October because of the significant drop in ice density from 900 kg/m$^3$ to 875 kg/m$^3$ between September and October. With one exception (October 2018), both LEGOS_og and LEGOS_mD observations present systemically higher SIV values than MULTIVAR simulation. And even if the MULTIVAR experiment remains the closest experiment to the LEGOS observations, it is still up to 10 million km$^3$ below the LEGOS_mD estimates. Over both 2017 and 2018 winters, the datasets present mean SIV of respectively 4.6, 8.0, 10.8, 15.2 and 18.5 million km$^3$ for the UNIVAR, FREE and MULTIVAR simulations, and the LEGOS_og and LEGOS_mD products. The LEGOS_og product displays a sea ice maximum in September, a month earlier than the FREE simulation. LEGOS_mD also has a SIV maximum in September for 2018 winter only, but the differences in densities make it unclear to identify the exact peak period in 2017. Similarly, the occurrence of polynyas in assimilated experiments makes it impossible to accurately determine the maximum period.

In both hemispheres, the MULTIVAR experiment shows the largest sea ice volume, while UNIVAR has the smallest. Among the different products, LEGOS_mD has the highest volume, followed by LEGOS_og and ─only in the Arctic─ CS2SMOS. Notably, the products are highly sensitive to variations in snow and ice densities, with LEGOS_mD showing in average respectively 1.48 million km³, resp. 5.6 million km³, more sea ice volume than the original LEGOS_og in the Arctic, resp. the Antarctic.

### 4.3.2 Comparison with SMOS satellite measurements

The CS2SMOS AWI product uses measurements from the SMOS satellite in addition to CS2 measurements. SMOS is known to have less uncertainties than CS2 on thin ice measurements (less than 1 m, Ricker et al., 2017). Based on CS2 measurements, the LEGOS_og logically displays a consistent sea ice thickness spatial distribution compared to the CS2SMOS product with the smallest RMSD (resp. mean difference) of 30 cm (resp. 5 cm, Figure 12). LEGOS_mD presents a higher RMSD (resp. mean difference) of 39 cm (resp. 34 cm). The FREE simulation shows thinner ice than the CS2SMOS data in the central Arctic and on the east coast of Greenland, and thicker ice elsewhere. The UNIVAR simulation has a globally much thinner ice coverage with approximately half of its ice area covered by ice below 1 m thickness and the other half with ice between 1 m and 2 m height. The MULTIVAR experiment shows a higher ice volume compared to the other experiments, with a significant ice accumulation thicker than in the CS2SMOS product on the north of the Canadian Archipelago and Greenland. In that area of important deviation between CS2SMOS and MULTIVAR values, the assimilated SNOW-KaKu measurements are not available. In the LEGOS SIV observation domain, the simulations present a similar RMSD against the CS2SMOS product of 33 cm (FREE, MULTIVAR) and 31 cm (UNIVAR). The MULTIVAR modelled ice thickness has the same positive biases as the LEGOS_mD product but keeps a thinner ice than the CS2SMOS data on the east coast of Greenland, similarly to the two other simulations. Outside of the LEGOS observations domain, the UNIVAR simulation shows the highest RMSD (65 cm) for the CS2SMOS SIT values thicker than 1 m, while the FREE simulation has the highest RMSD (48 cm) for CS2SMOS SIT values thinner than 1 m among the three experiments. The RFB and snow assimilation in the MULTIVAR simulation corrects the FREE and UNIVAR underestimation of the ice thickness in the central Arctic region (RMSD of 38 cm) and presents lower positive biases than the FREE simulation for the thin ice around the ice edge (RMSD of 27 cm).

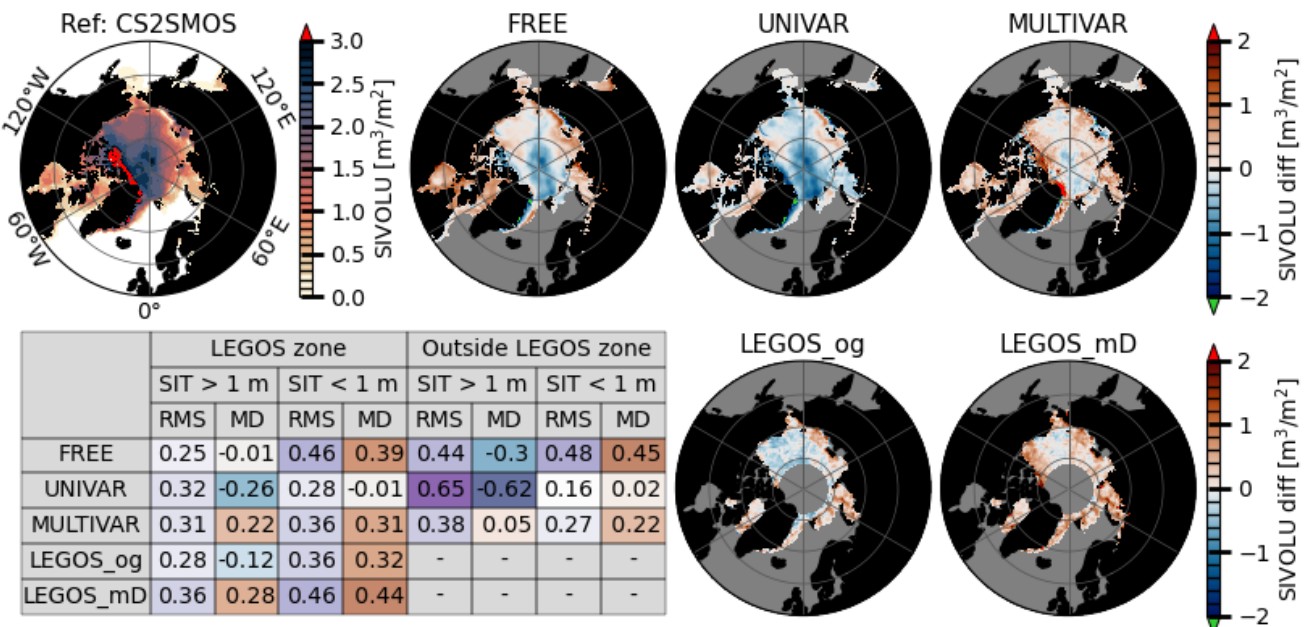

**Figure 12: April 2017 sea ice volume in the Arctic for CS2SMOS dataset (reference) and its difference with the FREE, UNIVAR, and MULTIVAR experiments (first line) and the observations LEGOS_og (original) and LEGOS_mD (with model constant densities). Table: root mean square error (RMS) and mean difference (MD) between FREE, UNIVAR, MULTIVAR, LEGOS_og, LEGOS_mD and CS2SMOS data, calculated on the LEGOS zone and outside the LEGOS zone and for CS2SMOS sea ice thickness of less than or greater than 1m. The table colours highlight the values close to 0 (white) and the extremes (green for the RMS, and blue/red for the negative/positive MD). The LEGOS zone corresponds to areas where the KaKu snow depth is available.**

In Antarctica, the SMOS product (Tian-Kunze and Kaleschke, 2021) detects ice thinner than 1m using brightness temperature measurements, hence the data is completely independent from the LEGOS altimetric data assimilated in the MULTIVAR experiment. The LEGOS observations, considering both fixed and varying densities, present a very thick ice volume in the southern hemisphere (Figure 13). Similarly to the Arctic, the LEGOS_mD shows thicker ice volumes than the LEGOS_og data. Compared to SMOS data, both LEGOS estimates show a different ice field: the CS2 Antarctic ice thickness processed by the LEGOS is thicker with RMSE values of 78 cm (resp. 97 cm) for LEGOS_og (resp. LEGOS_mD) and the ice accumulations are measured on the northernmost part of the Weddell Sea with CS2 measurements, whereas SMOS satellite detects thick ice on the southernmost part of the Weddell Sea. The FREE and UNIVAR simulations have spatially homogeneous SIV distributions and similar RMSD compared to the SMOS data on the LEGOS domain (respectively 24 and 26 cm). The FREE experiment has a consistent positive SIV bias compared to the SMOS dataset. Although most of the UNIVAR experiment's ice thickness is below 1 m, it underestimates SMOS ice thickness, except on areas close to the ice edge, where UNIVAR values align well with the SMOS measurements (mean difference of -2 cm). Compared to FREE and UNIVAR, the MULTIVAR simulation shows more important ice accumulations, in consistency with both LEGOS SIV data, and therefore has the highest RMSE relative to the SMOS data on the LEGOS domain (38 cm).

The MULTIVAR simulation does not reproduce the largest LEGOS SIV values and is therefore closer to the SMOS data than the LEGOS estimates. Outside the LEGOS domain, MULTIVAR corrects the positive bias noticed along the ice edge in the FREE simulation but degrades the performances of the UNIVAR simulation with a higher error (mean difference of 7 cm). The FREE simulation is the only experiment that does not reproduce correctly the Maud Rise polynya, which is seen in all observation products and in the two assimilated experiments.

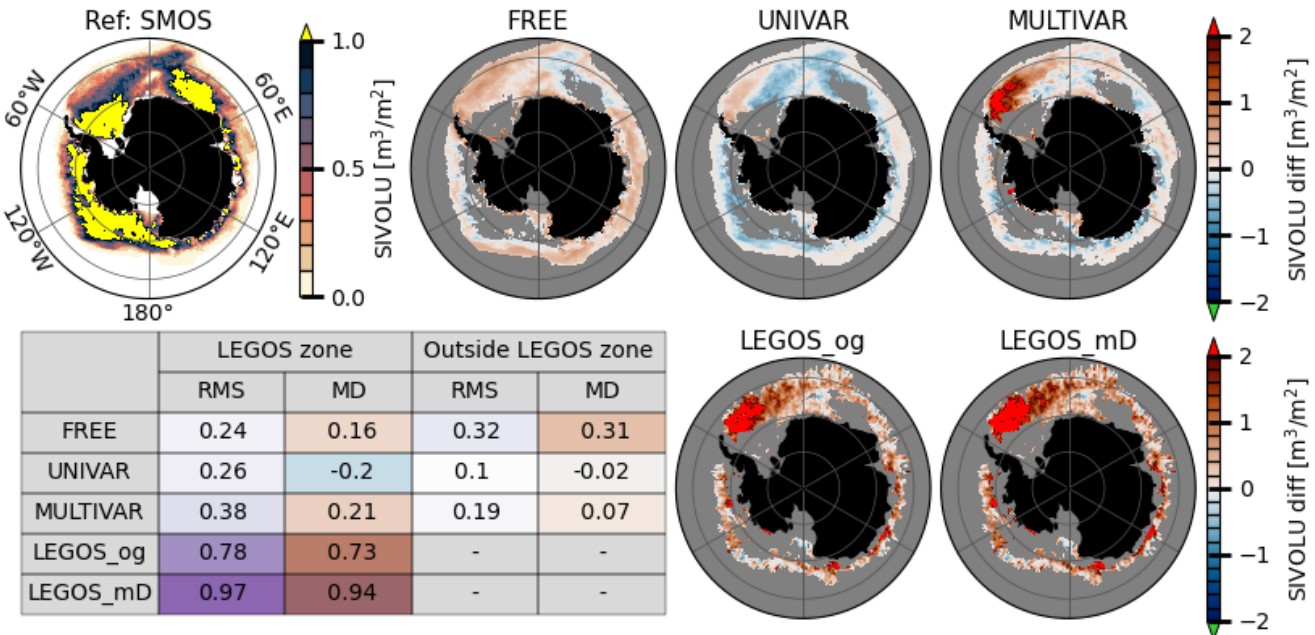

| | LEGOS zone | | Outside LEGOS zone | |
|---|---|---|---|---|
| | RMS | MD | RMS | MD |
| FREE | 0.24 | 0.16 | 0.32 | 0.31 |
| UNIVAR | 0.26 | -0.2 | 0.1 | -0.02 |
| MULTIVAR | 0.38 | 0.21 | 0.19 | 0.07 |
| LEGOS_og | 0.78 | 0.73 | - | - |
| LEGOS_mD | 0.97 | 0.94 | - | - |

**Figure 13: September 2017 sea ice volume in the Antarctic for the SMOS data (reference) and its difference to the FREE, UNIVAR, and MULTIVAR experiments (first line) and to the observations LEGOS_og (original) and LEGOS_mD (with model constant densities). The colorbar shows only which only measures the ice that is thinner than 1 m (thicker ice is represented in yellow). Table: root mean square error (RMS) and mean difference (MD) between FREE, UNIVAR, MULTIVAR, LEGOS_og, LEGOS_mD and SMOS data, calculated on the LEGOS zone and outside the LEGOS zone. The table colours highlight the values close to 0 (white) and the extremes (green for the RMS, and blue/red for the negative/positive MD). The LEGOS zone corresponds to areas where the KaKu snow depth is available.**

In both hemispheres, for SIT < 1 m, using the multivariate assimilation scheme better aligns the modelled sea ice volume with the SMOS data, presenting a lower RMSE for the MULTIVAR experiment than the FREE experiment and the LEGOS_mD data. However, the UNIVAR experiment shows more accurate sea ice volume estimates for thin ice than the MULTIVAR experiment when using SMOS measurements as a reference.

# 5 Discussion

## 5.1 Performances of the multivariate assimilation

The radar freeboard and snow thickness assimilation allows the multivariate assimilation experiment to correct the model biases against the assimilated datasets: the MULTIVAR simulation has the closest results to the RFB LEGOS and SNOW-KaKu products in both hemispheres. However, the comparison of the Antarctic snow and RFB equivalents shows less agreement with the assimilated observations than in the Arctic.

The univariate assimilation system only corrects the SIC variable and keeps a constant SNV. In the Antarctic, and to a lesser extent in the Arctic, the UNIVAR experiment displays a lower SNV compared to the FREE experiment. Thanks to the snow assimilation, in the MULTIVAR simulation, the total volume of snow is adjusted but does not recover the total amount of observed snow in the Antarctic. The SNOW-KaKu assimilation enables the simulations to reproduce the snow observations spatial distribution in both hemispheres. The snow cover completely melts in summer in both hemispheres, and while the timing of melt should influence the sea ice evolution, our results do not indicate a persistent or clearly attributable long-term impact of the winter snow assimilation.

The MULTIVAR simulation shows higher RFB values than the FREE and UNIVAR simulations in both hemispheres, even in the absence of observations during the summer. However, a drift in the RFB equivalent is still observed during this season, leading to a negative bias in November/May, when the assimilation begins. This small negative bias suggests that the model's trajectory is below the observed values, a hypothesis supported by the significantly more pronounced bias observed in the FREE and UNIVAR simulations. In the Antarctic, the RFB is significantly underestimated in the FREE and UNIVAR simulations, reflecting an imbalance between snow and ice thicknesses: the snow cover is too thick and the sea ice too thin, resulting in radar freeboard values that are more negative than observed. The initial state of ice and snow in the southern hemisphere found in the FREE experiment is much more different from the assimilated observations compared to the north. The multivariate assimilation process is then less effective in aligning the model with the observed data in the Antarctic than it is in the Arctic.

In both hemispheres, the MULTIVAR simulation produces RFB extremums that extend beyond the minimums and maximums observed in the FREE and UNIVAR simulations, and more closely align with the LEGOS observations. Despite this improvement, the MULTIVAR simulation does not capture the thickest and thinnest RFB LEGOS measurements. This discrepancy could be attributed to the spatial resolution mismatch between the observations and the model. Furthermore, it is important to recognize that the MULTIVAR simulation is not designed to replicate every extreme observation (such as a notably high SARin RFB of 4.3 m observed in October 2017 in Antarctica) as the assimilation scheme seeks to balance observational data with the model's physical constraints. Given the use of unfiltered RFB data in the assimilation, we do not expect the model to reproduce the exact observed values but rather a smoothed representation that respects the model's inherent dynamics.

The LEGOS observations are characterized by spatially significant data gaps in the central Arctic and in the Canadian Archipelago. The MULTIVAR simulation smoothly assimilates the RFB and SNOW-KaKu data in these areas without any visible demarcations. Furthermore, due to the choice of parameters for the localisation algorithm in the assimilation scheme, the assimilated satellite tracks do not print on the modelled patterns. However, the largest RFB differences between the MULTIVAR experiment and the RFB LEGOS assimilated observations are located on the north of the Canadian Archipelago and Greenland, with an especially thin RFB in our simulation locally north of Greenland. No snow observations are available in this area, and the MULTIVAR presents thicker snow values than the FREE and UNIVAR simulations. No particular RFB bias is present in the large snow KaKu observation gap around the North pole, suggesting that in the absence of snow observations, an inaccurate modelled snow depth does not affect the RFB assimilation performance on a large scale, but can result in higher RFB biases very locally. When considering the sea ice volume, the experiments provide similar results in both hemispheres: the assimilation of SIC with the univariate method decreases the ice volume compared to the FREE simulation. The assimilation of RFB LEGOS and SNOW-KaKu creates the highest sea ice volume of all the simulations. The MULTIVAR experiment also displays a more accurate spatial distribution of the ice than the other experiments. The MULTIVAR modelled ice volume in the Arctic is very coherent with the LEGOS_mD dataset in the Arctic, which is more consistent with our observation operator in terms of sea water, snow and sea ice densities. In the Antarctic, the modelled sea ice volume is consistently lower than the LEGOS_mD product, probably due to lower model skills in representing sea ice in the Antarctic than in the Arctic (Massonnet et al., 2011) and more divergence between the modelled initial state and the assimilated observations, as discussed earlier.

## 5.2 Comparison with independent data

The ICESat-2 satellite measures the total freeboard through laser altimetry instruments, it is therefore completely independent from the radar altimetry-based LEGOS freeboard estimates. Previous section shows that assimilating LEGOS data reduces the errors in the simulations total freeboard estimates compared to ICESat-2 measurements. The comparison in the Antarctic also shows weaker correlations between ICESat-2 data and the experiments than in the Arctic. It should be emphasized that most of the comparisons made in the southern hemisphere with ICESat-2 data is done during summertime, without assimilation of radar freeboard and snow. The summer period of the southern ice is also known to be poorly represented by the models (e.g. Roach et al., 2020; Shu et al., 2020). In addition, the LEGOS data present less coherence with ICESat-2 compared to the Arctic. Nevertheless, the MULTIVAR simulation exhibits higher performance in terms of total freeboard compared to the other two simulations, particularly during the summer months. This demonstrates that the multivariate assimilation process induces changes in total freeboard that persist even when radar freeboard and snow are not assimilated.

Further comparison with in-situ independent observations in the Arctic only show general improvement with the multivariate assimilation system compared to the FREE and the UNIVAR experiments. The MULTIVAR experiment is able to maintain the remarkable agreement found with the FREE experiment with ULS moorings in the Beaufort Sea and

favourably thickens all types of ice in the Fram Strait region. At the same time, the multivariate approach also positively increases the thickest ice even in the absence of snow data. Comparisons during the summer season show no particular deterioration or improvement with the multivariate system.

Sea ice thickness products obtained from brightness temperature measured by the SMOS satellite can be considered complementary to the altimetric ice products because they provide thin ice estimates (Kaleschke et al., 2024). In the Arctic, the CS2SMOS data shows thinner ice thicknesses than the LEGOS products (same as other CS2 products in Sallila et al. (2019) but the observational datasets are still coherent (better spatial alignment and RMSD of the same order as the FREE simulation). In that hemisphere, differences between the simulations and the CS2SMOS data show a generally better agreement for the MULTIVAR simulation compared to the FREE and UNIVAR simulations. The predominant positive biases observed in the MULTIVAR simulation are consistent with the biases in the LEGOS_mD product (i.e., north of the Canadian Archipelago and Greenland). However, Sallila et al. (2019) established that the CS2SMOS product tends to underestimate the thickness of thick ice in the Arctic when compared to in-situ measurements. Therefore, an overestimation of the CS2SMOS estimates is not an unexpected outcome for thicker ice. The CS2SMOS product estimates of thin ice, however, are in closer alignment with the in-situ Arctic measurements (Sallila et al., 2019). The more precise thin ice estimates from the UNIVAR experiment are compromised by the assimilation of CS2 data in the MULTIVAR experiment, when compared to the CS2SMOS values. It may be beneficial to increase the observation errors for the thicker RFB or in the marginal ice zone in order to reduce this degradation in comparison to the UNIVAR simulation.

In Antarctica, the SMOS product is restricted to ice with SIT < 1 m, and a similar situation as with the thin Arctic ice arises: the comparison with the SMOS Antarctic data shows a better agreement with the UNIVAR simulation. The MULTIVAR simulation predominantly overestimates the SMOS measurements, due to an overestimation of the assimilated LEGOS data compared to the SMOS estimates. The SMOS data however display a systematic underestimation of sea ice thickness in areas of ice divergence (Kaleschke et al., 2024); and the Antarctic sea ice shows generally divergent ice drifts (e.g. Petty et al., 2021). Moreover, the assimilated LEGOS data present little resemblance with the SMOS Antarctic measurements. However, the Southern Ocean lacks consistent in-situ data measurements of sea ice and snow to better evaluate satellite observations and models estimates. While the assimilation improves the agreement between assimilated products, the contrasting patterns seen in LEGOS and SMOS sea ice thickness highlight the current observational uncertainty in Antarctica, making it difficult to assert which product more accurately represents the true state of the sea ice. In the future, the system could also assimilate both CryoSat-2 (for thick ice) and SMOS (for thin ice) products in both hemispheres, provided that Antarctic sea ice thickness estimates have greater consistency and agreement. Here, The MULTIVAR simulation provides better statistics than the two other experiments against the ICESat-2 data thanks to the multivariate assimilation of LEGOS observation product, and it shows a better alignment with the SMOS data than the FREE simulation despite the assimilation of a LEGOS product that does not align with the SMOS data. The validation against these two independent datasets hence proves that the multivariate ice assimilation scheme in the Antarctic created an intermediate sea ice state between the LEGOS observations and the model.

## 5.3 Ice and snow densities

Sea ice thickness products obtained from CS2 radar altimetry measurements have significant uncertainties due to the assumptions made on values of snow thickness and ice and snow densities during the radar freeboard to ice thickness conversion (Garnier et al., 2021; Kern et al., 2015; Kwok and Cunningham, 2015; Mallett et al., 2020). Assimilating directly the radar freeboard allows us to control the origin of the uncertainties by using the rawest measurement possible and controlling all the assumptions made during the assimilation process. We decided to assimilate a satellite observed altimetry snow thickness, which uses the same radar altimetry techniques as the RFB product. Garnier et al. (2022) show that using coherent measurement techniques between the snow and freeboard datasets gives an accurate total freeboard value even when the snow-ice interface is biased.

The multivariate data assimilation proceeds for the RFB volume observations by constructing a model equivalent using the model SIV and SNV variables and the model fixed densities for water, ice and snow. The water density is nearly consistent in all the sea ice volume datasets, with values varying by only a few kg/m$^3$. However, the ice and snow density values vary a lot. The model's constant ice density is 917 kg/m$^3$, but the ice density in the Arctic depends on the ice age for LEGOS_og and CS2SMOS with the values from Alexandrov et al. (2010) as extremums: 882 kg/m$^3$ for the MYI and 917 kg/m$^3$ for the FYI. Hence, assimilating radar freeboard and snow with the model constant ice density primarily affects regions dominated by MYI in the Arctic, which corresponds to the thicker ice regions that do not melt during summer, in the north of the Canadian Archipelago and Greenland. The difference of ice density results in an ice thickness 32% higher on MYI in the Arctic. The model constant snow density is 330 kg/m$^3$. Garnier et al. (2022) used a constant snow density of 300 kg/m$^3$ in the Arctic for the LEGOS_og product, with a consequently lower sea ice thickness than the model for equal RFB, snow thickness and ice density values. Densities in the observation products in the Antarctic are generally seasonally varying densities. The model's ice density (constant 917 kg/m$^3$) exceeds that of the LEGOS_og observations (895 kg/m$^3$ on average), with a particularly significant difference in October (LEGOS value: 875 kg/m$^3$). The model snow density is comparable on average to the LEGOS observation's snow densities in Antarctica but presents differences up to 40 kg/m$^3$ for some winter months. This discrepancy between ice and snow densities brings additional variability in sea ice volume even when similar radar freeboard and snow measurements are used, as illustrated by the difference between the LEGOS_og and LEGOS_mD datasets. The constant densities parametrization in the model enhances the positive bias of the sea ice volume in the Arctic compared to the CS2SMOS product. In the experiments presented here, the uncertainties due to the densities are related to the RFB observation operator. Hence, these uncertainties increase the representation error in the analysis. Varying ice and/or snow densities are crucial features to be incorporated in the next version of the sea ice model: it would ensure a more accurate radar freeboard assimilation by lowering this representation error. One could for instance use the method from (Zhang et al., 2022) to select the optimal freeboard-to-thickness conversion ratios values by fitting the resulting ice thickness to in-situ or airborne measurements. Moreover, implementing seasonally evolving densities in the model could improve the realism of key physical processes such as snow–ice formation particularly in the Antarctic. For instance, Mallett et al. (2020)

offers a linear evolution of the snow density to account for the densification of the snow as winter passes. Sievers et al. (2023) use this relationship to implement a radar freeboard assimilation scheme with a varying snow density, but did not modify the density in the model physics.

## 5.4 Sea ice openings in Antarctica

In both hemispheres, results showed that all assimilated experiments successfully corrected the biases of the FREE experiment with respect to the SIC variable. Univariate SIC assimilation provides the best performance for sea ice concentration as the covariances are not negatively affected by other quantities. The degradation of modelled SIC in summer in the MULTIVAR configuration, while UNIVAR uses the same SIC observations, suggests that the multivariate assimilation may introduce erroneous corrections through model covariances between SIC, SIV, and SNV. These propagated increments, applied in the absence of direct summer observations of SIV or SNV, appear to deteriorate SIC consistency, underscoring the need to reassess or seasonally adapt the covariances used in the assimilation. Still, summer remains the most difficult season for systems to reproduce in both hemispheres. SIC passive microwave observations also have the greatest uncertainties during the melting season (Ivanova et al., 2014).

Sea-ice models using Viscous-Plastic or Elastic-Viscous-Plastic rheologies have been shown to reproduce the observed sea ice deformations only with high resolution horizontal grids (4.5 km grid spacing or lower, Hutter et al., 2018; Spreen et al., 2017; Wang et al., 2016). Both assimilated experiments increased the amount of open water compared to the FREE experiment and increase the amount of sea ice leads on a coarser grid of ¼°, i.e. grid cells of size between 10 km and 24 km in the Arctic. The multivariate experiment shows an even higher presence of open waters than the UNIVAR experiment during the peak period in the boreal summer. These features are not supported by the assimilated SIC SSMIS observations and are likely artificial, though some may be related to the assimilation of along-tracks RFB data, which is capable of detecting finer-scale polynyas that are not visible in the coarser SIC SSMIS product.

The assimilated experiments timeseries in the Antarctic display oscillations that are due to the occurrence of very localized low-SIC or open water areas, e.g. the so-called polynyas (Figure 1(b)). These openings only appear in the assimilated experiments. As none of these openings occur in the FREE experiment, the thick snow and ice layer likely insulates the ocean from the atmosphere, maintaining the temperature inversion beneath the ice and limiting oceanic heat flux toward the ice base. The occurrence of the Maud Rise Polynya in Sept-Oct 2017 (Jena et al., 2019) is reproduced by the UNIVAR experiment, but its size is underestimated (Fig 1b)). On the other hand, the size of this polynya is greatly overestimated by MULTIVAR and appears about 3 months in advance of the one observed by satellite. Furthermore, the MULTIVAR (and UNIVAR to a lesser extent) experiments show the presence of other polynyas this winter 2017 and a few more during winter 2018. These events are the combination of a general reduction of snow and increase of ice freeboard with respect to the FREE simulation, but in specific areas where SIC or RFB observations show local minima. These reductions in the areas covered by ice finally expose the surface to the warm waters of the ocean. Once triggered, assimilation is no longer able to counteract the strong vertical instability and oceanic warming that prevent these openings from closing.

However, some of these activation zones correspond to fracture zones that have already been identified, either for reasons of atmospheric divergence (low pressure systems in (Kwok et al., 2017) or linked to the local bathymetry (Reiser et al., 2019). These polynyas are the consequences of intense interactions between the ocean and the surface in our simulations in places where the equilibrium of the model is very sensitive to any disturbance. The modifications in the assimilation scheme of the SST and in-situ profiles described in section 2.1.2 have reduced the likelihood of triggering polynyas in both UNIVAR and MULTIVAR simulations, but have not been able to prevent their occurrence.

## 6 Conclusion

This study presents the first implementation of a multivariate sea ice assimilation scheme in both the Arctic and Antarctica within a global ¼° modelling and analysis system. This system, largely based on the Mercator operational system, already includes a multivariate ocean assimilation but currently only assimilates sea ice concentration (SIC). Our study enhances this capability by incorporating a multivariate ice assimilation approach, assimilating along tracks radar freeboard and snow depth jointly with sea ice concentration. By comparing simulations without assimilation, with univariate SIC assimilation, and with this innovative multivariate system, we assess the capabilities of the assimilation scheme. The univariate SIC assimilation method systematically decreases the ice volume compared to the FREE experiment and shows a thin ice bias compared to observations. The multivariate assimilation increases the sea ice volume in both hemispheres, enabling the modelled sea ice to converge on assimilated data sets. The spatial distribution of the sea ice and the snow is modified in accordance with the assimilated observations. Even in summer and in the observation's spatial holes, when no satellite altimetry observations are assimilated, the MULTIVAR experiment's ice variables are favourably modified by the multivariate ice assimilation. Moreover, the diagnosed freeboard from the multivariate system compares better with Iceat-2 independent observations in the Arctic and, to a lesser extent, in Antarctica. Despite the heterogeneous nature and varying resolutions of the assimilated data sets, the multidata/multivariate assimilation system demonstrates robust behaviour even in the absence of certain observations (summer, spatial hole), indicating a consistent and physically coherent adjustment of the sea ice state.

The comparison with observations coming from SMOS satellite shows that the UNIVAR experiment agrees better with the more reliable SMOS sea ice volume estimates for thin ice (less than 1 m) than the MULTIVAR experiment. In the Antarctic, CS2 and SMOS sea ice volume estimates diverge, so assimilating CS2 radar freeboard takes the model results away from SMOS measurements. Increasing the error of altimetry measurements over marginal zones and thin ice surfaces or merging altimetry with SMOS estimates for ice are potential options in this multi-variate approach. Ultimately, the results of the assimilation scheme reflect a balance driven by our selection of assimilated observations: the simulation is restricted to an intermediate position between the assimilated data and the model's trajectory. Therefore, a degree of consistency between the assimilated and independent validation datasets is essential to effectively detect an improvement of the sea ice fields thanks to data assimilation techniques.

The multivariate assimilation system performs better in the Arctic than in the Antarctic, largely due to differences in the model's initial free state. In the southern hemisphere, the initial biases in the free simulation are larger than those in the northern part, making it more challenging for the assimilation to reconcile the model with observations. This highlights the critical role of the model's baseline state in a data assimilation system. Further, the significant differences in ice volume estimates due to the use of constant or non-constant densities show and confirm the importance of having a comprehensive modelled physics with observations measurements.

In the Southern Hemisphere, the results highlight the strong interactions between sea ice and the upper ocean layers. These interactions lead to complex impacts on polynya dynamics, which underlines the need for further investigation and the development of assimilation strategies that are better suited to these sensitive, coupled environments. The choice of the assimilation parameters (observation errors, localization radius) is still an ongoing work and further study in the assimilation methodology is needed to fully handle the strong coupled ocean/ice interactions at work in the Southern Ocean.

This multivariate assimilation system paves the way for the future integration of CIMR and CRISTAL satellite measurements in synergy into operational systems. The CRISTAL satellite, set for launch in 2028, will carry altimetry radar instruments equipped with both Ku-band and Ka-band radars, enabling simultaneous altimetry measurement of the air–snow and ice–snow interfaces. Moreover, a higher inclination orbit will enable measurements with a smaller hole around the North pole with the CRISTAL satellite. The CIMR satellite will measure the sea ice concentration with passive microwave imagers, allowing for sub-daily and high resolution (5 km) polar measurements. CIMR will also provide thin ice estimates from L-band radiometry, similar to SMOS.

*Data availability.* All the sea-ice reanalysis experiments are available on request. This study has been conducted using E.U. Copernicus Marine Service Product: Global Ocean Sea Ice Concentration Time Series REPROCESSED (OSI-SAF); https://doi.org/10.48670/moi-00136, available on Global Ocean Sea Ice Concentration Time Series REPROCESSED (OSI-SAF) | Copernicus Marine Service. The LEGOS data (FBR, SNOW-KaKu and SIV LEGOS_og) used in this study (doi 10.6096/CTOH_SEAICE_2019_12) were developed, validated by the CTOH/LEGOS, France and distributed by Aviso+: Altimetry Sea Ice products from CTOH. ICESat-2 total freeboard was downloaded from https://nsidc.org/data/atl20/versions/4 on the 06/06/2024, using the 'monthly' group of the netcdf files (Petty et al., 2023). SMOS Antarctic data was downloaded from Tian-Kunze, X; Kaleschke, L (2021): SMOS-derived sea ice thickness in the Antarctic from 2010 to 2020 (pangaea.de), version 3.2, last accessed on the 14/08/2024. The merging of CryoSat-2 and SMOS data (CS2SMOS) was funded by the ESA project SMOS & CryoSat-2 Sea Ice Data Product Processing and Dissemination Service and data from 01/12/2016 to 27/03/2019 were obtained from https://www.meereisportal.de (grant: REKLIM-2013-04, Ricker et al., 2017). The data presented in the Appendix A consists in the BGEP ULS measurements, collected and made available by the Beaufort Gyre Exploration Program based at the Woods Hole Oceanographic Institution (https://www2.whoi.edu/site/beaufortgyre/) in collaboration with researchers from Fisheries and Oceans Canada at the

Institute of Ocean Sciences; the ULS measurements in the Fram Strait are from the website https://data.npolar.no (Sumata et al., 2021); and the Operation IceBridge Quick Look measurements, available at https://nsidc.org/data/nsidc-0708/versions/1 (Kurtz et al., 2016).

*Author contribution.* AC, GG and CET designed the analysis and the experiments and AC carried them out. AC and GG wrote the paper and CET, GR, MH and PYLT revised it. MH and GR helped with the experiments' setup. FG provided the LEGOS datasets and shared valuable insights on the satellite altimetry observations. All named authors have participated in the present article and have brought contributions to the elaboration of its final version.

*Competing interests.* The authors declare that they have no conflict of interest.

*Acknowledgements:* Special thanks to Sara Fleury, and the LEGOS team for providing the altimetric satellite observations (radar freeboard and snow depth data) and for their useful advice regarding these observations.
Thank you to Guillaume Samson, Clément Bricaud and Laurent Parent from Mercator Océan for technical assistance and support in conducting the experiments.
This study was supported by Mercator Ocean International (France) and Centre National des Etudes Spatiales (CNES, France) as part of the doctoral grant of AC.
Provision of datasets used within this study is also acknowledged: sea ice concentration products from OSI-SAF and NSIDC, ERA5 atmospheric reanalysis from ECMWF, ICESat-2 data from NASA, CS2SMOS and SMOS ice thickness from the Alfred Wegener Institute Helmholtz Centre for Polar and Marine Research (AWI), BGEP ice drafts from the Beaufort Gyre Exploration Program based at the Woods Hole Oceanographic Institution, Fram Strait ice drafts from the Norwegian Polar Institute, Operation Ice Bridge Quick-Look ice thickness from the NASA National Snow and Ice Data Center.
The Scientific colour maps lipari and vik (Crameri, 2023) are used in this study to prevent visual distortion of the data and exclusion of readers with colour-vision deficiencies (Crameri et al., 2020).

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
