# Peer review of "Assimilation of radar freeboard and snow altimetry observations in the Arctic and Antarctic with a coupled ocean/sea ice modelling system"

_EGUsphere, 2024_

## Author Comment (AC1)

**RC2**

**1 General comments**

**We would like to thank the reviewer for their careful reading of the manuscript and the helpful comments. The answers to the comments are written in bold, the reviewer's comment in normal font, and the sentences added to**
5 **correct the manuscript in italic.**

Review of "Assimilation of radar freeboard and snow altimetry observations in the Arctic and Antarctic with a coupled ocean/sea ice modelling system" by Chenal and co-authors.
I am mostly an expert of sea ice data assimilation, but not so much of altimeter remote sensing.

The manuscript presents "the first implementation of a multivariate sea ice assimilation scheme in both the Arctic and Antarctica within a global ¼° modelling and analysis system", as stated by the authors, which is correct to my knowledge, other comparable studies are cited in the text but, at much lower resolution.
The study takes a data assimilation system as used operationally and develops both a multivariate (multiple sea ice variables,
15 though not ocean variables) assimilation scheme assimilating two new data sources on top of sea ice concentrations: radar freeboard and a novel snow thickness product. This represents an ambitious piece of work, of high technical complexity, associated with expensive computations.

The choice of satellite observations is of high standards and makes the experiments very relevant and timely in the present
20 literature.
There are however some weaknesses that the authors should address before the paper is accepted for publication.
**Thank you for your assessment of our paper. We answer each of the concerns in the following discussion.**

The two assimilation experiments do not allow to evaluate separately the multivariate assimilation scheme from the effect of
25 the assimilated data (multidata), as intermediate combinations of multivariate-monodata experiments would have allowed. The only exceptions are the locations and times when one or two input data types are missing. The paper is therefore shy on practical take-home messages. Rather than expensive additional experiments, the discussion of the benefits or weaknesses of the multivariate scheme could exploit better these special cases. In the present state of the manuscript, the two issues are mixed and hard to disentangle.
30 **Thank you for this remark. Given the distinction monodata/multidata and monovariate/multivariate for the assimilation system, different sets of experiments could be done to evaluate the different system's impacts: monodata+monovariate (UNIVAR), multidata+multivariate (MULTIVAR), but also the in-between configuration monodata+multivariate. Only the first two configurations' results are presented in the manuscript.**
**However, another experiment with a monodata+multivariate configuration was performed: it consisted in**
35 **assimilating the SIC OSISAF SSMIS product only (monodata) using the same multivariate assimilation system as the MULTIVAR experiment described in the manuscript (multivariate). As such, the information for the SIC SSMIS observation is propagated to the 5 increments (SIC, SIV, SNV, RFB, and SNT) by the Kalman filter, using the model covariances from the background error covariance matrix. We call this experiment Monodata/multivariate in this discussion. The authors chose not to present the results of this experiment as the manuscript is already long, we**
40 **wanted to provide a clearer message as we believe that this experience does not add any value to the discussion in the paper. You can find in the supplementary materials (file: 2_Supplementary Materials) a short assessment of the Monodata/multivariate experiment, with illustrations. This experiment shows intermediate performances between monodata/univariate and multidata/multivariate for the sea ice concentration (Figure S16) and similar performance as monodata/univariate for the leads content (Figure S17). Similarly, this experiment provides intermediate**
45 **performance for the RFB (Figures S18 and S19). The sea ice volume in this experiment increases the sea ice volume**

**compared to the monodata/univariate simulation but is significantly less than the multidata/multivariate experiment and far from the LEGOS altimetric observations, in both hemispheres (Figure S22). The sea ice volume estimate is significantly less with a mean of 6.75 km$^3$ and 5.74 km$^3$ for the Monodata/multivariate experiment in the Arctic and Antarctica respectively, compared to 8.73 km$^3$ and 10.80 km$^3$ for the MULTIVAR experiment, over the entire simulation period. The sea ice volume in Monodata/multivariate still increases the sea ice volume compared to the UNIVAR simulation, in both hemispheres. Further, RFB, SNV and SIV in this experiment have similar biases to those of the experiment without assimilation and the monodata/monovariate experiment. More work is needed to evaluate the impact of the monodata/multivariate method used here, which is another reason why we chose not to present this simulation in the paper.**

**We propose to add a few sentences in the methodology of the experiments in the paper to inform the reader that the Monodata/multivariate experiment was performed and that the results were not presented to avoid confusions,stay with the scope of the paper and because this experiment needs further investigations:**

**L.164**: *"Assimilation systems can be described by the terms monodata or multidata, depending on the number of observations assimilated. Two different methods exist for the assimilation system: univariate and multivariate. They refer to the number of variables in the Kalman filter state vector, determining for which variables the increments are created. In a univariate configuration, the Kalman filter runs for each observation to create only one increment. In a multivariate configuration, multiple analysis increments are created at once, using the model covariances to simultaneously correct a number of variables in a coherent manner. Hence, different assimilation systems could be defined: monodata/univariate, monodata/multivariate and multidata/multivariate.*

**L. 198***: the sentence "The different experiments presented in this paper show the evolution of the sea ice assimilation methods from a univariate and mono-data system, updating only SIC, to a multivariate and multidata setup." has been removed as the rationale for the methodology is explained in subsection 2.3.*

**L. 237***: "To assess the impact of the multivariate and/or the multidata approach versus the more widespread SIC monodata/multivariate assimilation approach, we have considered the most relevant approaches that can be combined with a single-variety or multi-variety approach and the use of data in multi-data or single-data mode. We performed a monodata/multivariate experiment assimilating the SIC OSISAF SSMIS product only with the multivariate assimilation system described previously. The results of this experiment are presented in supplementary material (Section 2) to let the article focus on the major differences brought by the innovative multidata/multivariate configuration. We then restricted the study to the comparison of the results using the monodata/univariate and the multidata/multivariate configurations."*

Second, some explanations are omitted, or left implicit, which hampers the fluidity of the arguments: I am for example missing a paragraph in the introduction that sets upfront what the experiments are expected to deliver (which improvements and which possible degradations) and what are yet open questions. In practical terms, this means that the research questions should be coming two paragraphs earlier in the introduction and should be linked more concretely to the results. The physical instability mechanisms of the Southern Ocean polynyas are also exposed too late in the paper and should have been given earlier in the introduction.

**We agree and propose to add sentences, in the introduction part, on the possible degradation with the implementation of MULTIVAR given that covariances SIC/SIV are not necessarily informative. We also briefly mention the ice-ocean interactions and the existence of polynyas. L.107:** *"Prior studies have shown that assimilating SIC alone significantly reduces concentration errors but yields limited improvement in ice thickness, despite strong correlations between both variables (Lisæter et al., 2003, Duliere and Fichefet, 2007). Up to now, as far as we know, no a priori links between SIC and the depth of the snow over sea ice have been studied. We therefore anticipate the following outcomes for each experiment: monodata/univariate assimilation should improve modeled SIC but may degrade SIT and SNT due to the necessary adjustment for SIV and SNV implemented in the analysis scheme (Table 2). Conversely, the multidata/multivariate assimilation is expected to better fit all assimilated variables (SIC, RFB, SNT), but may impact SIC accuracy due to uncertain SIC-SIT/SNT covariances. The different spatio-temporal resolutions of SIC, RFB, and SNT (e.g. daily gridded SIC vs. sparse altimeter tracks with seasonal gaps) may also introduce uncertainty into the impact of assimilation. Finally, few studies have focused on the constraints of the ice/snow system by assimilation in Antarctica, a region where the interaction between the ice and the upper ocean is much more dynamic than in the Arctic. In regions of open water surrounded by sea ice — known as polynyas — the ice-ocean interactions are particularly strong (e.g. Kjellsson et al., 2015, Cheon and*

*Gordon, 2019) and difficult to reproduce by models (Mohrmann et al., 2021). The outcomes of the assimilation experiments could reveal whether improvements in SIC are offset by errors in SIT/SNT, how additional data sources interact, and how the scheme affects coupled ice–ocean behaviour.*"

**Added references:**

**Cheon, W. G. and Gordon, A. L.: Open-ocean polynyas and deep convection in the Southern Ocean, Sci. Rep., 9, 6935, https://doi.org/10.1038/s41598-019-43466-2, 2019.**

**Dulière, V. and Fichefet, T.: On the assimilation of ice velocity and concentration data into large-scale sea ice models, Ocean Sci., 3, 321–335, https://doi.org/10.5194/os-3-321-2007, 2007.**

**Kjellsson, J., Holland, P. R., Marshall, G. J., Mathiot, P., Aksenov, Y., Coward, A. C., Bacon, S., Megann, A. P., and Ridley, J.: Model sensitivity of the Weddell and Ross seas, Antarctica, to vertical mixing and freshwater forcing, Ocean Model., 94, 141–152, https://doi.org/10.1016/j.ocemod.2015.08.003, 2015.**

**Lisæter, K. A., Rosanova, J., and Evensen, G.: Assimilation of ice concentration in a coupled ice–ocean model, using the Ensemble Kalman filter, Ocean Dyn., 53, 368–388, https://doi.org/10.1007/s10236-003-0049-4, 2003.**

**Mohrmann, M., Heuzé, C., and Swart, S.: Southern Ocean polynyas in CMIP6 models, The Cryosphere, 15, 4281–4313, https://doi.org/10.5194/tc-15-4281-2021, 2021.**

The experimental setup may have omitted some information that is important to understand the results later on. I was missing a clear indication of where the RFB and KaKu snow data are assimilated until I found Figures 3 and 4. More importantly, I am missing an explanation of what the assimilation does in the absence of one or two of these datasets. Perhaps introducing the observations before the data assimilation method could make the logic more fluid. The authors do not lay out the limitations of the data assimilation method upfront, so the reader discovers them by surprise as the negative results appear. For example, when the MULTIVAR scheme deteriorates the SIC in Figure 2. I would expect - by elimination - that the assimilation of RFB is responsible for this bias. Since the multivariate (negative) correlation between RFB and SIC stems from a long model simulation, it could be that this long simulation is to be blamed. See for example Counillon and Bertino (2009) although in an ocean-only application (as an example, you do not have to cite this reference).

**Thank you for your comment.**

**We agree and put the description of the assimilated data before the assimilation methodology. Moreover, a table (see Table 2 below) has been added to provide a clearer view of the availability, spatially and in time, of the assimilated observations. We have added the comments on Table 2 in the subsections describing SIC, RFB and snow thicknesses observations:**

**L. 203**:*" Ocean and Sea Ice Satellite Application Facility (OSISAF) OSI-450 (OSI SAF, 2022) (Table 2)."*

**L. 224:** *"The data are only available during winter in both hemispheres, November to April in the Arctic and May to October in the Antarctic (Table 2). Apart from north of 88°N, CS2 satellite tracks cover the entire ice domain of each hemisphere in about a month: during each assimilation cycle, important areas remain unobserved, especially at lower latitudes (Antarctica)."*

**L.230:** *"The data are provided in monthly gridded files, available during the same winter periods as RFB, in each hemisphere (Table 2)."*

**L. 231:"** *Due to SARAL orbital characteristics, no data are available for latitudes higher than 81.5°N".*

L.199:

| Observations | SIC SSMIS | RFB-LEGOS | SNOW-KaKu |
|---|---|---|---|
| Producer | EUMETSAT OSI-SAF | LEGOS | LEGOS |
| Temporal resolution | Daily | 20 Hz | Monthly → weekly (linear interpolation) |
| Temporal coverage | All-time | Winter: November to April in the Arctic; May to October in the Antarctic. | |
| Spatial resolution | 40 km (effective resolution); 25 km (grid resolution). | Along-tracks | 12.5 km (grid resolution). |
| Spatial gaps | None (reprocessed). | Central Arctic (latitude > 88°N); in-between satellite tracks. | Central Arctic (latitude > 81.5°N); coastal areas. |
| nsel | 400 | 4000 | 400 |

135 *Table 2: Assimilated observation products and their specificities.*

**Figure below shows the full extent of the SAR/SARin instruments measuring regions used to get the radar freeboard data. The SNOW KaKu product that is assimilated in the MULTIVAR experiment only uses the SAR mode measurements, so snow depth along the coastlines is not available in that observation product. Moreover, the Ka radar measurements used for the SNOW KaKu product comes from the SARAL-AltiKa satellite, which is limited to**
140 **81.5°N due to its orbit.**

[Figure]

**Figure RC2fig1: Geographical masks used for SAR and SARIn modes (version 5.0) for CryoSat-2 measurements in the Arctic and Antarctica.SAR and SARin modes activation regions in the Arctic and Antarctica. Data from ESA: https://earth.esa.int/eogateway/instruments/siral/geographical-mode-mask.**

145

**In the absence of one or two of the datasets, the assimilation system builds the increments using the covariances calculated from a long simulation. This long simulation is indeed biased (the detailed analysis of this long simulation is not presented in the paper). A sentence is added in the "Assimilation scheme" section to explain more clearly this process: L.171:** *"Similarly, the model covariances are used to create increments even if the full set of observed data to be*
150 *assimilated is not available. These covariances, constructed from a "long" free simulation, will therefore potentially spread the biases (partially presented in this paper) to all the variables concerned."*

Some of the minor design choices could have been better justified, for example the threshold on snow depths, which is not justified and comes back as a limitation later on.

**Since submitting this article, an experiment has shown that the snow threshold is not responsible for the unexpected**
155 **results obtained with snow in the UNIVAR simulation. We agree with your comment and decide to remove the comments related to this algorithm's small specificity. Specifically, we removed the sentences L.188:** *"If the updated SNT exceeds a threshold defined as half the analysis SIT, it is capped to avoid unrealistic values. In such cases, the total snow volume may decrease compared to the forecast."***; L.552:** *"with a dynamic threshold on the SNT"***, L.555:** *"This result shows that our threshold is not appropriate in most of the Antarctic, and in some regions in the Arctic."* **and L.556:** *"A*
160 *modification of the SNT threshold would improve the snow assimilation algorithm in that sense."***; L.736:** *"analysis snow depth threshold"***.**

The snow thickness is the difference between AltiKa and CryoSAT-2 RFBs, so it is not independent from the CryoSAT-2 RFB, whereas the assimilation system assumes observations errors are independent. Although I do not see an immediate solution for that, this may be acknowledged in Section 2.2.3.

165  **That is right. A sentence has been added to the manuscript to account for this information L.235:** "*Multiple processing are applied to the Ku-band CryoSat-2 measurements to create the SNOW-KaKu product: a degraded version of the SAR measurements (pseudo-LRM mode) is used to get a similar footprint as the SARAL-AltiKa measurements, a 25 km radius median smoothing is applied, and the data is gridded at a monthly frequency, as described by Garnier et al. (2021). However, the SNOW-KaKu product remains not fully independent from RFB-LEGOS measurements.*".

170

The text puts the same emphasis on results that are trivial (i.e. that assimilation runs agree better with the assimilated data) and those that are less obvious. The discussion could flow more easily if the description of the experiments had a few sentences with the a priori expectations from each run.

**See answer above with a paragraph added in the introduction part to set upfront what the experiments are expected**
175  **to deliver (which improvements and which possible degradations). We also add these few sentences to put emphasis on the obvious results:**

**L.239: "**
● *FREE: experiment without any assimilation, which has consistent biases in all sea ice variables due to model and forcing limitations, providing a baseline for evaluating the impact of assimilation.*
180  ● *UNIVAR: experiment similar to the current operational system, using the previously described univariate SIC assimilation method. Assimilating SIC alone is expected to significantly reduce sea ice concentration errors but may induce unrealistic adjustments in sea ice thickness (SIT) and snow depth (SNT).*
● *MULTIVAR: experiment with the multivariate assimilation scheme described previously, assimilating SIC, RFB and SNT observations, and updating the SIC, SIV and SNV model variables. Assimilating multiple variables is anticipated to*
185  *improve agreement with all assimilated observations (SIC, RFB, SNT), though possibly at the cost of reduced SIC accuracy and increased risk of numerical or dynamical imbalances, especially in a coupled ice–ocean model.***"**

**And in the conclusion, we highlight the results that were not expected and highlight the results in a more positive manner:**

**L.719: "***Despite the heterogeneous nature and varying resolutions of the assimilated data sets, the multidata/multivariate*
190  *assimilation system demonstrates robust behavior even in the absence of certain observations (summer, spatial hole), indicating a consistent and physically coherent adjustment of the sea ice state.*"

**L.735, we replace the sentence "***The results for the southern hemisphere also show the strong interactions with the oceanic surface layers in the life cycle of the sea ice cover.***" with** *"In the Southern Hemisphere, the results highlight the strong interactions between sea ice and the upper ocean layers. These interactions lead to complex impacts on polynya dynamics,*
195  *which underlines the need for further investigation and the development of assimilation strategies that are better suited to these sensitive coupled environments.*"**.**

The case for assimilating RFB rather than SIT could be more conclusive. As long as the auxiliary data used in the RFB-to-SIT conversion are the same, there is no obvious benefit of assimilating RFB rather than a converted SIT product. The ice
200  and snow density do not seem to take an active role in the sea ice model so belong in the observation operator. However the snow depth is a proper state variable, so I would expect that more accurate (satellite) snow depths justifies alone the assimilation of RFB. By opposition, the assimilation may lead to deteriorations of SIT in areas where the KaKu snow depths are missing, but I could not find any evidence of that in the manuscript.

**Sentences in the introduction stress how the conversion RFB-SIT depends on the choice for alternative parameters**
205  **such as the snow depth, and the ice and snow densities: L.66 "***The sea water, ice and snow densities and the snow depth above the ice are required for the RFB-SIT conversion, and the assumptions made on these variables result in a significant uncertainty in the sea ice volume products (Kern et al., 2015; Kwok and Cunningham, 2015). The snow layer accounts for most of the uncertainty in the calculation of SIT from RFB (Garnier et al., 2021).***"; L.79 "***Other sources of uncertainty in the RFB-SIT conversion stems from the choice of ice and snow densities.***".**
210  **The choice of assimilating RFB rather than a SIT product comes from a desire to stay closer to the satellite measurement. Our experiment is a first step showing that direct RFB assimilation is possible, with a full high-resolution along-tracks product. No interpolation is needed, and the data is available at a sub-daily frequency since there is no post-processing of the measurements. Then, future studies will be able to build upon this framework and**

**implement varying snow and ice densities, and to reproduce the parameterization used in the observations for the model.**

**Thanks to a direct RFB assimilation, we can also constrain the snow depth separately, which profoundly affects the sea ice state. We hope, in the future, to assimilate the snow measurements along-tracks.**

**We did not compare these experiments to an experiment assimilating a SIT product.**

**Concerning the deterioration where snow KaKu data is missing, we noticed a very localized high bias in the north of Greenland, which could be due to the lack of snow observations in this area for the MULTIVAR simulation in April 2017 (Figure 3c), see L381 of the preprint. However, we did not notice any particular bias in the large snow KaKu observation gap around the North pole. We added the following sentences to clarify this finding:**

**L.585: "***However, the largest RFB differences between the MULTIVAR experiment and the RFB LEGOS assimilated observations are located on the north of the Canadian Archipelago and Greenland, with an especially thin RFB in our simulation locally north of Greenland. No snow observations are available in this area, and the MULTIVAR presents thicker snow values than the FREE and UNIVAR simulations. No particular RFB bias is present in the large snow KaKu observation gap around the North pole, suggesting that in the absence of snow observations, an inaccurate modelled snow depth does not affect the RFB assimilation performance on a large scale, but can result in higher RFB biases very locally.***"**

The manuscript overall structure is good, with a few exceptions noted above. However some repetitions could be avoided across different sections. The writing style is generally good and very careful, sometimes too careful to the point of becoming confusing. The statements are often too neutral, not indicating whether the results are as expected, good or bad. This hampers the reading of the paper.

**We have taken care to reduce repetitions and improve clarity in the sentences that were modified or added in response to reviewers comments. In these revised sections, we made a deliberate effort to be more direct and explicit about the results.**

The graphics are very clear and highlight well the main messages.

**Thank you.**

Overall I am very positive that the study makes an important and interesting contribution to the field and is worthy of publication in The Cryosphere, provided that the issues pointed out above are corrected in a revised version.

**2 Detailed comments**

- L40 Chen et al. 2024 is not in the Arctic but in an idealised square model.

**Thank you. This citation was removed in L. 40: "***Experiments have used EnKF or variations of this multivariate scheme with multidata frameworks: both SIC and SIT products have been assimilated in the Arctic (e.g. Cheng et al., 2023; Williams et al., 2023)***".**

- L100 Since only RFB_mD is used in the experiments, it could be mentioned that RFB_og is only shown indicatively for users of the original product.

**Thank you. The sentence has been added L. 101: "***Data using varying sea ice and snow densities are only shown in the figures indicatively for users of the original product.***"**

- L120 The section on the sea ice model does not mention the treatment of submerged sea ice. Since negative RFBs occur in the results it is necessary to know if the assimilative model will freeze submerged snow into more saline ice.

**A sentence has been added L.132 to explain the process in the model: "***Snow exclusively comes from the solid precipitations of the atmospheric forcing and disappears either by melting processes or by snow-ice conversion when the snow base gets below the sea level. The model accounts for snow-ice formation when snow is deep enough to depress the***

*snow-ice interface below the sea level. Then seawater infiltrates and refreezes into the snow, creating a new ice layer whose thickness depends on the ice and snow densities (Fichefet and Maqueda, 1997; Vancoppenolle et al., 2023)."*

**Added reference:**

**Fichefet, T. and Maqueda, M. A. M.: Sensitivity of a global sea ice model to the treatment of ice thermodynamics and**
**dynamics, J. Geophys. Res. Oceans, 102, 12609–12646, https://doi.org/10.1029/97JC00480, 1997.**

**A sentence is included in the article to explain this negative RFB phenomenon L.214: "***Radar freeboard values can be negative because of the term accounting for the radar speed reduction in the snow layer: it is not a real physical distance contrarily to ice freeboard.***". We change it to: "***Radar freeboard measurements depend on the radar speed reduction in the snow layer and are consequently not physical measurements. The ice/snow interface is therefore not necessarily underwater when the RFB is negative.***". When a radar freeboard is negative, it is mostly due to a thick snow layer and a thin ice layer.**

- L135 The data assimilation scheme does not mention that the ensemble is static and indirectly indicates a "long simulation without assimilation". The - strong - assumption that the long simulation is representative of the model errors during the assimilation period should be stated explicitly, including an indication of the years of the long simulation.
**The manuscript has been corrected L.146 to explain that "***The static anomalies are computed from a long simulation (2010-2020) without assimilation, using the same model configuration and parameters with respect to a 7-day running mean. This approach is based on statistical ensembles in which the ensemble of these anomalies is representative of the error covariances (Lellouche et al., 2013).***".**

- L148 Why the reduced grid for the ocean? Maybe this point is unimportant for the paper and should be omitted to simplify this section.
**We agree to remove this precision. L.146 the sentence "***Anomalies are computed on a reduced grid for the ocean (1 out of 2 points) and on a full grid for the sea ice.***" has been removed from the manuscript.**

- L162 Indicate that assimilating the OSTIA freezing point temperature is intended to make the sea ice and ocean assimilation consistent with each other. If the freezing point temperatures in OSTIA and NEMO differ due to sea surface salinity, can the difference induce sea ice melt or freeze?
**We didn't assess the impact alone of such assimilation of OSTIA. In the model physics, the SST is set to the freezing point to maintain thermodynamic equilibrium with the ice above. If SST is above (resp. under) the freezing point, a heat flux is estimated to melt (resp. create) ice at the bottom.**

- L178 Why a Gamma distribution? What are its parameters?
**Several results have shown a gamma distribution of ice thicknesses in the observations. We have applied it to the model distribution with the parameters k=2.0 and theta=0.4 for the gamma law, see the figure below. We have modified the text L.178: "***Then, the total ice concentration is redistributed into each existing thickness category using a Gamma-type distribution commonly found in observed measurements (Toppaladoddi et al., 2023; Petty et al., 2020). This chosen distribution (with parameters k=2.0 and theta=0.4) adds most of the increment to the middle and smallest thickness categories and less to the extreme categories.***"**
**The following references have been added to the reference list:**
**Toppaladoddi, S., Moon, W., & Wettlaufer, J. S. (2023). Seasonal evolution of the Arctic sea ice thickness distribution.** *Journal of Geophysical Research: Oceans*, **128, e2022JC019540. https://doi.org/10.1029/2022JC019540**
**Petty, A. A., Kurtz, N. T., Kwok, R., Markus, T., & Neumann, T. A. (2020). Winter Arctic sea ice thickness from ICESat-2** **freeboards.** *Journal of Geophysical Research: Oceans*, **125, e2019JC015764. https://doi.org/10.1029/2019JC015764**

[Figure]

**Figure RC2fig2: Sea ice categories, defined by the values [0., 0.15, 0.32, 0.51, 0.74, 1.00, 1.33, 1.73, 2.21, 2.83, 3.61, 99.00]; and the redistribution function for the SIC increment in the different ice categories.**

- L183 Are there any cases when the SIT exceeds the bounds of its thickness category? How is that handled by the model?
**In the assimilation algorithm, in the case of a positive SIV increment, the increment is added first to the thinnest ice category, increasing its thickness until it reaches the upper limit. Then, the remainder of the increment is added to the next category, and the algorithm keeps doing so until the whole SIV increment is used. Similarly, in the case of a negative increment, the algorithm begins with removing ice from the thinnest ice category, and when there is no more ice in said category, it moves up a thicker category to continue removing ice until the increment has been entirely used up. A sentence describing this process is added L.183: "***If the change of thickness of a category exceeds its bounds, any excess or deficit in volume is transferred to the next thicker category, and this redistribution continues until the entire SIV increment is applied.***"**

- L189 This threshold should be explained since I did not expect it to become important later. This is especially mysterious since it reduces snow depths that are otherwise allowed in the forward model.
**The discussion about the snow threshold has been removed in the new version of the manuscript since you rightly said that it was unclear and since a new experiment showed no impact of this threshold on the simulation's results.**

- L235 I am not sure what this statement refers to. By construction the SNV depends directly on SIC and on SNT so the Kalman filter should estimate it well, unless the long free simulation has a pathological behaviour.
**Yes, that is true, we chose to remove the sentence L.235: "***It is important to note that the snow volume increment depends on all the assimilated data and reflects how well the volume correlates with them.***".**

- Tables 1 and 2 are very welcome to summarise the complex information, but Table 2 could include a little more information like the seasons and latitudes at which the different observations are available, which would prepare the reader to what happens when and where RFB or snow are not assimilated.
**A new table (see *Table 2* below) has been added to the manuscript. The different resolutions and data gaps of the 3 assimilated products are described in this new table.**

| Observations | SIC SSMIS | RFB-LEGOS | SNOW-KaKu |
|---|---|---|---|
| Producer | EUMETSAT OSI-SAF | LEGOS | LEGOS |
| Temporal resolution | Daily | 20 Hz | Monthly → weekly (linear interpolation) |
| Temporal coverage | All-time | Winter: November to April in the Arctic; May to October in the Antarctic. | |

| Spatial resolution | 40 km (effective resolution); 25 km (grid resolution). | Along-tracks | 12.5 km (grid resolution). |
|---|---|---|---|
| Spatial gaps | None (reprocessed). | Central Arctic (latitude > 88°N); in-between satellite tracks. | Central Arctic (latitude > 81.5°N) ; coastal areas. |
| nsel | 400 | 4000 | 400 |

Table 2: Assimilated observation products and their specificities.

340   - L255 Ivanova et al. compared 11 algorithms. Coming down to 2 is not really quantifying the uncertainty. There are weaknesses of PMW observations in the summer and different choices of tie points made at OSI-SAF and NSIDC to cope with them. Maybe they provide a lower and upper bound rather than an uncertainty estimate.

**The lower and upper bound part might be true, but I did not check whether or not these products actually estimate the maximum and minimum values. We modify the sentence to describe it as "***showcase the variety of estimates in the***
345 ***different observation products***".**

  - L270 The SSMIS data is assimilated, not the NSIDC, so it should be expected that the results agree better with SSMIS.
**It is the free simulation that compares favourably with NSIDC, perhaps by chance. It also shows that observations can have very different estimates.**
350

  - L307 Between 0.04 and 0.13 is rather vague, can you provide the number?
**The exact RMSE mean value of 0.08 has been added in the manuscript.**
**L.307:** *"During the other months, the RMSE of 0.08 for the MULTIVAR simulation is lower, falling between the mean RMSEs of the UNIVAR and FREE simulations, which are 0.04 and 0.13, respectively."*
355

  - L318 "Unobserved polynyas" This sounds like the blame is on observations, unless you mean "too large polynyas"? An oceanographic discussion of the ice-ocean conditions for polynyas should precede this sentence but is only coming near the end of the paper.
**Here, the word "unobserved" was meant to point out that even if the polynyas do not appear in the observations, they**
360 **could still exist (because the observations may be too low resolutions or may have missed the detection of the open waters) or they could be an artifact of the model (this often happens, see for instance Mohrmann et al. (2021)). To be clearer, the sentence was modified to "***In the Antarctic, the two assimilated experiments generate  a large number of polynyas which are not detected bythe satellite observations, with the MULTIVAR experiment showing them more frequently and broadly across the region (Figure S2). While some smaller polynyas may go undetected in the observational data, the***
365 ***modelled polynyas are likely overestimated***.".**

  - L324 Shouldn't it be expected that the assimilated run matches the assimilated data?
**We mitigate this statement L.324**: *"The MULTIVAR snow distribution is logically very close to the Arctic SNOW-KaKu during winter..."*
370

  - Comparing Figure 3 to Figure 2, I see that the deterioration of SIC in the summer happens outside of the KaKu snow data coverage, so it appears to be caused by the assimilation of RFB. Since the free model underestimates the RFB, I expect that the positive innovation of RFB turns into a negative increment of SIC. Is it so that the RFB is somehow negatively correlated to the SIC in the summer covariances, and is that model-based correlation trustworthy?
375 **The SIC deterioration area in the Arctic summer for the MULTIVAR experiment is actually larger than the orbital gap of the snow KaKu observation (see figure below). We did not analyze the correlations between the different variables in our simulations, so we are not able to be conclusive on this question. The RFB indeed seems negatively correlated to the SIC in the Arctic summer. The model based-covariances are however biased since they are based on a long (2010-2020) free simulation. This long simulation has issues maintaining the ice cover in the northern**
380 **hemisphere in summer, with a sea ice extent that falls below the one of the SSMIS OSISAF observations. Hence, the model-based correlations in the summer may not be reliable.**

[Figure]

**Figure RC2fig3: Same as Figure 2 of the paper, only keeping the MULTIVAR experiment's result, highlighting which areas are covered by the KaKu snow depth measurements (right) and which are not (left).**

- L355-360 If the FREE run has excessive sea ice cover, does it also receive snow that should have fallen in open water? Does that explain why UNIVAR does better?

**We assume that this comment refers to excess snow in the FREE experiment and not ice cover. This is indeed the case, as the FREE experiment has a larger surface area covered by ice than the UNIVAR experiment, so it intercepts more snow precipitation. We have not quantified this effect.**

- L365 Conversely, is the missing snow falling into too large simulated polynyas?

**Yes, for the same amount of snowfall, MULTIVAR should intercept less snow because of the presence of open waters. A regional study would be needed to quantify this effect.**

- L370 With local data assimilation, one would expect that the spatial distribution is respected.

**Yes, the sentence L.370 has been modified from : "***The assimilation of SNT is also able to rapidly modify the snow spatial distribution in accordance with the SNOW KaKu observations distribution.***" to "***While a localized assimilation scheme is expected to modify the spatial distribution of the variable to match the observations, it is noteworthy that the assimilation of SNT leads to rapid corrections, with most spatial biases already reduced within the first month of the simulation (Figure S3).***".**

- L376 The small bias should be expected for the assimilated observation.

**Yes, this statement has been mitigated to account for that remark L.376 "***The MULTIVAR simulation logically exhibits a very small bias of -0.5 cm in the assimilated region and a RMSE of 2.2 cm.***"**

- L380 The observation errors should be given earlier in Section 2.2.2.

**The observation errors are not constant. We have added this precision in the section 2.2.1, 2.2.2 and 2.2.3 as follows:**
**L..206:" We** *use the daily- and spatially-varying "standard_error" provided with the dataset to construct the observation error for the assimilation ...*"
**L. 217:" We** *use the uncertainty provided for each track as the observation error, constraining it to a range of 0.01 m to 5 m.*"
**L. 232:" The** *observation error used in the analysis comes from the monthly varying uncertainty supplied with the data, constrained to an arbitrary range of 0.01 m to 5 m.*"
**The values given L.380 refer to the mean values of the discussed month, namely April 2017. The sentence "***The SARin data are assimilated with higher observation errors compared to SAR data, with mean values of 19.2 cm and 9.2 cm, respectively.***" refers to the monthly mean values, averaged over the northern hemisphere ice-covered area. The term "typically" is added to emphasize that the error varies for each measurement: "***The SARin data are provided with higher observation errors compared to SAR data.***"**

- L383 More positively skewed. Explain why the values of a sea ice variable like RFB should be skewed.

**The positive skewness of the RFB distribution arises because sea ice freeboard typically has a long tail toward thicker ice, particularly in regions where thick, deformed ice (e.g., ridges) persists. The sea ice thickness is also positively skewed (Toppaladoddi and Wettlaufer, 2023), and this property of the sea ice thickness is reflected in the RFB measurements. This results in a positively skewed probability density function, which is more pronounced in the MULTIVAR experiment due to its improved representation of spatial variability and persistence of thick ice structures even in the absence of assimilated RFB observations in summer.**

**The sentence is modified as follows, L.383:** "*In summer, when no RFB observations are assimilated, the probability density function of the MULTIVAR RFB values remains more positively skewed than in other simulations, reflecting the persistence of localized thick ice areas among generally thinner ice.*"

- L385 Biases cancel off if you average enough of them. I don't think that biases can be compared to observation error standard deviations.

**Yes, we decided to remove the comment, L.385 "***still below the mean observation error***".**

- L392 Is the ice submerged when the RFB is negative?

**Not necessarily, see the answer above.**

- L410 The comparison to ICESat-2 is interesting but could be better introduced. For example, is the ICESat-2 total freeboard physically the same as the LEGOS (AltiKa) total freeboard? Indicate upfront that ICESat-2 is also available in the summer.

**Physically, the total freeboard measured by IceSat-2 is the same as the one measured by the SARAL/AltiKa satellite: their beams reflect on the snow-air interface. However, the measurement technique differs: IceSat-2 uses laser altimetry while SARAL/AltiKa uses radar altimetry.**

**The beginning of paragraph 4.1 is modified L.411:** "*Both ICESat-2 (Ice, Cloud and Land Elevation Satellite) and SARAL/Altika satellites measure total freeboard but the first one with a laser altimeter (Markus et al., 2017), the second one with a radar altimeter. However, the ICESat-2 product presents a smaller orbital hole (88° latitudinal limit) and a full-year availability, starting from the 14th of October 2018. The monthly ICESat-2 NSIDC ATL-20 gridded along-tracks product (Petty et al., 2023) is used on Figure 5, as a scatterplot between its total freeboard values and the total freeboard collocated in time and space for the LEGOS data and the FREE, UNIVAR and MULTIVAR experiments in the Arctic.*"

**Reference to add: Markus, T.; Neumann, T.; Martino, A.; Abdalati, W.; Brunt, K.; Csatho, B.; Farrell, S.; Fricker, H.; Gardner, A.; Harding, D.; et al. The Ice, Cloud, and land Elevation Satellite-2 (ICESat-2): Science requirements, concept, and implementation.** *Remote Sens. Environ.* **2017,** *190,* **260–273.**

- L410 Can the validation of ICESat-2 separate cases of bare ice from snow-covered ice? Could this validate the RFB alone instead of the more ambiguous RFB+snow?

**We are not aware of the existence of such a product derived from ICESat-2 measurements, does it mean a use of a different snow product to isolate bare ice and snow-covered ice?**

- L414 "the constant densities": do you mean the model densities or other constants?

**We mean the model constant densities. We added this in the manuscript to avoid confusion L414:** "*The LEGOS total freeboard is made using LEGOS RFB and SNOW-KaKu data, and the constant water, ice and snow densities of the model.*"**.**

- L416, 418, 422: should we expect that MULTIVAR yields better statistics?

**This is expected when MULTIVAR results and LEGOS data are compared but not necessarily when MULTIVAR and ICESat-2 are compared. So, we mitigate the following sentences: L. 415:** "*The MULTIVAR simulation and LEGOS data present anticipated similar linear correlation statistics (slopes and r-values), MULTIVAR has then logically better statistics than the FREE and UNIVAR experiments, MULTIVAR simulation and the LEGOS data have similar mean RMSE compared to ICESat-2 data (6.7 cm and 7.2 cm respectively) and the MULTIVAR simulation and LEGOS data also display comparable mean total freeboard in January-February 2019, with values of 22.2 cm and 22.0 cm respectively, slightly thinner than the ICESat-2 estimate of 23.7 cm.*"

- L420 Why does UNIVAR have lower freeboard than the free run?

**The lower total freeboard in the UNIVAR experiment compared to the FREE run during January–February 2019 in the Arctic is due to both reduced snow depth and reduced sea ice thickness in the UNIVAR simulation. As shown in the scatterplots below, total freeboard is calculated from the modeled snow and ice volumes per unit area, each weighted by its respective coefficient. In these plots, the individual contributions of snow and ice are shown in red and green, while their sum (the total freeboard) is shown in black. For nearly all cases, both the snow and ice components are thinner in UNIVAR than in FREE, leading to the observed reduction in total freeboard.**

**We added the precision L.420 "***due to thinner sea ice and snow cover in the UNIVAR experiment***" in the sentence.**

[Figure]

Arctic IceSat-2 scatterplots, January and February 2019

**Figure RC2fig4: Scatterplots of the monthly Arctic ICESat-2 total freeboard against FREE, UNIVAR, and MULTIVAR experiments for January and February 2019. The total freeboard model equivalents are computed following the equation:**

$$Model\ eqv\ =\ [1 - \frac{\rho_{ice}}{\rho_{water}}] * sivolu + [1 - \frac{\rho_{snow}}{\rho_{water}}] * snvolu \qquad \text{(Equation RC2eq1)}$$

**Each member of the equation is shown on the scatterplot: black dots represent the *Model eqv* ; red dots the quantity $[1 - \frac{\rho_{ice}}{\rho_{water}}] * sivolu$ ; and green dots the quantity $[1 - \frac{\rho_{snow}}{\rho_{water}}] * snvolu$. The x=y line (full grey line) and linear regressions (dashed lines) are shown for each quantity in their corresponding color.**

- L425 Do you have insights whether this is thanks to the update of SIV or to the snow that insulates the ice?

**From the figure shown above, updates of snow are rather similar in both UNIVAR and MULTIVAR; the update of SIV in MULTIVAR is the major component of the total freeboard change compared to UNIVAR.**

**A sentence is added in the article: "***The change in the total freeboard modelled by the MULTIVAR experiment is mainly due to a larger SIV, thanks to the assimilation update, compared to the UNIVAR experiment.***"**

- L427 I don't have an impression of how good a spatial correlation of 0.6 is. Maybe remove this statement.

**We agree and removed the statement L.427: "***All the experiments exhibit correlations higher than 0.6 reflecting a general consistency with ICESat-2 total freeboard in terms of spatial distributions.***"**

**However, we decide to emphasize the better statistics of MULTIVAR experiment's slopes in both hemispheres and for all represented periods by changing the sentence L.441 to "***The MULTIVAR shows a favourable systematic increase of the slopes in winter as in summer.***", and place it in the L.444 to apply it to both hemispheres.**

- L438 "unobserved ice-free zones" again implies observations are wrong, while the model obviously has excessive ice.

**The FREE simulation has a large deficit of sea ice cover during summer in Antarctica, we changed the sentence L. 438: "***The FREE experiment again has large unrealistic ice-free zones with total freeboard values at 0 cm.***"**

- Figures 5 and 6: Should we expect that the summer freeboard is lower than the winter?

**Yes, it is expected that the total freeboard is lower in summer than in winter. As shown in Equation RC2eq1, total freeboard is a linear combination of sea ice thickness and snow thickness, each weighted by coefficients dependent on**

their respective densities. During summer, both sea ice and snow melt, leading to a reduction in their thicknesses. Snow cover may also disappear entirely in some regions. Since both components contribute positively to the total freeboard, a seasonal decrease in either the ice or the snow results in a lower total freeboard in summer.

However, the figures 5 and 6 emphasize that the total freeboard is more reduced in the summer in our experiments than in the ICESat-2 measurements. See for instance the sentences L.426: "*However, compared to ICESat-2, MULTIVAR still underestimates the thickness of the total freeboard at the end of Arctic summer.*" and L.437: "*The melting season (January-February 2019) highlights the excessive thinning of the total freeboard in the simulations compared to the ICESat-2 data.*".

We add the sentence L.423: "*In summer, total freeboard has decreased during the melting season; however, the thinning is more pronounced in our simulations than in the ICESat-2 observations which does not seem to show a reduction in the mean freeboard compared with winter.*" to better describe these phenomenons.

- L493 Please clarify that the LEGOS and CS2SMOS data are not completely independent due to the use of CryoSat-2.

We changed the next sentence L. 494: "Based on CS2 measurements, the LEGOS_og logically displays a consistent sea ice thickness spatial distribution compared to the CS2SMOS product …"

- Figure 8: That the MULTIVAR is performing best in the polar hole is interesting. But I am confused by the notion of "LEGOS zone", is it the zone where RFB data or KaKu data is available?

The "LEGOS zone" refers to the region where both the RFB and KaKu snow depth products are available. The spatial extent of this zone is primarily constrained by the KaKu product, which is limited to 81.5°N due to the orbital inclination of SARAL/AltiKa and the coverage of CryoSat-2's SAR mode (excluding SARin mode). In contrast, the RFB product covers a broader region, extending up to 89°N and closer to coastlines, thanks to the use of CryoSat-2's SARin mode. Therefore, the KaKu product defines the common overlap area used in this study.

The precision is added to complete the L.460: "*Figure 7 also presents the experiments collocated within the spatial coverage of the assimilated observations, which excludes the central Arctic orbital gap and limited coverage of marginal seas (solid lines).*", we added: "*This area, where both the RFB and KaKu data are available, is hereafter referred to as the 'LEGOS zone' or the 'LEGOS observations domain'.*". We also added in the caption for Figures 8 and 9: "The LEGOS zone corresponds to areas where the KaKu snow depth is available.".

- L520 Clarify that LEGOS and SMOS data are independent.

We added the sentence L.518: "*In Antarctica, the SMOS product (Tian-Kunze and Kaleschke, 2021) detects ice thinner than 1m using brightness temperature measurements, hence the data is completely independent from the LEGOS altimetric data assimilated in the MULTIVAR experiment.*".

Added reference:

Tian-Kunze, X. and Kaleschke, L.: SMOS-derived sea ice thickness in the Antarctic from 2010 to 2020, https://doi.org/10.1594/PANGAEA.934732, 2021.

- Figure 9 has masked the areas where the SMOS data is saturated. So, it is difficult to see where the "LEGOS zone" stops. Is it at 80S latitude?

The LEGOS zone includes the areas where both the RFB and the KaKu snow depth are available. In Antarctica, this corresponds to the SAR mode mask only (see Figure RC2fig1), which excludes the SARin mask defined over a wide band along the Antarctic coast.

- L545 The section is inconclusive because the authors have not indicated which of LEGOS or SMOS data is more realistic. In the Antarctic, the SMOS thickness increases gradually into the ice, which seems more realistic to me, but I would appreciate the authors' view on this.

We thank the reviewer for this thoughtful comment. Indeed, this section shows the current uncertainty in the realism of available sea ice thickness products in Antarctica. The SMOS product may appear more physically consistent, because of the gradual increase of thickness into the ice pack. However, this product is relatively recent (described as a 'preliminary product'), and its validation over the Antarctic remains limited due to the scarcity of independent

**reference data (Kaleschke et al., 2024). The LEGOS products show some agreement with ICESat-2 observations, but**
560 **both rely on altimetry-based retrievals and may share similar biases.**
**L.630, we add a sentence to explain better this point: "***While the assimilation improves the agreement between the assimilated products, the contrasting patterns seen in LEGOS and SMOS sea ice thickness highlight the current observational uncertainty in Antarctica, making it difficult to assert which product more accurately represents the true state of the sea ice.***"**

565
- L552-557 This passage could be less contorted if the pros and cons of a threshold was stated upfront (see comment above).
**Following a previous comment, all sentences regarding the snow threshold in the assimilation algorithm has been removed from the manuscript.**

570 - L559 Even if the snow melts away completely in the summer, the timing of the snow melt also affects the melting of the sea ice below, which can hold a longer memory than the snow itself.
**We agree that, in principle, the timing of snow melt can influence the evolution of sea ice through its effect on surface albedo and insulation. However, in our experiments, we do not find direct or conclusive evidence linking changes in the timing of snow melt with a corresponding shift in sea ice evolution. Moreover, the assimilation of snow**
575 **observations does not appear to significantly alter the timing of snow melt in our simulations. We have revised the manuscript to reflect this nuance more clearly.**
**In the manuscript, the sentence L.559: "***The snow cover completely melts in summer in each hemisphere and shows no long-term effect of the winter snow assimilation.***" is modified to "***The snow cover completely melts in summer in both hemispheres, and while the timing of melt should influence the sea ice evolution, our results do not indicate a persistent or***
580 ***clearly attributable long-term impact of the winter snow assimilation.***".**

- L566 Is it unrealistic that FREE and UNIVAR have submerged sea ice?
**A negative radar freeboard does not necessarily imply that the sea ice is physically submerged. It reflects the relative distribution of snow and ice thicknesses under hydrostatic equilibrium, and the slower velocity of the radar wave**
585 **when penetrating into the snow, meaning that it is not a real distance (see previous answers). The concern in our simulations is not the negative sign per se, but the fact that the radar freeboard is significantly more negative than observed values, suggesting an imbalance in the modeled snow–ice system. Diagnostic analyses indicate that this underestimation stems from a combination of an overlying thick snow cover (particularly in the FREE experiment) and a too thin sea ice, both of which contribute to lower modeled radar freeboard compared to observations.**
590 **The revised sentence is the following, L. 566: "***In the Antarctic, the RFB is significantly underestimated in the FREE and UNIVAR simulations, reflecting an imbalance between snow and ice thicknesses: the snow cover is too thick and the sea ice too thin, resulting in radar freeboard values that are more negative than observed.***".**

- L586 This is the first time that the authors mention what happens in the absence of KaKu snow observations, and in
595 indirect wording. See previous recommendation to lay the special cases in plain sight earlier in the description of experiments.
**We modified the manuscript regarding this issue thanks to the previous comments.**

- L587 The snow and RFB are indeed related both by Equation 1 and by the model dynamics, it would be interesting to see
600 what this relationship becomes in the long model simulation.
**We have not explicitly studied these relationships, neither in free mode nor in assimilated mode, to understand the impact of the observations on these physical relationships. This is a study that really needs to be undertaken.**

- L622 Why not assimilate SMOS data as well?
605 **Thank you for the suggestion. The primary reason SMOS data were not assimilated is our aim to apply a consistent assimilation system across both hemispheres. While SMOS and CryoSat-2 sea ice thickness products are reasonably consistent in the Arctic and have been validated over several years, their agreement is much weaker in the Antarctic, where discrepancies between the two measurements remain significant. Assimilating both products without a clear**

understanding of their relative reliability in the Southern Ocean could introduce inconsistencies or reinforce biased
610   signals. That said, assimilating complementary data sources such as SMOS for thin ice and CryoSat-2 for thick ice is
indeed a promising direction; once further validation and cross-comparison efforts improve our confidence in the
Antarctic observational products.

**Added sentence L.630:** "*In the future, the system could also assimilate both CryoSat-2 (for thick ice) and SMOS (for thin ice) products in both hemispheres, provided that Antarctic sea ice thickness estimates have greater consistency and*
615   *agreement.*".

- L650 "mostly the MYI" should be "only the MYI", right?

**Thank you for this observation. While it is true that the model uses a constant ice density equivalent to that of FYI in the observational dataset, the observed ice density varies spatially as a function of the proportion of FYI and MYI**
620   **within each grid cell. Therefore, the impact of using a constant model density is most significant in regions dominated by MYI, but not exclusively limited to them — since even MYI-rich cells generally contain some fraction of FYI, which influences the cell-averaged density. For this reason, we chose to describe the affected regions as "mostly the MYI regions."**

**Modified sentence L.650:** "*Hence, assimilating radar freeboard and snow with the model constant ice density primarily*
625   *affects regions dominated by MYI in the Arctic*".

- L655 Is 895 kg/m3 the model or observation density?

**This is the observation's mean ice density. The model density is 917 kg/m3.**

**The sentence is changed to L.655:** "*The model's ice density (constant 917 kg/m3) exceeds that of the LEGOS_og*
630   *observations (895 kg/m3 in average), with a particularly significant difference in October.*".

- L666 the "physical accuracy" is misleading since the densities are only used for calculating the RFB in the output.

**While it is true that the densities are used diagnostically for calculating the RFB model equivalent in the observation operator, they also play a more active role in the physical model itself. For example, the ice and snow densities are**
635   **involved in parameterizations such as snow–ice formation processes, which are especially relevant in the Antarctic. Therefore, implementing seasonally evolving densities could improve not just the consistency of the assimilation system, but also the realism of physical processes in the model.**

**We have modified the sentence L.666:** "*Moreover, implementing seasonally evolving ice and snow densities in the model could improve the realism of key physical processes such as snow–ice formation, particularly in the Antarctic.*"
640

- L675 is another instance when multivariate covariances are discretely mentioned. This sentence could imply for example that the SIV and SNT are incorrect in summer and then feedback negatively to the SIC, but the authors should clarify if this is the case.

645   **We indeed already mentioned the impact of multivariate covariance on SIC during summer in the modified text L.171.**

**In MULTIVAR, the SIC observations are used not only to update SIC but also to generate increments in SIV and SNV via model-derived covariances. The resulting SIV and SNV increments (though unverified because there are no RFB and snow observations in summer) appear to degrade SIC, suggesting that the covariances used to propagate**
650   **SIC information to other variables may be inaccurate or less reliable during summer. Alternatively, the structure of the increments or the dynamical model's sensitivity could also contribute to the degradation. Without direct observations of SIV or SNV in summer, we cannot isolate the exact cause, but the degradation of SIC performance in MULTIVAR highlights the need to improve or seasonally adapt these multivariate relationships.**

**Modified sentence L.673:** "*The degradation of modelled SIC in summer in the MULTIVAR configuration, while UNIVAR*
655   *uses the same SIC observations, suggests that the multivariate assimilation may introduce erroneous corrections through model covariances between SIC, SIV, and SNV. These propagated increments, applied in the absence of direct summer observations of SIV or SNV, appear to deteriorate SIC consistency, highlighting the need to reassess or seasonally adapt the covariances used in the assimilation.*".

660     - L682 Indicate whether these open waters are correct or artificial.

**The open water features present in the multivariate experiment are not detected in the assimilated SIC SSMIS observations and appear too large and persistent to be considered realistic. While we cannot confirm their presence in the high-resolution RFB data (as we have only examined averaged fields) we note that this product is capable of detecting polynyas that are missed by the SSMIS SIC product. Therefore, it is possible that some of these features are**

665     **consistent with RFB data, although we cannot assert this definitively. Since the RFB product is assimilated in the MULTIVAR experiment, its influence may also contribute to the emergence of these unrealistic polynyas in the model.**

**Modified sentence L.682: "***The multivariate experiment shows an even higher presence of open waters than the UNIVAR experiment during the peak boreal summer. The scale and duration of these features are not supported by the assimilated***

670     *SIC SSMIS observations and are likely artificial. The assimilated along-tracks RFB data is capable of detecting finer-scale polynyas that are not visible in the coarser SIC SSMIS product, but the direct link with the modelled polynya has not yet been established.***"**

    - L687 Indicate the inversion of ocean temperatures below the ice.

675     **Yes, thank you for this input. We change the sentence to L.687: "***As none of these openings occur in the FREE experiment, the thick snow and ice layer likely insulates the ocean from the atmosphere, maintaining the temperature inversion beneath the ice and limiting oceanic heat flux toward the ice base.***".**

    - L699-705 These assimilation settings are repeated from earlier methods section. It would be more logical to indicate first

680     there that these settings are different from the default in order to mitigate the appearance of polynyas.

**The description of the changes in SST and in situ assimilation has been updated accordingly to explain why we modified the original set up of the oceanic assimilation system. We then modified L.161 in paragraph 2.1.2: "***The ocean observations are not assimilated under the sea ice in the original operational system. Following experiments to set up the new ice assimilation system, instabilities in the water column appeared in the Southern Ocean. To reduce these static***

685     *instabilities, we activated the OSTIA SST assimilation under the ice to maintain the ocean temperature at the freezing point. We also stopped assimilating in situ data to the south of 60°S, regardless of the season, because the surface thermohaline properties were being durably modified on large spatial scales, despite the few profiles present. Assimilating these in situ data modified ocean stratification, causing upwellings of warm water at the surface and creating unrealistic open water areas within the sea ice cover.***"

690     **We removed the paragraph L.698-704: "***Changes have been implemented in the assimilation system to mitigate the occurrence of these simulated polynyas (see paragraphs 2.1.2 and 2.2.1). The SST assimilation under the ice has been activated to keep the surface waters close to the freezing point. Very few in situ profiles are available in the Southern Ocean, and some of them were radically changing the thermohaline properties of the ocean in a large area and over a long period of time, thus we did not activate the in-situ profile assimilation poleward 60°S to keep the modelled ocean stratification. We***

695     *increased the maximum SIC observation error to 40% to moderate the intensity of sea ice assimilation in the Southern hemisphere***".**

**We then modified the sentence L.704: "***The modifications in the assimilation scheme of the SST and in situ profiles described in section 2.1.2 have reduced the likelihood of triggering polynyas in both UNIVAR and MULTIVAR simulations but has not been able to prevent their occurrence.***".**

700

    - L744 CIMR should also include SMOS-like measurements of thin ice thickness.

**A sentence has been added in the conclusion to convey this information: "***CIMR will also provide thin ice estimates from L-band radiometry, similar to SMOS.***".**

705     - L796 Have the moorings data been used to calibrate the LEGOS product or are they strictly independent?

**To my knowledge, the BGEP moorings measurements have not been used to calibrate the LEGOS products.**

    - L811 Is an ice draft of 0 m synonymous of open water?

**Yes.**

710

**3 Typos and grammar**

The English can be improved by paying attention that verbs, adjectives and nouns are chosen carefully for clarity.

- L21-23: This sentence is too contorted, avoid the "seem" for something that should be certain and rephrase the last clause.

715 **The sentence has been modified to L.21-23: "***The multivariate system performs better in the Arctic than in Antarctica where the ice and ocean separate analyses are not designed to handle properly the strong interactions between upper oceanic layers and sea ice cover in the Southern Ocean.***"**

- L38 "The EnKF".

720 **We modified it in the newer version of the manuscript.**

- L62: Incomplete sentence, or remove "that".
**"that" is removed.**

725 - L79: Replace "stems" by stem (according to "sources") and remove the bold 's'.
**Yes, thank you.**

- L84: Times disagreement between "stated" and "recommend".
**We kept the past tense with "***recommended***" to stay coherent with the rest of the verbs.**

730
- L315 "higher presence of leads" -> "larger lead area"
**Yes, thank you.**

- L376 biase -> bias
735 **Yes, thank you.**

- L436 "mostly" here is misleading, I believe you mean "UNIVAR is underestimating [...] freeboard values the most".
**Yes, thank you.**

740 - Figure 5 caption has swapped the dotted and dashed lines.
**Yes thank you.**

- Figure 7 caption: use SIV instead of SIVOLU.
**Yes, thank you.**

745
- Figure 7 caption: SIC "either from the supplier or from OSI-SAF", for which product?
**For the AWI-CS2SMOS product, there is a SIC variable provided with each file and we use it in Figure 7. The LEGOS files do not provide a SIC variable, hence we use the OSISAF SSMIS SIC to compute sea ice volume. The sentence is modified to "***SIV is computed using either SIC data provided by the supplier (CS2SMOS SIV) or the SIC***
750 ***OSISAF SSMIS data (LEGOS SIV).***".**

- L495 "coherent" is slightly misleading, do you mean "consistent" or "similar"?
**Yes. Changed to "***consistent***".**

755 - L509 "fewer" should be "lower".

**Yes, thank you.**

- L552: "aims at keeping" -> "keeps"?
**Yes, thank you.**

760

- L581 "demarcation" is unclear: do you mean the edge of the observations domain?
**Yes, thank you.**

- L655 "a particularly pronounced discrepancy", is it much higher or much lower?
765 **The LEGOS ice density is much lower than the model ice density. See Figure RC2fig5 below: the model's ice density is n917 kg/m3; and the LEGOS ice density is 875 kg/m3 in October.**
**The sentence is modified to: "***The model's ice density (constant 917 kg/m³) exceeds that of the LEGOS_og observations (895 kg/m³ on average), with a particularly significant difference in October (LEGOS value: 875 kg/m³).***".**

[Figure]

770

**Figure RC2fig5: Sea ice and snow densities in Antarctica for the LEGOS products (circles grey Ls and dashed colored line) and for the model (horizontal colored lines).**

- L697 ... in places where the equilibrium of the model ...
775 **Yes, thank you.**

- L699, 700 Enumerate "First, ... Second, ..." for clarity.
**This paragraph has been removed following a previous comment.**

780 - L718 Iceat-2. Also capitalise ICESat-2 consistently throughout the paper.
**Yes, thank you.**

---

## Author Comment (AC2)

**Supplementary Materials:**
**Assimilation of radar freeboard and snow altimetry observations in the Arctic and Antarctic with a coupled ocean/sea ice modelling system**

5  Aliette Chenal[1,2], Gilles Garric[1], Charles-Emmanuel Testut[1], Mathieu Hamon[1], Giovanni Ruggiero[1], Florent Garnier[3], Pierre-Yves Le Traon[1,4]

1 Mercator Océan International, Toulouse, 31400, France
2 CNES, Toulouse, 31400, France
3 LEGOS, Toulouse, 31400, France
10  4 Ifremer, Plouzané, 29280, France

*Correspondence to:* Gilles Garric (ggarric@mercator-ocean.fr) and Aliette Chenal (aliette.chenal@yahoo.com)

**1 Additional figures**

15  ## 1.1 Sea ice concentration

[Figure]

**Figure S1: Sea ice concentration in March 2018 in the Arctic (a) and in the Antarctic (b) for observation SSMIS (first column), and the difference between the experiments (FREE, UNIVAR and MULTIVAR) and the reference SSMIS observation on the following columns. Root mean squared errors (RMS) are provided under each map.**

[Figure]

[Figure]

**Figure S2: Same as figure S1 for September 2018.**

**1.2 Radar freeboard and snow**

[Figure]

[Figure]

[Figure]

Figure S3: Top row panels: Probability density functions (%) of the snow thickness, the radar freeboard SAR and radar freeboard SARin observations (dotted black) and their model equivalent for the FREE (blue), UNIVAR (green) and MULTIVAR (pink) experiments in the Arctic for January 2017. Middle, resp. bottom, row panels: snow volume per unit area [m³/m2] , resp. radar freeboard volume per unit area, from SNOW-KaKu, resp. RFB LEGOS, (first column) and differences with FREE, UNIVAR and MULTIVAR experiments. Total snow and RFB volumes values and root mean squared difference (RMS) are provided under each map.

[Figure]

(a) November 2017 Arctic distributions

[Figure]

November 2017 Arctic snow volume differences relative to SNOW-KaKu

[Figure]

November 2017 Arctic radar freeboard volume differences relative to RFB LEGOS

**Figure S4: Same as Figure S3 for November 2017.**

**(a) March 2018 Arctic distributions**

[Figure]

March 2018 Arctic snow volume differences relative to SNOW-KaKu

[Figure]

March 2018 Arctic radar freeboard volume differences relative to RFB LEGOS

[Figure]

**Figure S5: Same as Figure S3 for March 2018.**

**(a) December 2018 Arctic distributions**

[Figure]

December 2018 Arctic snow volume differences relative to SNOW-KaKu

[Figure]

December 2018 Arctic radar freeboard volume differences relative to RFB LEGOS

[Figure]

35

**Figure S6: Same as Figure S3 for December 2018.**

**(a) May 2017 Antarctic distributions**

[Figure]

**May 2017 Antarctic snow volume differences relative to SNOW-KaKu**

[Figure]

**May 2017 Antarctic radar freeboard volume differences relative to RFB LEGOS**

[Figure]

**Figure S7: Same as Figure S3 for Antarctic and May 2017.**

(a) May 2018 Antarctic distributions

[Figure]

May 2018 Antarctic snow volume differences relative to SNOW-KaKu

[Figure]

May 2018 Antarctic radar freeboard volume differences relative to RFB LEGOS

[Figure]

40    **Figure S8: Same as Figure S3 for Antarctic and May 2018.**

**(a) October 2018 Antarctic distributions**

[Figure]

October 2018 Antarctic snow volume differences relative to SNOW-KaKu

[Figure]

**Figure S9: Same as Figure S3 for Antarctic and October 2018.**

**1.3 Sea ice volume: comparison to CS2SMOS and SMOS products**

[Figure]

| | LEGOS zone | | | | Outside LEGOS zone | | | |
| | SIT > 1 m | | SIT < 1 m | | SIT > 1 m | | SIT < 1 m | |
| | RMS | MD | RMS | MD | RMS | MD | RMS | MD |
|---|---|---|---|---|---|---|---|---|
| FREE | 0.29 | -0.18 | 0.21 | 0.17 | 0.59 | -0.57 | 0.19 | 0.16 |
| UNIVAR | 0.33 | -0.25 | 0.16 | 0.01 | 0.71 | -0.71 | 0.13 | -0.01 |
| MULTIVAR | 0.26 | 0.19 | 0.16 | 0.13 | 0.44 | -0.37 | 0.18 | 0.16 |
| LEGOS_og | 0.21 | 0.11 | 0.19 | 0.1 | - | - | - | - |
| LEGOS_mD | 0.42 | 0.39 | 0.23 | 0.16 | - | - | - | - |

45

**Figure S10: Sea ice volume in the Arctic in January 2017 for CS2SMOS dataset (reference) and its difference with the FREE, UNIVAR, and MULTIVAR experiments (first line) and the observations LEGOS_og (original) and LEGOS_mD (with model constant densities). Table: root mean square error (RMS) and mean difference (MD) between FREE, UNIVAR, MULTIVAR, LEGOS_og, LEGOS_md and CS2SMOS data, calculated on the LEGOS zone and outside the LEGOS zone and for CS2SMOS sea**

50 **ice thickness of less than or greater than 1m. The table colors highlight the values close to 0 (white) and the extremes (green for the RMS, and blue/red for the negative/positive MD).**

[Figure]

**Figure S11: Same as Figure S10 for November 2017.**

[Figure]

55

**Figure S12: Same as Figure S10 for March 2018.**

[Figure]

May 2017 Antarctic sea ice volume, comparison with SMOS product

| | LEGOS zone | | Outside LEGOS zone | |
|---|---|---|---|---|
| | RMS | MD | RMS | MD |
| FREE | 0.21 | -0.17 | 0.13 | -0.03 |
| UNIVAR | 0.23 | -0.2 | 0.11 | -0.07 |
| MULTIVAR | 0.19 | 0.0 | 0.13 | 0.03 |
| LEGOS_og | 0.42 | 0.35 | - | - |
| LEGOS_mD | 0.57 | 0.53 | - | - |

**Figure S13: Same as Figure S10 for May 2017 in Antarctic.**

[Figure]

May 2018 Antarctic sea ice volume, comparison with SMOS product

| | LEGOS zone | | Outside LEGOS zone | |
|---|---|---|---|---|
| | RMS | MD | RMS | MD |
| FREE | 0.22 | -0.1 | 0.05 | 0.0 |
| UNIVAR | 0.26 | -0.24 | 0.02 | -0.01 |
| MULTIVAR | 0.24 | 0.1 | 0.04 | 0.02 |
| LEGOS_og | 0.62 | 0.58 | - | - |
| LEGOS_mD | 0.84 | 0.81 | - | - |

60

**Figure S14: Same as Figure S10 for May 2018 in Antarctic.**

[Figure]

**Figure S15: Same as Figure S10 for September 2018 in Antarctic.**

**2 Monodata/multivariate experiment**

We performed a monodata/multivariate experiment assimilating the SIC OSISAF SSMIS product only with the multivariate assimilation system described previously. This experiment shows intermediate performances between monodata/univariate and multidata/multivariate for the sea ice concentration (Figure S16) and similar performance as monodata/univariate for the leads content (Figure S17). Similarly, this experiment provides intermediate performance for the RFB (Figures S18 and S19). The sea ice volume in this experiment increases the sea ice volume compared to the monodata/univariate simulation but is significantly less than the multidata/mutlivariate experiment and far from the LEGOS altimetric observations, in both hemispheres (Figure S22).

**2.1 Performance of the assimilation system**

[Figure]

[Figure]

75

**Figure S16: Same as Figure 1. in the paper. July 2018 in the Arctic (a) and September 2017 in the Antarctic (b) maps of the sea ice concentration, representing the observation SSMIS on the first column, and the difference between the experiments and the reference SSMIS observation on the following columns. The simulations are, in that order: FREE, UNIVAR, monodata/multivariate and MULTIVAR. Root mean squared errors (RMS) are provided under each map.**

[Figure]

80

**Figure S17: Same as Figure 2 in the paper. Daily time evolution of Arctic (a) and Antarctic (b) surface covered by sea ice leads in millions of km² for SSMIS (black), AMSR2 (dashed black), NSIDC (dotted black) satellite data with the surface range covered by them (shaded grey) and for FREE (blue), UNIVAR (green), monodata/multivariate (orange) and MULTIVAR (pink) experiments.**

(a) April 2017 Arctic distributions

(b) April 2017 Arctic snow volume differences relative to SNOW-KaKu

[Figure]

(c) April 2017 Arctic radar freeboard volume differences relative to RFB LEGOS

[Figure]

**Figure S18: Same as Figure 3 in the paper. Top panels (a): Probability density functions (%) of the snow thickness, the radar freeboard SAR and radar freeboard SARin observations (dotted black) and their model equivalent for the FREE (blue), UNIVAR (green), monodata/multivariate (orange) and MULTIVAR (pink) experiments in the Arctic for April 2017. Middle (b), resp. bottom (c), row panels: snow volume per unit area [m³/m2] , resp. radar freeboard volume per unit area, from SNOW-KaKu, resp. RFB LEGOS, (first column) and differences with FREE, UNIVAR, monodata/multivariate and MULTIVAR experiments. Total snow and RFB volumes values and root mean squared difference (RMS) are provided under each map.**

[Figure]

[Figure]

[Figure]

**Figure S19: Same as Figure 4 in the paper. Top panels (a): Probability density functions (%) of the snow thickness, the radar freeboard SAR and radar freeboard SARin observations (dotted black) and their model equivalent for the FREE (blue), UNIVAR (green), monodata/multivariate (orange) and MULTIVAR (pink) experiments in the Antarctic for May and October 2017. Middle (b), resp. bottom (c), row panels: snow volume per unit area, resp. radar freeboard volume per unit area, from SNOW-KaKu, resp. RFB LEGOS, (first column) and differences with FREE, UNIVAR, monodata/multivariate and MULTIVAR experiments in October 2017. Total snow and RFB volumes values and root mean squared difference (RMS) are provided under each map.**

95

**2.2 Validation with independent datasets**

[Figure]

**Figure S20: Same as Figure 5 in the paper. Scatterplots of the monthly Arctic ICESat-2 total freeboard against FREE, UNIVAR, monodata/multivariate, MULTIVAR experiments and LEGOS RFB/SND-KaKu data computed with model densities (black) for October 2018, beginning on the 14/10/2018 (experiments respectively in blue, green, orange and pink; no LEGOS data), and for January-February 2019 (experiments respectively in orange, red, grey and cyan). The x=y line (grey) and linear regressions for Oct 2018 (dashed black) and Jan-Feb 2019 (dotted black) are shown. Values of the linear slopes (s) and the r-values (r) are provided and all statistics are significant.**

[Figure]

**Figure S21: Same as Figure 6 in the paper. Idem Figure S20 but for Antarctica.**

[Figure]

110 **Figure S22: Same as Figure 7 in the paper. Time evolution of Arctic (a) and Antarctic (b) sea ice volume. The daily values are presented for the simulations FREE (blue), UNIVAR (green), monodata/mutivariate (orange) and MULTIVAR (pink), integrated over the whole h hemisphere (dotted) and over the observation domain (plain lines). SIV observations used for comparison are computed over the LEGOS observation domain: LEGOS original SIT (LEGOS_og, grey L in circles), SIT constructed from LEGOS observations of RFB and snow and the model constant ice and snow densities (LEGOS_mD, black stars), and CS2SMOS**
115 **AWI data in the Arctic (black dashes). The SIVOLU is computed using either SIC data provided by the supplier or the SIC OSISAF SSMIS data.**

April 2017 Arctic sea ice volume

[Figure]

|  | LEGOS zone | | | | Outside LEGOS zone | | | |
|---|---|---|---|---|---|---|---|---|
|  | SIT > 1 m | | SIT < 1 m | | SIT > 1 m | | SIT < 1 m | |
|  | RMS | MD | RMS | MD | RMS | MD | RMS | MD |
| FREE | 0.25 | -0.01 | 0.46 | 0.39 | 0.44 | -0.3 | 0.48 | 0.45 |
| UNIVAR | 0.32 | -0.26 | 0.28 | -0.0 | 0.65 | -0.62 | 0.16 | 0.02 |
| monodata multivariate | 0.27 | -0.11 | 0.26 | 0.11 | 0.51 | -0.35 | 0.23 | 0.16 |
| MULTIVAR | 0.31 | 0.22 | 0.36 | 0.31 | 0.38 | 0.05 | 0.27 | 0.22 |
| LEGOS_og | 0.27 | -0.12 | 0.35 | 0.32 | - | - | - | - |
| LEGOS_mD | 0.35 | 0.27 | 0.46 | 0.44 | - | - | - | - |

**Figure S23: Same as Figure 8 in the paper.** April 2017 sea ice volume in the Arctic for CS2SMOS dataset (reference) and its difference with the FREE, UNIVAR, monodata/multivariate and MULTIVAR experiments (first line) and the observations LEGOS_og (original) and LEGOS_mD (with model constant densities). Table: root mean square error (RMS) and mean difference (MD) between FREE, UNIVAR, MULTIVAR, LEGOS_og, LEGOS_md and CS2SMOS data, calculated on the LEGOS zone and outside the LEGOS zone and for CS2SMOS sea ice thickness of less than or greater than 1m. The table colours highlight the values close to 0 (white) and the extremes (green for the RMS, and blue/red for the negative/positive MD). The LEGOS zone corresponds to areas where the KaKu snow depth is available.

120

[Figure]

**September 2017 Antarctic sea ice volume, comparison with SMOS product**

|  | LEGOS zone | | Outside LEGOS zone | |
|---|---|---|---|---|
|  | RMS | MD | RMS | MD |
| FREE | 0.25 | 0.16 | 0.32 | 0.3 |
| UNIVAR | 0.26 | -0.2 | 0.11 | -0.03 |
| monodata/multivariate | 0.25 | -0.12 | 0.11 | -0.02 |
| MULTIVAR | 0.38 | 0.21 | 0.19 | 0.07 |
| LEGOS_og | 0.78 | 0.73 | - | - |
| LEGOS_mD | 0.98 | 0.94 | - | - |

**Figure S24: Same as Figure 9 in the paper. September 2017 sea ice volume maps in the Antarctic for the SMOS data (reference) and its difference to the FREE, UNIVAR, monodata/multivariate and MULTIVAR experiments (first line) and to the observations LEGOS_og (original) and LEGOS_mD (with model constant densities). The colorbar shows only which only measures the ice that is thinner than 1 m (thicker ice is represented in yellow). Table: root mean square error (RMS) and mean difference (MD) between FREE, UNIVAR, monodata/multivariate and MULTIVAR, LEGOS_og, LEGOS_mD and SMOS data, calculated on the LEGOS zone and outside the LEGOS zone. The table colours highlight the values close to 0 (white) and the extremes (green for the RMS, and blue/red for the negative/positive MD). The LEGOS zone corresponds to areas where the KaKu snow depth is available.**

---

## Author Comment (AC3)

**RC1**

**1 General comments**

**We would like to thank the reviewers for their careful reading of the manuscript and the helpful comments. The answers to the comments are written in bold, the reviewer's comment in normal font, and the sentences we propose to add to the manuscript are in italics.**

The research article "Assimilation of radar freeboard and snow altimetry observations in the Arctic and Antarctic with a coupled ocean/sea ice modelling system" introduces a multivariate assimilation system using LEGOS radar freeboard and KaKu altimetry snow depth in addition to sea ice concentration observations and compares the performance of this novel model run to a free model run and one using only sea ice concentration observations. The paper is well structured, provides a good literature overview and clear motivation for the work. Many comparisons against independent validation data are presented and the results are discussed appropriately. Furthermore, the figures are neat, and the results prove a clear overall improvement compared to the other model runs. Therefore, I recommend this article for publication after minor revisions: General comments:

1. As most plots are only for one specific month, it would be nice to have the same plots for all months in the appendix or to publish them as supplementary dataset.

   Moreover, there are several instances, where other months are discussed with a comment "not shown", making it hard to follow.

   I also wonder how you chose the months for each plot, as the choices are not consistent (Fig 2 shows July 2017 and September 2017, Fig 3 and 8 show April 2017 for the Arctic, Fig 4 shows some plots for May in addition to October 2017 for the Antarctic, but Fig 9 shows September 2017 instead of October and Figure 5 and 6 show October 2018 and Jan-Feb 2019). There might be good reasons for the choices, but to prove that the plots weren't cherry-picked to support the author's arguments most, all plots should be published somewhere alongside the manuscript.

**The plots for each month and for all variables are made available in the attached document 1_Figures_review_for_RC1.**

**However, we think all the figures would provide redundant information in the manuscript. We propose adding a few additional figures as supplementary material attached to the article (file: 2_Supplementary_Materials) to illustrate the most important results found during two seasonal cycles and also to show that the results shown in the manuscript remain robust over the 2 seasonal cycles:**
- **March and September 2018 for SIC highlighting the maximum of the extension for the SIC Arctic and Antarctic and the minimum in Arctic and close-to-minimum extension in Antarctic**
- **Snow thickness (SNT) and radar freeboard (RFB) in the Arctic: a) January 2017 to show the first month of the experiments, the biases that were already corrected and the ones remaining during this spin up b) November 2017 to show the situation after the first summer with no observations, and how and where the system has kept the memory of the RFB and SNT distributions c) March 2018 to show the variables after more than one year of assimilation and d) December 2018 to show the agreement between MULTIVAR and SNOW-KaKu measurements**
- **Snow thickness (SNT) and radar freeboard (RFB) in Antarctic: a) May 2017 to show the first month of presence of observations and their assimilation in the SH b) May 2018 to show the variables after a full seasonal cycle and the restart of data assimilation in the southern hemisphere c) October 2018 to show that**

45          the statements made are true and constant until the end of the periods of presence of assimilated
            observations.
      -     Sea ice volume (SIV) in the Arctic and comparison with CS2SMOS: same months as the SNT & RFB,
            January 2017, November 2017 and March 2018.
      -     SIV in the Antarctic and comparison with SMOS: same months as SNT & RFB, May 2017, May 2018 and
50          September 2018.
For the figures not shown, we have referred to new figures in the supplementary material. We have modified the
manuscript accordingly:
      -     L. 260: In both hemispheres, FREE is not able to prevent excessive melting and shows a significant lack of sea ice,
            mainly in marginal areas, during July-October in the Arctic and in January-April in Antarctica (See Figures S1
55          and S2).
      -     L.309: In Antarctica, the FREE simulation presents mainly positive SIC biases in winter, particularly in the
            marginal ice zone (MIZ, defined by SIC values between 15% and 80%), and places the ice edge too far north
            compared to SSMIS observations (Figures 2 and S2).
      -     L. 324: The MULTIVAR snow distribution is very close to the Arctic SNOW-KaKu during winter (Figures S3
60          and S5) and matches perfectly in April
      -     L.327: The linear correlation (r-value) computed against the SNOW-KaKu observations in the Arctic results is
            consistently above 0.5 for MULTIVAR, peaking at 0.7 in December 2018 (Figure S6 not shown)
      -     L. 374: After re-checking, the FREE experiment does not systematically exhibit lower values, we then removed
            the following sentence in the manuscript: "The FREE simulation exhibits lower RFB values than the other
65          experiments especially at the end of summer (not shown)."
Choosing which months to show is tricky because the periods observed remain discontinuous, differing from one
hemisphere to another, and we cannot insert many figures. A few explanations about the inconsistency of the dates
chosen:
      -     Fig. 2 shows the sea ice concentration, so the months shown were chosen to present the most relevant months
70          regarding the evolution of sea ice leads content (Fig. 1). July 2017 corresponds to the Arctic maximum sea ice
            leads area; whereas September 2017 in Antarctica shows the strong presence of leads (Figure 1-b) in the ice
            cover of MULTIVAR and UNIVAR (Figure 1 b) and the Weddell polynya appears in the observations.
      -     The other maps (Fig. 3, 4, and 8) show the variables during the last month of assimilation of the altimetric
            observations (snow and radar freeboard), i.e. April in the northern hemisphere and October in the southern
75          hemisphere.
      -     The same should have been done for Fig. 9, but the SMOS measurements are not available after the 15th of
            October, so the month of September has been chosen instead.
      -     The PDF in Fig. 4 shows the data in October 2017 but also in May 2017 to show the evolution of the variables
            in Antarctica from the beginning to the end of the winter.
80

            2.  The appendix adds a valuable comparison to completely independent in situ data and I wonder why you
                did not include these in the main text.
We wanted to keep the main manuscript as short as possible, given that it already is a long article. That is why the
comparison with in-situ measurements was put in the appendix. We agree that this comparison is valuable and
85 relevant; we leave it up to the reviewer and editor to decide whether to add this to the body of the text.
If so, we suggest:
      -     adding this comparison with in situ as a subsection 4.2 entitled "Comparison with in-situ measurements" of
            the part 4 "Validation with independent datasets". We have attached a separate document containing this
            subsection 4.2 for the review (file: 3_section4.2 Comparison with in situ measurements). The figures within
90          this subsection have been slightly modified to differentiate statistics by season. Further, we have added in this
            subsection the tables describing the root mean square error and mean differences, see comment below.
      -     adding this paragraph in the discussion part: L. 607: "*Further comparison with in-situ independent observations
            in the Arctic only show general improvement with the multivariate assimilation system compared to the FREE and
            the UNIVAR experiments. The MULTIVAR experiment is able to maintain the remarkable agreement found with the*

95    *FREE experiment with ULS moorings in the Beaufort Sea and favorably thickens all types of ice in the Fram Strait region. At the same time, the multivariate approach also positively increases the thickest ice even in the absence of snow data. Comparisons during the summer season show no particular deterioration or improvement with the multivariate system*"

- **modifying the text in the conclusion part as follows: L. 718 "***Moreover, the diagnosed freeboard from the multivariate system compares better with Iceat-2 and in-situ independent observations in the Arctic and, to a lesser extent, in Antarctica.*"

It would also be helpful to add RMSE values to these scatter plots and discussion.

**Yes, we agree. The RMSEs values between simulation and the three in-situ datasets are summarized in the tables below. The discussions and references to the Tables are added in sub-section 4.2 mentioned in the previous response.**

| BGEP ULS DATA | RMSE total | MD total | RMSE winter | MD winter | RMSE summer | MD summer |
|---|---|---|---|---|---|---|
| *LEGOS* | | | *0.194* | *0.113* | | |
| *FREE* | *0.134* | *0.011* | *0.121* | *0.095* | *0.150* | *-0.087* |
| *UNIVAR* | *0.139* | *-0.038* | *0.141* | *-0.020* | *0.137* | *-0.058* |
| *MULTIVAR* | *0.191* | *0.068* | *0.182* | *0.160* | *0.202* | *-0.039* |

*Table 4 : Root mean square error (RMSE) and mean differences (MD) between the BGEP ULS measurements and LEGOS data (only winter months : November to April), FREE, UNIVAR and MULTIVAR experiments, by season (summer: May to October, and winter) and over the two seasons as a total.*

| NPI ULS DATA | RMSE total | MD total | RMSE winter | MD winter | RMSE summer | MD summer |
|---|---|---|---|---|---|---|
| *LEGOS* | | | *0.427* | *0.366* | | |
| *FREE* | *1.040* | *-1.040* | *0.696* | *-0.696* | *1.402* | *-1.402* |
| *UNIVAR* | *1.238* | *-1.238* | *1.02* | *-1.029* | *1.458* | *-1.458* |
| *MULTIVAR* | *0.645* | *-0.571* | *0.316* | *-0.189* | *0.991* | *-0.972* |

*Table 5: Same as Table 4 with the NPI ULS measurements.*

| OiB AIRBORNE DATA | RMSE totale | MD totale | RMSE lat<81.5°N | MD lat<81.5°N | RMSE lat>81.5°N | MD lat>81.5°N |
|---|---|---|---|---|---|---|
| *LEGOS* | | | *0.449* | *0.068* | | |
| *FREE* | *0.639* | *-0.503* | *0.459* | *-0.200* | *0.744* | *-0.681* |
| *UNIVAR* | *0.869* | *-0.794* | *0.574* | *-0.416* | *1.042* | *-1.016* |
| *MULTIVAR* | *0.652* | *0.182* | *0.486* | *0.135* | *0.750* | *0.209* |

110 *Table 6: Same as Table 4 with the OiB Airborne data and, according to the areas where SNOW-KaKu data is present (<81,5°N) or not (> 81,5°N) and for all OiB Airborne data.*

3.  Throughout the paper, English grammar is not always used correctly, but the text is generally understandable. I, therefore, trust that copy editing will deal with this.

115   **Thank you. We apologize for the English mistakes, we do our best to correct the text.**

**2 Specific comments**

L.110: SSMIS is not explained. Here, I would maybe just talk about SIC.

120   **We have modified the text accordingly, L110:** "*What are the impacts of using altimetric radar freeboard and altimetric snow observations in addition to the SIC data?*"

L.200 onwards: In the methods section, SSMIS should be explained and mentioned in the text, also to make the difference to the OSISAF AMSR2 product, which is used later on, clearer.

125 **The sentence has been modified to :** "*The observation data used for sea ice concentration (SIC) assimilation is the global daily reprocessed passive microwave dataset, measured with Special Sensor Microwave Imager / Sounder (SSMIS) satellites instruments, from the European Organization for the Exploitation of Meteorological Satellites (EUMETSAT) Ocean and Sea Ice Satellite Application Facility (OSISAF) OSI-450 (OSI SAF, 2022).*"

130 L.207: How was the 40% value for Antarctica chosen?
**We increased the observation error in Antarctica to avoid moving too far away from the free model solution. Increasing the observation error reduces the impact of the assimilation on the sensitive coupling between ocean and ice in the Southern Ocean . The value of 40% has been however chosen arbitrarily.**

135 L.213: 'measure' rather than 'detect'
**Thank you.**

L.225 and 230: Which months are counted as winter? Please specify.
**A table (see below) has been added to specify the gaps in the assimilated observations and commented in subsections**
140 **2.2.1, 2.2.2 and 2.2.3. See answers to the reviewer's comments 2.**
**L. 199:**

| Observations | SIC SSMIS | RFB-LEGOS | SNOW-KaKu |
|---|---|---|---|
| Producer | EUMETSAT OSI-SAF | LEGOS | LEGOS |
| Temporal resolution | Daily | 20 Hz | Monthly → weekly (linear interpolation) |
| Temporal coverage | All-time | Winter: November to April in the Arctic; May to October in the Antarctic. | |
| Spatial resolution | 40 km (effective resolution); 25 km (grid resolution). | Along-tracks | 12.5 km (grid resolution). |
| Spatial gaps | None (reprocessed). | Central Arctic (latitude > 88°N); in-between satellite tracks. | Central Arctic (latitude > 81.5°N) ; coastal areas. |
| nsel | 400 | 4000 | 400 |

Table 2: Assimilated observation products and their specificities.

L.226: Do you average all altimetry observations within a model grid cell or within a radius from the grid cell? Any
145 weighting? Please specify.
**We use a gaussian weight for the observations. This is done for all the observations in the localization algorithm of the assimilation system (see sentence L. 150: "***The increments at each model grid point are calculated independently in a local scheme, where a localization algorithm controls the spatial influence of observations. This approach helps to limit the impact of sampling noise on the increments.***"). The radius of the localization scheme is the minimum between a fixed**
150 **distance of 176 km and a radius defined by the inclusion of a number 'nsel' of observations. This number 'nsel' is set to 400 for gridded products such as the SIC and snow depth data and to 4000 for the high resolution radar freeboard data. The following sentence has been added L. 151** "*The radius of the localization scheme is set as the minimum between an arbitrary fixed distance of 176 km and a radius defined by the inclusion of a number of observation nsel (see the chosen nsel values in Table 2).*"

155
L.232: Why do you scale the uncertainties and how do you decide on this range?
**The observation errors for the snow and the radar freeboard are not modified from the ones given by producers. We set a maximum/minimum to these observation errors to make sure no observation error would be erroneous.**

**Especially, the numerical computation could become impossible if a 0 value exists, then we set the minimum error to**
160 **0.01 m. The maximum value of 5m is chosen arbitrarily and to be large enough to exclude outliers.**

L.245: How were these dates chosen?
**The assimilation cycle lasts 7 days and begins on a Wednesday to match the design of the weekly analysis cycles of Mercator's operational systems (real-time or reanalysis systems). The first day of the analysis was chosen arbitrarily.**
165 **The period 2017-2018 was selected to capture interesting features of the sea ice such as the Weddell Polynya in September 2017 and the Greenland polynya in February 2018. We ran the experiment for 2 years to have 2 complete seasonal cycles and also to have the ICESat data available at the beginning of 2019.**

L.252: referred to 'as' leads
170 **Thank you.**

L.254: I would call it 'lead fraction' rather than 'lead content'
**Thank you.**

175 L.257: explain CDR
**Thank you. L.256: "***... and the Climate Data Record (CDR) dataset …***"**

Figure 1 caption: Do you mean 'range' rather than 'surface' covered by them?
**Thank you. Figure 1 caption: "***Daily time evolution of Arctic (a) and Antarctic (b) surface covered by sea ice leads in***
180 ***millions of km² for SSMIS (black), AMSR2 (dashed black), NSIDC (dotted black) satellite data with the range covered by them (shaded grey) and for FREE (blue), UNIVAR (green) and MULTIVAR (pink) experiments.***"**

L.259-262 and L.285-314 + Fig 2: I suggest a separate section for SIC e.g. before the lead section. Especially the first paragraph on sea ice concentration (L259-262) currently sits between two paragraphs on lead fraction and interrupts the
185 flow.
**This actually makes sense, we've moved the 2 paragraphs concerned and we modified the text accordingly. We separated the two ways of evaluating the modelled sea ice concentration: a first paragraph evaluating the spatial distribution of SIC in both hemispheres, and a second one evaluating the area of leads in sea ice. We've modified the title of the subsection 3.1 L. 250: "***Sea ice concentration and sea ice leads***". We have inverted the figure numbers, with**
190 **figure 2 becoming the figure 1 and vice versa. We have attached a separate document containing this subsection 3.1 for the review (file: 4_section3.1 Sea ice concentration and sea ice leads).**

Figure 2: I would stick to SIC rather than SICONC in the titles and colourmap legend;
**Thank you. The figure has been modified accordingly.**
195

In the caption 'experiences' should be 'experiments'?!
**Thank you.**

L.318: ..but there are more unobserved polynyas for MULTIVAR according to Fig.2 ?!
200 **We mentioned this result in the conclusion and we can indeed mention it in this section and refer to a figure in the supplementary materials. L.318: "***In the Antarctic, the two assimilated experiments generate variability and occurrence of unobserved polynyas, but MULTIVAR creates them more frequently all around Antarctica (Figure S2).***"**

Figure 3 and 4: Shouldn't the RMS have the same unit as SNV and RFBV?
205 **No, because the values for the SNV and RFBV are the variables integrated over the whole iced area, to be able to discuss a 3D volume. The RMS is computed on each grid cell, with the variables expressed as SNV and RFBV averaged over the cell's iced area, meaning values in [m^3]/[m^2], following the model's definition of these variables.**

I really like the distribution plots in a). Maybe these could also be added to the plots for SIC (Fig.2), total fb (Fig. 5, 6) and SIV (Fig. 8,9)?

**Thank you for this comment. Indeed, the PDF nicely provides a synthetic evaluation. We don't think they are as relevant for SIC and this study does not aim at presenting a detailed validation of SIC. We did not conduct a detailed analysis of the total FB and SIV distributions for comparisons with IceSat-2 and SMOS because the article is already too long and focuses on showcasing the potential of the multivariate/multi-data approach.**

Figure 4: It would be nice to see the full first peak around 0. Maybe add an inset with a higher y axis.

**The 0 peak goes up to 25% and above. We chose to limit the view in the y-axis to be able to see the second mode of this PDF.**

[Figure]

**Figure RC1fig1: Panels (a): Probability density functions (%) of the snow thickness, the radar freeboard SAR and radar freeboard SARin observations (dotted black) and their model equivalent for the FREE (blue), UNIVAR (green) and MULTIVAR (pink) experiments in the Antarctic for May and October 2017. The full extent of the 0-mode is shown for the snow thickness PDFs.**

L.354: FREE diverges the most, but also matches best with the observations in May

**That is true. The sentence has been modified to "***Among the simulations, the FREE experiment matches better the observations in May 2017 but then diverges the most from the observations, showing an increasing accumulation of snow as winter progresses, with a main mode 11.2 cm higher than the observed mode in October 2017***".**

L.460: 'excludes' rather than 'includes'?
**Thank you.**

L.472: Stick to SIV instead of SIVOLU (I think this is what you mean?!)
**Thank you.**

L.486: I am missing a paragraph on the Antarctica plot in Figure 7b and specifically also a comment/explanation why the timings of sea ice volume decrease are offset between observations and model. In 2018, the observations clearly drop between September and October, whereas the models are still increasing.

**The paragraph describing the sea ice volume in Antarctica has been added. Thank you for noticing. The added paragraph starts now L.487: "***As in the Arctic, MULTIVAR has the highest freezing rate and the highest total sea ice volume in Antarctica among the experiments for the most part of the simulation periods (Figure 7(b)), with, on average, 25% and 141% higher ice volume than FREE and UNIVAR estimates respectively. UNIVAR consistently presents the lowest ice***

*volume. The assimilated experiments have irregular time series during the second half of the growing season, the MULTIVAR simulation especially collapses many times before reaching its peak. These collapses coincide between the two*
245 *assimilated experiments and are also present in the observation space (solid lines, Figure 7(b)). These sudden ice volume losses are due to the occurrence of large open waters or polynias within the sea ice cover which first and foremost causes an increase of sea ice leads from July to September 2017 and in August and September 2018 (Figure 1(b)). Some of them also appear in the observation products such as the well-known Maud-rise polynya in the Weddell Sea in 2017.*

*The use of the model constant densities (LEGOS_mD) results in higher SIV estimates than the LEGOS_og product using*
250 *seasonally varying ice and snow densities to convert RFB into ice thickness (Figure 7 (b)). The deviation between these two datasets is maximum in October because of the significant drop in ice density from 900 kg.m-3 to 875 kg.m-3 between September and October. With one exception (October 2018), both LEGOS_og and LEGOS_mD observations present systemically higher SIV values than MULTIVAR simulation. And even if the MULTIVAR experiment remains the closest experiment to the LEGOS observations, it is still up to 10 million km3 below the LEGOS_mD estimates. Over both 2017 and*
255 *2018 winters, the datasets present mean SIV of respectively 4.6, 8.0, 10.8, 15.2 and 18.5 million km3 for the UNIVAR, FREE and MULTIVAR simulations, and the LEGOS_og and LEGOS_mD products. The LEGOS_og product displays a sea ice maximum in September, a month earlier than the FREE simulation. LEGOS_mD also has a SIV maximum in September for 2018 winter only, but the differences in densities make it unclear to identify the exact peak period in 2017. Similarly, the occurrence of polynias in assimilated experiments makes it impossible to accurately determine the maximum period.*"

260

Figure 8: I find greener colours in the table to mark worse results counterintuitive and would suggest using another colour like yellow.
**We agree. Instead of yellow, for which a color gradient is not so visual, we changed the green colormap to a purple one. (See Figures 8 and 9 below)**

265

[Figure]

*Figure 8: April 2017 sea ice volume in the Arctic for CS2SMOS dataset (reference) and its difference with the FREE, UNIVAR, and MULTIVAR experiments (first line) and the observations LEGOS_og (original) and LEGOS_mD (with model constant densities).*
270 *Table: root mean square error (RMS) and mean difference (MD) between FREE, UNIVAR, MULTIVAR, LEGOS_og, LEGOS_mD and CS2SMOS data, calculated on the LEGOS zone and outside the LEGOS zone and for CS2SMOS sea ice thickness of less than or greater than 1m. The table colours highlight the values close to 0 (white) and the extremes (green for the RMS, and blue/red for the negative/positive MD). The LEGOS zone corresponds to areas where the KaKu snow depth is available.*

[Figure]

275

*Figure 9: September 2017 sea ice volume in the Antarctic for the SMOS data (reference) and its difference to the FREE, UNIVAR, and MULTIVAR experiments (first line) and to the observations LEGOS_og (original) and LEGOS_mD (with model constant densities). The colorbar shows only which only measures the ice that is thinner than 1 m (thicker ice is represented in yellow). Table: root mean square error (RMS) and mean difference (MD) between FREE, UNIVAR, MULTIVAR, LEGOS_og, LEGOS_mD and SMOS data, calculated on the LEGOS zone and outside the LEGOS zone. The table colours highlight the values close to 0 (white) and the extremes (green for the RMS, and blue/red for the negative/positive MD). The LEGOS zone corresponds to areas where the KaKu snow depth is available.*

Figure 9: Explain the white areas in the figure caption

**We changed the figure to have the ice area for thicknesses above 1 m in yellow (see Figure 9 above). Now, the white areas are only related to the free ocean surfaces. We added the precision** "*The colorbar shows only the ice that is thinner than 1 m (thicker ice is represented in yellow),*" **in the caption.**

L. 601: You say most of the analysis in Antarctica was done in summer when no data is assimilated, however, most plots for the Antarctic are shifted by 6 months compared to the Arctic and if not, why don't you show those plots to make it a fairer comparison? Ideally, as mentioned above, all plots should be available to the reader anyway.

**The issue we encounter with the comparison with ICESat-2 data is that they are available from the 15th October 2018. Then, only 15 days of altimetric observations assimilation are available in Antarctica for that season. We stopped the simulations in March 2019, so there is no other period where both ICESat-2 measurements and the altimetric observations are available. We don't provide maps of differences with ICESAT-2 in the manuscrit, we don't understand which plots you are referring to? On Figure 6, the period January-February 2018 is shown in Antarctica and presents results in summer, with a comparison between our simulations' results and IceSat-2 measurements.**

L.678: Explain VP and EVP

**Thank you. VP stands for Viscous-Plastic, EVP for Elastic-Viscous-Plastic. The modification has been done in the manuscript: L.578 "***Sea-ice models using Viscous-Plastic or Elastic-Viscous-Plastic rheologies***".**

L.742: CRISTAL will also have a higher inclination orbit and hence provide these measurements with a much smaller hole (data gap) around the poles.

**Thank you. A sentence has been added to the manuscript to convey this information L.741: "***Moreover, a higher inclination orbit will enable measurements with a smaller hole around the North pole with the CRISTAL satellite.***"**

L.744: CIMR will also provide thin ice estimates like SMOS from L-band radiometry.

**Thank you. A sentence has been added in the conclusion to convey this information: "***CIMR will also provide thin ice estimates from L-band radiometry, similar to SMOS.***".**

---

## Author Comment (AC4)

**Figures for RC1**

**Monthly Sea ice concentration**

       The following figures, similar to Figure 2 in the manuscript, display monthly means of sea ice concentration (%) from January 2017 to January 2019 for Arctic and Antarctic. SSMIS observations are shown in the first column. Differences between the experiments (FREE, UNIVAR, MULTIVAR) and SSMIS are shown from the second to the fourth column. Root means squared errors (RMS) are provided under each plot.

[Figure]

[Figure]

(a) March 2017 Arctic SIC differences relative to OSISAF SSMIS

(b) March 2017 Antarctic SIC differences relative to OSISAF SSMIS

(a) April 2017 Arctic SIC differences relative to OSISAF SSMIS

(b) April 2017 Antarctic SIC differences relative to OSISAF SSMIS

(a) Mai 2017 Arctic SIC differences relative to OSISAF SSMIS

(b) Mai 2017 Antarctic SIC differences relative to OSISAF SSMIS

(a) June 2017 Arctic SIC differences relative to OSISAF SSMIS

(b) June 2017 Antarctic SIC differences relative to OSISAF SSMIS

(a) July 2017 Arctic SIC differences relative to OSISAF SSMIS

(b) July 2017 Antarctic SIC differences relative to OSISAF SSMIS

(a) August 2017 Arctic SIC differences relative to OSISAF SSMIS

(b) August 2017 Antarctic SIC differences relative to OSISAF SSMIS

(a) September 2017 Arctic SIC differences relative to OSISAF SSMIS

(b) September 2017 Antarctic SIC differences relative to OSISAF SSMIS

(a) October 2017 Arctic SIC differences relative to OSISAF SSMIS

(b) October 2017 Antarctic SIC differences relative to OSISAF SSMIS

(a) November 2017 Arctic SIC differences relative to OSISAF SSMIS

(b) November 2017 Antarctic SIC differences relative to OSISAF SSMIS

(a) December 2017 Arctic SIC differences relative to OSISAF SSMIS

(b) December 2017 Antarctic SIC differences relative to OSISAF SSMIS

(a) March 2018 Arctic SIC differences relative to OSISAF SSMIS

(b) March 2018 Antarctic SIC differences relative to OSISAF SSMIS

(a) April 2018 Arctic SIC differences relative to OSISAF SSMIS

(b) April 2018 Antarctic SIC differences relative to OSISAF SSMIS

(a) May 2018 Arctic SIC differences relative to OSISAF SSMIS

(b) May 2018 Antarctic SIC differences relative to OSISAF SSMIS

(a) June 2018 Arctic SIC differences relative to OSISAF SSMIS

(b) June 2018 Antarctic SIC differences relative to OSISAF SSMIS

**(a) July 2018 Arctic SIC differences relative to OSISAF SSMIS**

[Figure]

**(b) July 2018 Antarctic SIC differences relative to OSISAF SSMIS**

**(a) August 2018 Arctic SIC differences relative to OSISAF SSMIS**

**(b) August 2018 Antarctic SIC differences relative to OSISAF SSMIS**

[Figure]

(a) September 2018 Arctic SIC differences relative to OSISAF SSMIS

(b) September 2018 Antarctic SIC differences relative to OSISAF SSMIS

(a) October 2018 Arctic SIC differences relative to OSISAF SSMIS

(b) October 2018 Antarctic SIC differences relative to OSISAF SSMIS

(a) November 2018 Arctic SIC differences relative to OSISAF SSMIS

(b) November 2018 Antarctic SIC differences relative to OSISAF SSMIS

(a) December 2018 Arctic SIC differences relative to OSISAF SSMIS

(b) December 2018 Antarctic SIC differences relative to OSISAF SSMIS

(a) January 2019 Arctic SIC differences relative to OSISAF SSMIS

(b) January 2019 Antarctic SIC differences relative to OSISAF SSMIS

**Radar Freeboard, snow thickness and probability density functions in the Arctic.**

The following figures, similar to Figure 3 in the manuscript, display radar freeboard, snow thickness and probability functions in the Arctic for the periods January 2017-April 2017, November 2017-April 2108 and November 2018-January 2019. Top panels (a): Probability density functions (%) of the snow thickness, the radar freeboard SAR and radar freeboard SARin observations (dotted black) and their model equivalent for the FREE (blue), UNIVAR (green) and MULTIVAR (pink) experiments in the Arctic. Middle (b), resp. bottom (c), row panels: snow volume per unit area [m³/m2] , resp. radar freeboard volume per unit area, from SNOW-KaKu, resp. RFB LEGOS, (first column) and differences with FREE, UNIVAR and MULTIVAR experiments. Total snow and RFB volumes values and root mean squared difference (RMS) are provided under each map.

(a) January 2017 Arctic distributions

[Figure]

January 2017 Arctic snow volume differences relative to SNOW-KaKu

[Figure]

January 2017 Arctic radar freeboard volume differences relative to RFB LEGOS

(a) February 2017 Arctic distributions

[Figure]

February 2017 Arctic snow volume differences relative to SNOW-KaKu

[Figure]

February 2017 Arctic radar freeboard volume differences relative to RFB LEGOS

[Figure]

**(a) March 2017 Arctic distributions**

[Figure]

March 2017 Arctic snow volume differences relative to SNOW-KaKu

[Figure]

March 2017 Arctic radar freeboard volume differences relative to RFB LEGOS

[Figure]

**(a) April 2017 Arctic distributions**

[Figure]

**April 2017 Arctic snow volume differences relative to SNOW-KaKu**

[Figure]

**April 2017 Arctic radar freeboard volume differences relative to RFB LEGOS**

[Figure]

**(a) November 2017 Arctic distributions**

[Figure]

**November 2017 Arctic snow volume differences relative to SNOW-KaKu**

[Figure]

**November 2017 Arctic radar freeboard volume differences relative to RFB LEGOS**

[Figure]

**(a) December 2017 Arctic distributions**

[Figure]

**December 2017 Arctic snow volume differences relative to SNOW-KaKu**

[Figure]

**December 2017 Arctic radar freeboard volume differences relative to RFB LEGOS**

[Figure]

(a) January 2018 Arctic distributions

[Figure]

January 2018 Arctic snow volume differences relative to SNOW-KaKu

[Figure]

January 2018 Arctic radar freeboard volume differences relative to RFB LEGOS

[Figure]

**(a) February 2018 Arctic distributions**

[Figure]

February 2018 Arctic snow volume differences relative to SNOW-KaKu

[Figure]

February 2018 Arctic radar freeboard volume differences relative to RFB LEGOS

(a) March 2018 Arctic distributions

[Figure]

March 2018 Arctic snow volume differences relative to SNOW-KaKu

[Figure]

March 2018 Arctic radar freeboard volume differences relative to RFB LEGOS

[Figure]

**(a) April 2018 Arctic distributions**

[Figure]

**April 2018 Arctic snow volume differences relative to SNOW-KaKu**

[Figure]

**April 2018 Arctic radar freeboard volume differences relative to RFB LEGOS**

[Figure]

**(a) November 2018 Arctic distributions**

[Figure]

November 2018 Arctic snow volume differences relative to SNOW-KaKu

[Figure]

November 2018 Arctic radar freeboard volume differences relative to RFB LEGOS

[Figure]

**(a) December 2018 Arctic distributions**

[Figure]

December 2018 Arctic snow volume differences relative to SNOW-KaKu

[Figure]

December 2018 Arctic radar freeboard volume differences relative to RFB LEGOS

[Figure]

**(a) January 2019 Arctic distributions**

[Figure]

**January 2019 Arctic snow volume differences relative to SNOW-KaKu**

[Figure]

**January 2019 Arctic radar freeboard volume differences relative to RFB LEGOS**

[Figure]

**(a) February 2019 Arctic distributions**

[Figure]

**February 2019 Arctic snow volume differences relative to SNOW-KaKu**

[Figure]

**February 2019 Arctic radar freeboard volume differences relative to RFB LEGOS**

[Figure]

**Radar Freeboard, snow thickness and probability density function in the Antarctic.**

The following figures, similar to Figure 4 in the manuscript, display radar freeboard, snow thickness and probability functions in the Antarctic for the periods May 2017-October 2017 and May 2018-October 2018. Top panels (a): Probability density functions (%) of the snow thickness, the radar freeboard SAR and radar freeboard SARin observations (dotted black) and their model equivalent for the FREE (blue), UNIVAR (green) and MULTIVAR (pink) experiments in the Arctic. Middle (b), resp. bottom (c), row panels: snow volume per unit area [m³/m2] , resp. radar freeboard volume per unit area, from SNOW-KaKu, resp. RFB LEGOS, (first column) and differences with FREE, UNIVAR and MULTIVAR experiments. Total snow and RFB volumes values and root mean squared difference (RMS) are provided under each map.

**(a) May 2017 Antarctic distributions**

[Figure]

**May 2017 Antarctic snow volume differences relative to SNOW-KaKu**

[Figure]

**May 2017 Antarctic radar freeboard volume differences relative to RFB LEGOS**

[Figure]

(a) June 2017 Antarctic distributions

[Figure]

June 2017 Antarctic snow volume differences relative to SNOW-KaKu

[Figure]

June 2017 Antarctic radar freeboard volume differences relative to RFB LEGOS

[Figure]

**(a) July 2017 Antarctic distributions**

[Figure]

**July 2017 Antarctic snow volume differences relative to SNOW-KaKu**

[Figure]

**July 2017 Antarctic radar freeboard volume differences relative to RFB LEGOS**

[Figure]

**(a) August 2017 Antarctic distributions**

[Figure]

August 2017 Antarctic snow volume differences relative to SNOW-KaKu

[Figure]

August 2017 Antarctic radar freeboard volume differences relative to RFB LEGOS

[Figure]

**(a) September 2017 Antarctic distributions**

[Figure]

September 2017 Antarctic snow volume differences relative to SNOW-KaKu

[Figure]

September 2017 Antarctic radar freeboard volume differences relative to RFB LEGOS

[Figure]

**(a) October 2017 Antarctic distributions**

[Figure]

**October 2017 Antarctic snow volume differences relative to SNOW-KaKu**

[Figure]

**October 2017 Antarctic radar freeboard volume differences relative to RFB LEGOS**

[Figure]

**(a) May 2018 Antarctic distributions**

[Figure]

May 2018 Antarctic snow volume differences relative to SNOW-KaKu

[Figure]

May 2018 Antarctic radar freeboard volume differences relative to RFB LEGOS

[Figure]

**(a) June 2018 Antarctic distributions**

[Figure]

June 2018 Antarctic snow volume differences relative to SNOW-KaKu

[Figure]

June 2018 Antarctic radar freeboard volume differences relative to RFB LEGOS

[Figure]

**(a) July 2018 Antarctic distributions**

[Figure]

July 2018 Antarctic snow volume differences relative to SNOW-KaKu

[Figure]

July 2018 Antarctic radar freeboard volume differences relative to RFB LEGOS

[Figure]

**(a) August 2018 Antarctic distributions**

[Figure]

**August 2018 Antarctic snow volume differences relative to SNOW-KaKu**

[Figure]

**August 2018 Antarctic radar freeboard volume differences relative to RFB LEGOS**

[Figure]

**(a) September 2018 Antarctic distributions**

[Figure]

**September 2018 Antarctic snow volume differences relative to SNOW-KaKu**

[Figure]

**September 2018 Antarctic radar freeboard volume differences relative to RFB LEGOS**

[Figure]

**(a) October 2018 Antarctic distributions**

[Figure]

**October 2018 Antarctic snow volume differences relative to SNOW-KaKu**

[Figure]

**October 2018 Antarctic radar freeboard volume differences relative to RFB LEGOS**

[Figure]

**Sea ice volume in the Arctic**

The following figures, similar to Figure 8 in the manuscript, display monthly sea ice volume with tables of statistics in the Arctic for the periods January 2017-April 2017, November 2017-April 2018 and November 2018-February 2019.. Monthly sea ice volume for CS2SMOS dataset (reference) and its difference with the FREE, UNIVAR, and MULTIVAR experiments (first line) and the observations LEGOS_og (original) and LEGOS_mD (with model constant densities). Table: root mean square error (RMS) and mean difference (MD) between FREE, UNIVAR, MULTIVAR, LEGOS_og, LEGOS_md and CS2SMOS data, calculated on the LEGOS zone and outside the LEGOS zone and for CS2SMOS sea ice thickness of less than or greater than 1m. The table colors highlight the values close to 0 (white) and the extremes (green for the RMS, and blue/red for the negative/positive MD).

[Figure]

January 2017 Arctic sea ice volume

| | LEGOS zone | | | | Outside LEGOS zone | | | |
|---|---|---|---|---|---|---|---|---|
| | SIT > 1 m | | SIT < 1 m | | SIT > 1 m | | SIT < 1 m | |
| | RMS | MD | RMS | MD | RMS | MD | RMS | MD |
| FREE | 0.29 | -0.18 | 0.21 | 0.17 | 0.59 | -0.57 | 0.19 | 0.16 |
| UNIVAR | 0.33 | -0.25 | 0.16 | 0.01 | 0.71 | -0.71 | 0.13 | -0.01 |
| MULTIVAR | 0.26 | 0.19 | 0.16 | 0.13 | 0.44 | -0.37 | 0.18 | 0.16 |
| LEGOS_og | 0.21 | 0.11 | 0.19 | 0.1 | - | - | - | - |
| LEGOS_mD | 0.42 | 0.39 | 0.23 | 0.16 | - | - | - | - |

February 2017 Arctic sea ice volume

| | LEGOS zone | | | | Outside LEGOS zone | | | |
|---|---|---|---|---|---|---|---|---|
| | SIT > 1 m | | SIT < 1 m | | SIT > 1 m | | SIT < 1 m | |
| | RMS | MD | RMS | MD | RMS | MD | RMS | MD |
| FREE | 0.24 | -0.01 | 0.25 | 0.18 | 0.52 | -0.46 | 0.25 | 0.23 |
| UNIVAR | 0.27 | -0.16 | 0.22 | -0.08 | 0.69 | -0.68 | 0.15 | -0.03 |
| MULTIVAR | 0.31 | 0.25 | 0.21 | 0.16 | 0.37 | -0.11 | 0.23 | 0.2 |
| LEGOS_og | 0.22 | 0.04 | 0.24 | 0.15 | - | - | - | - |
| LEGOS_mD | 0.36 | 0.3 | 0.29 | 0.23 | - | - | - | - |

**March 2017 Arctic sea ice volume**

[Figure]

| | LEGOS zone | | | | Outside LEGOS zone | | | |
|---|---|---|---|---|---|---|---|---|
| | SIT > 1 m | | SIT < 1 m | | SIT > 1 m | | SIT < 1 m | |
| | RMS | MD | RMS | MD | RMS | MD | RMS | MD |
| FREE | 0.24 | -0.0 | 0.35 | 0.25 | 0.45 | -0.33 | 0.35 | 0.33 |
| UNIVAR | 0.3 | -0.22 | 0.26 | -0.05 | 0.63 | -0.6 | 0.16 | -0.01 |
| MULTIVAR | 0.31 | 0.22 | 0.29 | 0.23 | 0.37 | 0.06 | 0.26 | 0.21 |
| LEGOS_og | 0.25 | -0.03 | 0.28 | 0.22 | - | - | - | - |
| LEGOS_mD | 0.36 | 0.28 | 0.36 | 0.32 | - | - | - | - |

**April 2017 Arctic sea ice volume**

[Figure]

| | LEGOS zone | | | | Outside LEGOS zone | | | |
|---|---|---|---|---|---|---|---|---|
| | SIT > 1 m | | SIT < 1 m | | SIT > 1 m | | SIT < 1 m | |
| | RMS | MD | RMS | MD | RMS | MD | RMS | MD |
| FREE | 0.25 | -0.01 | 0.46 | 0.39 | 0.44 | -0.3 | 0.48 | 0.45 |
| UNIVAR | 0.32 | -0.26 | 0.28 | -0.01 | 0.65 | -0.62 | 0.16 | 0.02 |
| MULTIVAR | 0.31 | 0.22 | 0.36 | 0.31 | 0.38 | 0.05 | 0.27 | 0.22 |
| LEGOS_og | 0.28 | -0.12 | 0.36 | 0.32 | - | - | - | - |
| LEGOS_mD | 0.36 | 0.28 | 0.46 | 0.44 | - | - | - | - |

**November 2017 Arctic sea ice volume**

[Figure]

| | LEGOS zone | | | | Outside LEGOS zone | | | |
|---|---|---|---|---|---|---|---|---|
| | SIT > 1 m | | SIT < 1 m | | SIT > 1 m | | SIT < 1 m | |
| | RMS | MD | RMS | MD | RMS | MD | RMS | MD |
| FREE | 0.48 | -0.45 | 0.13 | -0.04 | 0.97 | -0.96 | 0.12 | -0.01 |
| UNIVAR | 0.71 | -0.7 | 0.2 | -0.16 | 1.16 | -1.16 | 0.13 | -0.1 |
| MULTIVAR | 0.36 | 0.24 | 0.16 | 0.12 | 0.29 | -0.09 | 0.13 | 0.1 |
| LEGOS_og | 0.27 | 0.17 | 0.29 | 0.22 | - | - | - | - |
| LEGOS_mD | 0.64 | 0.62 | 0.38 | 0.32 | - | - | - | - |

**December 2017 Arctic sea ice volume**

[Figure]

| | LEGOS zone | | | | Outside LEGOS zone | | | |
|---|---|---|---|---|---|---|---|---|
| | SIT > 1 m | | SIT < 1 m | | SIT > 1 m | | SIT < 1 m | |
| | RMS | MD | RMS | MD | RMS | MD | RMS | MD |
| FREE | 0.36 | -0.31 | 0.17 | 0.09 | 0.79 | -0.75 | 0.11 | 0.03 |
| UNIVAR | 0.61 | -0.59 | 0.16 | -0.11 | 0.99 | -0.98 | 0.13 | -0.03 |
| MULTIVAR | 0.49 | 0.44 | 0.22 | 0.19 | 0.34 | 0.18 | 0.12 | 0.08 |
| LEGOS_og | 0.24 | 0.11 | 0.29 | 0.22 | - | - | - | - |
| LEGOS_mD | 0.53 | 0.48 | 0.38 | 0.33 | - | - | - | - |

**January 2018 Arctic sea ice volume**

[Figure]

| | LEGOS zone | | | | Outside LEGOS zone | | | |
|---|---|---|---|---|---|---|---|---|
| | SIT > 1 m | | SIT < 1 m | | SIT > 1 m | | SIT < 1 m | |
| | RMS | MD | RMS | MD | RMS | MD | RMS | MD |
| FREE | 0.27 | -0.12 | 0.21 | 0.15 | 0.71 | -0.67 | 0.2 | 0.17 |
| UNIVAR | 0.43 | -0.38 | 0.18 | -0.11 | 0.92 | -0.91 | 0.13 | -0.02 |
| MULTIVAR | 0.44 | 0.38 | 0.29 | 0.26 | 0.45 | 0.29 | 0.17 | 0.15 |
| LEGOS_og | 0.27 | 0.1 | 0.32 | 0.24 | - | - | - | - |
| LEGOS_mD | 0.48 | 0.4 | 0.43 | 0.37 | - | - | - | - |

**February 2018 Arctic sea ice volume**

[Figure]

| | LEGOS zone | | | | Outside LEGOS zone | | | |
|---|---|---|---|---|---|---|---|---|
| | SIT > 1 m | | SIT < 1 m | | SIT > 1 m | | SIT < 1 m | |
| | RMS | MD | RMS | MD | RMS | MD | RMS | MD |
| FREE | 0.23 | -0.01 | 0.25 | 0.2 | 0.51 | -0.38 | 0.26 | 0.24 |
| UNIVAR | 0.33 | -0.28 | 0.19 | -0.06 | 0.7 | -0.65 | 0.11 | -0.01 |
| MULTIVAR | 0.37 | 0.29 | 0.37 | 0.34 | 0.51 | 0.4 | 0.23 | 0.21 |
| LEGOS_og | 0.25 | 0.05 | 0.39 | 0.35 | - | - | - | - |
| LEGOS_mD | 0.41 | 0.31 | 0.51 | 0.48 | - | - | - | - |

**March 2018 Arctic sea ice volume**

[Figure]

| | LEGOS zone | | | | Outside LEGOS zone | | | |
|---|---|---|---|---|---|---|---|---|
| | SIT > 1 m | | SIT < 1 m | | SIT > 1 m | | SIT < 1 m | |
| | RMS | MD | RMS | MD | RMS | MD | RMS | MD |
| FREE | 0.24 | -0.02 | 0.28 | 0.22 | 0.45 | -0.26 | 0.36 | 0.35 |
| UNIVAR | 0.37 | -0.33 | 0.22 | -0.04 | 0.63 | -0.56 | 0.15 | 0.03 |
| MULTIVAR | 0.4 | 0.3 | 0.4 | 0.37 | 0.61 | 0.52 | 0.28 | 0.26 |
| LEGOS_og | 0.27 | 0.09 | 0.39 | 0.36 | - | - | - | - |
| LEGOS_mD | 0.42 | 0.35 | 0.52 | 0.5 | - | - | - | - |

**April 2018 Arctic sea ice volume**

[Figure]

| | LEGOS zone | | | | Outside LEGOS zone | | | |
|---|---|---|---|---|---|---|---|---|
| | SIT > 1 m | | SIT < 1 m | | SIT > 1 m | | SIT < 1 m | |
| | RMS | MD | RMS | MD | RMS | MD | RMS | MD |
| FREE | 0.27 | 0.08 | 0.42 | 0.36 | 0.48 | -0.19 | 0.44 | 0.42 |
| UNIVAR | 0.34 | -0.25 | 0.26 | 0.1 | 0.62 | -0.51 | 0.14 | 0.06 |
| MULTIVAR | 0.43 | 0.35 | 0.47 | 0.45 | 0.61 | 0.49 | 0.28 | 0.24 |
| LEGOS_og | 0.28 | 0.1 | 0.45 | 0.42 | - | - | - | - |
| LEGOS_mD | 0.43 | 0.36 | 0.58 | 0.56 | - | - | - | - |

**November 2018 Arctic sea ice volume**

[Figure]

|  | LEGOS zone | | | | Outside LEGOS zone | | | |
|---|---|---|---|---|---|---|---|---|
|  | SIT > 1 m | | SIT < 1 m | | SIT > 1 m | | SIT < 1 m | |
|  | RMS | MD | RMS | MD | RMS | MD | RMS | MD |
| FREE | 0.94 | -0.93 | 0.15 | -0.07 | 1.35 | -1.34 | 0.13 | -0.07 |
| UNIVAR | 1.39 | -1.39 | 0.18 | -0.16 | 1.69 | -1.69 | 0.17 | -0.16 |
| MULTIVAR | 0.42 | 0.23 | 0.16 | 0.06 | 0.53 | -0.44 | 0.12 | 0.01 |
| LEGOS_og | 0.39 | 0.27 | 0.34 | 0.22 | - | - | - | - |
| LEGOS_mD | 1.0 | 0.96 | 0.42 | 0.32 | - | - | - | - |

**December 2018 Arctic sea ice volume**

[Figure]

|  | LEGOS zone | | | | Outside LEGOS zone | | | |
|---|---|---|---|---|---|---|---|---|
|  | SIT > 1 m | | SIT < 1 m | | SIT > 1 m | | SIT < 1 m | |
|  | RMS | MD | RMS | MD | RMS | MD | RMS | MD |
| FREE | 0.63 | -0.63 | 0.15 | 0.09 | 1.01 | -1.0 | 0.13 | 0.06 |
| UNIVAR | 0.97 | -0.97 | 0.17 | -0.13 | 1.29 | -1.29 | 0.12 | -0.07 |
| MULTIVAR | 0.77 | 0.73 | 0.23 | 0.16 | 0.29 | 0.03 | 0.15 | 0.11 |
| LEGOS_og | 0.36 | 0.2 | 0.31 | 0.16 | - | - | - | - |
| LEGOS_mD | 0.85 | 0.79 | 0.38 | 0.25 | - | - | - | - |

**January 2019 Arctic sea ice volume**

[Figure]

| | LEGOS zone | | | | Outside LEGOS zone | | | |
|---|---|---|---|---|---|---|---|---|
| | SIT > 1 m | | SIT < 1 m | | SIT > 1 m | | SIT < 1 m | |
| | RMS | MD | RMS | MD | RMS | MD | RMS | MD |
| FREE | 0.32 | -0.16 | 0.24 | 0.2 | 0.78 | -0.76 | 0.17 | 0.14 |
| UNIVAR | 0.46 | -0.4 | 0.17 | -0.04 | 1.04 | -1.04 | 0.11 | -0.05 |
| MULTIVAR | 0.48 | 0.42 | 0.31 | 0.26 | 0.38 | 0.29 | 0.17 | 0.15 |
| LEGOS_og | 0.25 | 0.05 | 0.35 | 0.22 | - | - | - | - |
| LEGOS_mD | 0.48 | 0.37 | 0.44 | 0.34 | - | - | - | - |

**February 2019 Arctic sea ice volume**

[Figure]

| | LEGOS zone | | | | Outside LEGOS zone | | | |
|---|---|---|---|---|---|---|---|---|
| | SIT > 1 m | | SIT < 1 m | | SIT > 1 m | | SIT < 1 m | |
| | RMS | MD | RMS | MD | RMS | MD | RMS | MD |
| FREE | 0.27 | -0.01 | 0.33 | 0.26 | 0.64 | -0.54 | 0.26 | 0.24 |
| UNIVAR | 0.33 | -0.26 | 0.17 | -0.01 | 0.85 | -0.81 | 0.11 | -0.02 |
| MULTIVAR | 0.33 | 0.24 | 0.39 | 0.34 | 0.47 | 0.4 | 0.23 | 0.2 |
| LEGOS_og | 0.25 | 0.04 | 0.41 | 0.33 | - | - | - | - |
| LEGOS_mD | 0.38 | 0.26 | 0.51 | 0.44 | - | - | - | - |

**Sea ice volume in the Antarctic**

The following figures, similar to Figure 9 in the manuscript, display monthly sea ice volume with tables of statistics in the Antarctic for the periods May 2017-September 2017 and May 2018 – September 2018. Monthly sea ice volume for SMOS dataset (reference) and its difference with the FREE, UNIVAR, and MULTIVAR experiments (first line) and the observations LEGOS_og (original) and LEGOS_mD (with model constant densities). Table: root mean square error (RMS) and mean difference (MD) between FREE, UNIVAR, MULTIVAR, LEGOS_og, LEGOS_md and CS2SMOS data, calculated on the LEGOS zone and outside the LEGOS zone and for CS2SMOS sea ice thickness of less than or greater than 1m. The table colors highlight the values close to 0 (white) and the extremes (green for the RMS, and blue/red for the negative/positive MD).

[Figure]

[Figure]

**July 2017 Antarctic sea ice volume, comparison with SMOS product**

[Figure]

|  | LEGOS zone | | Outside LEGOS zone | |
|---|---|---|---|---|
|  | RMS | MD | RMS | MD |
| FREE | 0.2 | -0.08 | 0.17 | 0.12 |
| UNIVAR | 0.29 | -0.26 | 0.1 | -0.05 |
| MULTIVAR | 0.35 | -0.04 | 0.16 | 0.04 |
| LEGOS_og | 0.48 | 0.39 | - | - |
| LEGOS_mD | 0.61 | 0.55 | - | - |

**August 2017 Antarctic sea ice volume, comparison with SMOS product**

[Figure]

|  | LEGOS zone | | Outside LEGOS zone | |
|---|---|---|---|---|
|  | RMS | MD | RMS | MD |
| FREE | 0.2 | 0.01 | 0.23 | 0.21 |
| UNIVAR | 0.3 | -0.25 | 0.1 | -0.03 |
| MULTIVAR | 0.38 | 0.0 | 0.17 | 0.03 |
| LEGOS_og | 0.61 | 0.55 | - | - |
| LEGOS_mD | 0.78 | 0.73 | - | - |

**September 2017 Antarctic sea ice volume, comparison with SMOS product**

[Figure]

| | LEGOS zone | | Outside LEGOS zone | |
|---|---|---|---|---|
| | RMS | MD | RMS | MD |
| FREE | 0.24 | 0.16 | 0.32 | 0.31 |
| UNIVAR | 0.26 | -0.2 | 0.1 | -0.02 |
| MULTIVAR | 0.38 | 0.21 | 0.19 | 0.07 |
| LEGOS_og | 0.78 | 0.73 | - | - |
| LEGOS_mD | 0.97 | 0.94 | - | - |

**May 2018 Antarctic sea ice volume, comparison with SMOS product**

[Figure]

| | LEGOS zone | | Outside LEGOS zone | |
|---|---|---|---|---|
| | RMS | MD | RMS | MD |
| FREE | 0.22 | -0.1 | 0.05 | 0.0 |
| UNIVAR | 0.26 | -0.24 | 0.02 | -0.01 |
| MULTIVAR | 0.24 | 0.1 | 0.04 | 0.02 |
| LEGOS_og | 0.62 | 0.58 | - | - |
| LEGOS_mD | 0.84 | 0.81 | - | - |

**June 2018 Antarctic sea ice volume, comparison with SMOS product**

[Figure]

|  | LEGOS zone | | Outside LEGOS zone | |
|---|---|---|---|---|
|  | RMS | MD | RMS | MD |
| FREE | 0.24 | -0.07 | 0.08 | 0.05 |
| UNIVAR | 0.31 | -0.28 | 0.03 | -0.02 |
| MULTIVAR | 0.35 | 0.19 | 0.05 | 0.03 |
| LEGOS_og | 0.63 | 0.57 | - | - |
| LEGOS_mD | 0.79 | 0.76 | - | - |

**July 2018 Antarctic sea ice volume, comparison with SMOS product**

[Figure]

|  | LEGOS zone | | Outside LEGOS zone | |
|---|---|---|---|---|
|  | RMS | MD | RMS | MD |
| FREE | 0.23 | 0.0 | 0.1 | 0.07 |
| UNIVAR | 0.31 | -0.28 | 0.04 | -0.02 |
| MULTIVAR | 0.45 | 0.22 | 0.1 | 0.06 |
| LEGOS_og | 0.66 | 0.6 | - | - |
| LEGOS_mD | 0.83 | 0.79 | - | - |

**August 2018 Antarctic sea ice volume, comparison with SMOS product**

[Figure]

| | LEGOS zone | | Outside LEGOS zone | |
|---|---|---|---|---|
| | RMS | MD | RMS | MD |
| FREE | 0.27 | 0.11 | 0.16 | 0.15 |
| UNIVAR | 0.28 | -0.23 | 0.04 | -0.0 |
| MULTIVAR | 0.48 | 0.25 | 0.12 | 0.08 |
| LEGOS_og | 0.82 | 0.78 | - | - |
| LEGOS_mD | 1.03 | 1.0 | - | - |

**September 2018 Antarctic sea ice volume, comparison with SMOS product**

[Figure]

| | LEGOS zone | | Outside LEGOS zone | |
|---|---|---|---|---|
| | RMS | MD | RMS | MD |
| FREE | 0.27 | 0.17 | 0.23 | 0.22 |
| UNIVAR | 0.26 | -0.2 | 0.04 | 0.01 |
| MULTIVAR | 0.51 | 0.35 | 0.17 | 0.14 |
| LEGOS_og | 0.83 | 0.8 | - | - |
| LEGOS_mD | 1.06 | 1.03 | - | - |